# Dynamic regulation of integrin β1 phosphorylation supports invasion of breast cancer cells

James R. W. Conway [1,8] ✉, Omkar Joshi [1], Jasmin Kaivola [1], Gautier Follain [1,2,3], Michalis Gounis [1], David Kühl[1] & Johanna Ivaska [1,4,5,6,7] ✉

Integrins provide an essential bridge between cancer cells and the extracellular matrix, playing a central role in every stage of disease progression. Despite the recognized importance of integrin phosphorylation in several biological processes, the regulatory mechanisms and their relevance remained elusive. Here we engineer a fluorescence resonance energy transfer biosensor for integrin β1 phosphorylation, screening 96 protein tyrosine phosphatases and identifying Shp2 and PTP-PEST as negative regulators to address this gap. Mutation of the integrin NPxY(783/795) sites revealed the importance of integrin phosphorylation for efficient cancer cell invasion, further supported by inhibition of the identified integrin phosphorylation regulators Shp2 and Src kinase. Using proteomics approaches, we uncovered Cofilin as a component of the phosphorylated integrin-Dok1 complex and linked this axis to effective invadopodia formation, a process supporting breast cancer invasion. These data further implicate dynamic modulation of integrin β1 phosphorylation at NPxY sites at different stages of metastatic dissemination.

Cellular adhesion to the extracellular matrix (ECM) is an essential prerequisite for multicellular organisms. Remodelling of this scaffold supports resistance to extracellular stresses and provides a reproducible environment to guide tissue functions. Thus, cell-ECM interactions are vital and are mediated by integrin adhesion receptors. Integrins form a diverse family, with 24 heterodimers arising from 18 α and 8 β subunit pairings[1]. Central to this family is integrin β1, which accounts for half of αβ receptors and exclusively mediates binding to several ECM components. Given this vital role in cell–ECM interactions, it is unsurprising that loss of integrin β1 is early embryonic lethal[2,3]. Similarly, two conserved tyrosine-motifs (NPxY) in the cytoplasmic domain are essential for integrin β1 function. Mutation to NPxA (YYAA) renders the receptor inactive and unstable, resulting in embryonic

lethality, whereas the non-phosphorylatable NPxF (YYFF) mutation shows a milder phenotype[4–6] (Fig. 1a). In a cancer setting, however, YYFF mutation reduces baseline signal transduction[7,8] and FAK activation and delays tumourigenesis[9]. Together, these data highlight the importance of the NPxY sites and integrin phosphorylation for receptor functionality.

Despite this established link to tumour initiation and growth, the role of integrin phosphorylation in cancer invasion has not been investigated. Furthermore, the molecular components recruited to the phosphorylated integrin are largely unknown, and the same is true for the regulatory kinases and phosphatases. This is due to a lack of tools to interrogate integrin β1 phosphorylation, which previously relied solely on suboptimal phospho-specific antibodies and mutagenesis.

[1]Turku Bioscience Centre, University of Turku and Åbo Akademi University, Turku, Finland. [2]Faculty of Science and Engineering, Cell Biology, Åbo Akademi University, Turku, Finland. [3]Turku Collegium for Science, Medicine and Technology TCSMT, University of Turku, Turku, Finland. [4]Department of Life Technologies, University of Turku, Turku, Finland. [5]InFLAMES Research Flagship Center, University of Turku, Turku, Finland. [6]Western Finnish Cancer Center, University of Turku, Turku, Finland. [7]Foundation for the Finnish Cancer Institute, Helsinki, Finland. [8]Present address: Department of Biochemistry and Developmental Biology, Faculty of Medicine, University of Helsinki, Helsinki, Finland. ✉e-mail: jdconw@utu.fi; joivaska@utu.fi

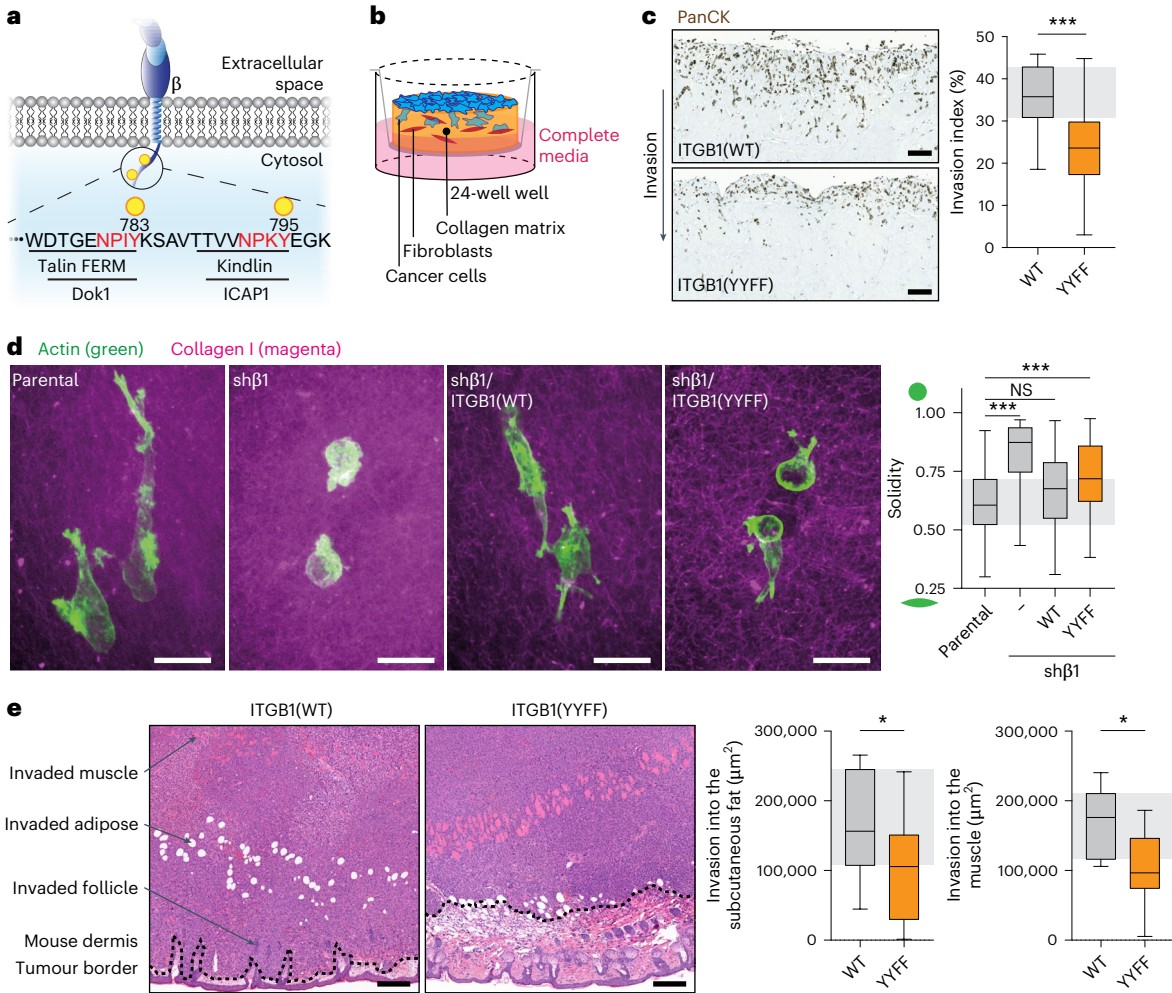

**Fig. 1 | ITGB1 phosphorylation supports efficient cancer cell invasion. a**, A schematic highlighting the membrane-proximal and membrane-distal NPxY motifs of ITGB1. **b**, A schematic of the fibroblast-contracted 3D collagen matrix invasion assay platform. **c**, Left: representative images of MM231 triple-negative breast cancer cells expressing either ITGB1(WT or YYFF), invading into 3D fibroblast-contracted collagen I and stained with pan-cytokeratin (PanCK) to specifically detect the cancer cells and exclude fibroblasts from the analysis. Scale bars, 100 μm. Right: quantification of invasion beyond 100 μm, normalized to the total number of cells per region (n = 24 regions per cell line pooled from three biological replicates; unpaired two-tailed Student's *t*-test with a Welch's correction). **d**, Representative images (left) and quantification of cell shape (that is, solidity) (right) of MM231 cells (green is the actin staining) embedded in 3D collagen matrices overnight (magenta) (n = 104 (parental), 76 (shβ1), 84 (ITGB1(WT)) and 116 (ITGB1(YYFF)) cells pooled from three biological replicates; Kruskal–Wallis test with a Dunn's correction for multiple comparisons). Scale bars, 20 μm. **e**, Representative H&E images (left) and quantification (right) of local invasion of MM231 ITGB1(WT or YYFF) cells into the mouse dermis from subcutaneous xenografts. Scale bars, 200 μm (n = 9 ITGB1(WT) or 11 ITGB1(YYFF) mice, respectively; unpaired two-tailed Student's *t*-test with a Welch's correction). The dotted line indicates the tumour boundary, where it meets the subcutaneous stromal cells. The boxplots represent the median and interquartile range (IQR). The whiskers extend to the minimum and maximum values. The grey areas on the boxplots highlight the IQR of the control conditions. NS, not significant. *P < 0.05, ***P < 0.001.

Early work in Rous-sarcoma virus transformed fibroblasts showed phosphorylated integrin β1 peptide localization in podosomes[10] but only hinted at the involvement of the Src family kinases in integrin phosphorylation. Later, integrin phosphorylation was demonstrated as a conserved mechanism regulating the recruitment of cytoplasmic components to the receptor. The best characterized of these components is Dok1, the only known adaptor protein to preferentially bind phosphorylated integrin β1 (ref. 11). By contrast, the non-phosphorylated receptor supports binding of talin, and while there is a well-established recruitment of proteins to talin through a stepwise process of ECM adhesion maturation and linkage to the actin cytoskeleton[12,13], the role and components of the phosphorylated integrin β1/Dok1 complex were unknown. In this work, we uncover a regulatory network of kinases and protein tyrosine phosphatases (PTPs) that dynamically recruit a phosphorylation-sensitive protein complex to mediate efficient cancer cell invasion.

## Results

### Integrin phosphorylation directs breast cancer cell invasion

To model integrin β1 (hereafter referred to as ITGB1) phosphorylation during invasive breast cancer progression, we knocked down the *ITGB1* mRNA in MDA-MB-231 (MM231) breast cancer cells using a lentiviral short-hairpin RNA (shRNA) (MM231 shβ1 cells; Extended Data Fig. 1a). We then reexpressed mRuby2-tagged ITGB1 wild-type (WT) or a variant with the cytoplasmic NPxY sites mutated to NPxF (that is, Y783F and Y795F, referred to as YYFF hereafter) (Fig. 1a) to mimic loss of ITGB1 phosphorylation at these sites. These integrin-reexpressing shβ1 MM231 cells are used throughout the study and are referred to as MM231 ITGB1(WT or YYFF). Of note, these cell lines showed no proliferative defects (Extended Data Fig. 1b,c). They also had comparable ITGB1 cell-surface levels, with no compensatory upregulation of integrin β3, as found in some cell line models[14,15] (Extended Data Fig. 1e and quantified in Extended Data Fig. 1f). Similarly, there were no significant

differences in integrin activation state nor the size and number of integrin adhesion complexes (IACs) (Extended Data Fig. 1g and quantified in Extended Data Fig. 1h,i). However, MM231 ITGB1(YYFF) cell invasion into a fibroblast-contracted three-dimensional (3D) collagen matrix[16] was significantly reduced (Fig. 1b,c), without impacting cell proliferation (Ki67 staining and ratio of negative-to-positive nuclei) (Extended Data Fig. 1j). In line with this invasion defect, we observed decreased cell spreading in 3D with *ITGB1*-knockdown (KD) MM231s (shβ1) and MM231 ITGB1(YYFF) cells, compared with parental cells (Fig. 1d). To assess the role of ITGB1 phosphorylation on local invasion in vivo, we subcutaneously xenografted MM231 ITGB1(WT or YYFF) cells and detected a significant reduction in MM231 ITGB1(YYFF) cell invasion into the subcutaneous fat and muscle (Fig. 1e), while again observing no significant effect on proliferation (Ki67 staining) (Extended Data Fig. 1k). Cumulatively, these data demonstrate the importance of the ITGB1 NPxY sites as essential mediators of effective breast cancer invasion.

## Generation of the first integrin phosphorylation reporter

The regulators of integrin phosphorylation are primarily unknown, largely owing to a lack of effective tools to interrogate phosphorylation in intact cells. Yet, we see that loss of ITGB1 phosphorylation at the NPxY sites has a profound effect on breast cancer invasion (Fig. 1), emphasizing the importance of this poorly understood aspect of integrin biology. To address this mechanistic gap, we developed a fluorescent reporter for ITGB1 phosphorylation based on the established models for substrate-based fluorescence resonance energy transfer (FRET) reporters for Src and other kinases[17,18]. The ITGB1 phosphorylation FRET reporter (hereafter referred to as 'Illusia', inspired by Finnish mythology[19]) relies upon a conformational change that occurs through an interaction between two domains of the reporter, the phosphotyrosine (pY) binding domain (PTB) from Dok1 and the C-terminal amino acids from ITGB1 (Fig. 2a). The interaction is mediated by phosphorylation of the ITGB1 NPxY sites, included in Illusia, which pulls apart the donor (mTurquoise2) and acceptor (YPet) FRET pair, a conformational change detected by a decrease in FRET efficiency and a corresponding increase in the fluorescence lifetime of the donor. Illusia supports live assessment of the kinase/phosphatase balance in the cell. However, it is important to note that Illusia reflects the phosphorylation status of the integrin tail motif (as a freely diffusing, membrane-tethered ITGB1 tail-like substrate), rather than directly reporting ITGB1 phosphorylation.

To ensure that the phosphorylation-dependent binding of the Dok1 PTB would be reversible, we assessed the effect of Dok1 overexpression on ITGB1 and found no increase in phosphorylation on the membrane-proximal tyrosine, suggesting that Dok1 does not play a protective role for the phosphorylation (Extended Data Fig. 2a). Similarly, we validated the phosphorylation-dependent binding of Dok1 in cellulo using an intermolecular FRET assay between the mRuby2-tagged ITGB1

WT and YYFF mutant receptors (Extended Data Fig. 2b). This confirmed that the interaction between ITGB1 and Dok1 was significantly reduced in the absence of phosphorylation in MM231 ITGB1(YYFF) cells (Extended Data Fig. 2c). Notably, the reverse was observed for a talin-1 head domain fragment ($F_0$–$F_3$ from mouse *Tln1*), which showed an increased interaction with the non-phosphorylatable YYFF receptor (Extended Data Fig. 2d) and is in line with talin-1 having a reduced affinity for phosphorylated integrins β1 and β3 (ref. [11]).

## Illusia demonstrates direct Src phosphorylation of ITGB1

While earlier reports have implied Src phosphorylation of integrins[20,21], Abl2/Arg (Abl-related gene) is the only tyrosine kinase that has been shown to directly phosphorylate ITGB1 on the membrane-proximal NPxY(783) site[22]. Thus, initial validation of Illusia was performed through overexpression of the non-receptor tyrosine kinase Arg. Here, overexpression of the WT but not the kinase-dead mutant (K281M) Arg significantly reduced Illusia FRET, indicating a phosphorylation-sensitive change in the reporter conformation (Extended Data Fig. 2f and quantified in Extended Data Fig. 2e). The parallel western blots confirmed that this change occurred in conjunction with an increase in ITGB1 phosphorylation on the NPxY(783) site (Extended Data Fig. 2g and quantified in Extended Data Fig. 2h). It has also been suggested that the cytoplasmic domain of ITGB1 is a substrate for Src kinase[20,21], with similar work demonstrating that Src phosphorylation of integrin β3 reduces receptor engagement with fibronectin[23]. Accordingly, we explored this link in cells using the Illusia FRET reporter, inducing expression of WT, constitutively active (Y527F and E378G) or kinase-dead (K295R) Src. We observed a decrease in FRET that was concordant with an increase in ITGB1 phosphorylation with the WT and constitutively active Src variants (Fig. 2b). This was reproduced using antibody-based approaches and correlated with an increase in the phosphorylation of p130Cas, a canonical Src substrate (Fig. 2c and quantified in Fig. 2d). Similarly, we developed an enzyme-linked immunosorbent assay (ELISA)-based method to assess the ability of recombinant proteins to alter the phosphorylation state of ITGB1 tail peptides (Fig. 2e), finding that Src directly phosphorylates these peptides (Fig. 2f). Moreover, through small-molecule inhibition of Src and Abl family kinases using saracatenib (Sara), we observed a significant decrease in both Src activation and ITGB1 phosphorylation (Extended Data Fig. 2i and quantified in Extended Data Fig. 2j). This was also detected with Illusia in MM231s and in telomerase-immortalized human fibroblasts (TIFs) (Fig. 2g). Altogether, we demonstrate ITGB1 as a direct substrate for Src and Arg kinases using Illusia, further supporting an active role for phosphorylation in integrin functions.

## ITGB1 is actively dephosphorylated by PTPs

Given the role of Src and Arg kinases in regulating ITGB1 phosphorylation, we further investigated the dynamics of this post-translational

**Fig. 2 | Src is a regulatory kinase for the ITGB1 NPxY motifs. a**, A schematic representation of the possible conformations for the ITGB1 intramolecular FRET biosensor (Illusia), where the mTurquoise2–YPet FRET pair is separated by the PTB from Dok1, a linker and the cytoplasmic domain (aa772–798) from ITGB1 (including the two NPxY motifs). Illusia is recruited to the membrane through an acylation substrate sequence derived from Lyn kinase. Ex, excitation; Em, emission. **b**, Representative FLIM images (left) and quantification of apparent FRET efficiency (right) of MM231 cells stably expressing Illusia after Dox-inducible overexpression of either Src(WT), kinase-dead Src(K295R) or constitutively active Src(Y527F)/Src(E378G) ($n = 60$ cells in all conditions with the exception of 70 for Src(Y527F) (−)Dox and 65 for Src(Y527F) (+)Dox pooled from three biological replicates; one-way analysis of variance (ANOVA) with a Šidák correction for multiple comparisons). Scale bars, 10 μm. **c**, Representative western blot of MM231 cells after Dox-inducible Src(WT), Src(K295R), Src(Y527F) or Src(E378G) overexpression. **d**, Densitometry analysis of western blots from **c** ($n = 4$ biological replicates; one-sample two-tailed *t*-test against the normalized

control values for each cell line without Dox). The data are mean ± s.e.m. **e**, A scheme of an ELISA for pY. **f**, An ELISA for changes in ITGB1 phosphorylation using recombinant ITGB1 peptide and Src kinase in the absence or presence of ATP and the Src inhibitor Sara (1 μM; $n = 3$ biological replicates; triplicate wells/replicates; one-way ANOVA with a Šidák correction for multiple comparisons). Unphosphorylated ITGB1 and phosphorylated ITGB1 p(Y783) peptides were included as negative and positive controls, respectively. The data are the mean ± s.e.m. **g**, Representative FLIM–FRET images (left) and quantification (right) of MM231 and TIF cells with stable expression of Illusia and Sara treatment (1 μM) for 24 h (MM231s, $n = 129$ (DMSO) and 125 (Sara) cells pooled from five biological replicates; TIFs, $n = 74$ (DMSO) and 75 (Sara) cells pooled from three biological replicates; unpaired two-tailed Student's *t*-test with a Welch's correction). Scale bars, 10 μm. The boxplots represent the median and IQR. The whiskers extend to the minimum and maximum values. The grey areas on the boxplots highlight the IQR of the control conditions. NS, not significant. *$P < 0.05$, ***$P < 0.001$.

modification by focusing on the opposing role of PTPs (Fig. 3a). Applying the broad-spectrum PTP inhibitor, sodium orthovanadate ($VO_4^{3-}$), we observed a rapid accumulation of ITGB1 phosphorylation over the treatment time course (Fig. 3b and quantified in Fig. 3c). A similar effect was again observed in the MM231 ITGB1(WT) and not in the ITGB1(YYFF)-expressing cells (Fig. 3d). Increased integrin phosphorylation was also evident by bulk pY immunoprecipitation (IP) and

subsequent blotting for ITGB1 following $VO_3^{3-}$ treatment (Extended Data Fig. 3a,b). Concordant with these data, $VO_4^{3-}$ treatment significantly reduced Illusia FRET efficiency, and mutation to a non-phosphorylatable YYFF reporter abolished this effect, confirming that the changes upon treatment are phosphorylation-specific in both the MM231 (Fig. 3e) and TIF (Extended Data Fig. 3c,d) cell lines. Furthermore, additional mutations of the NPxY sites of Illusia supported

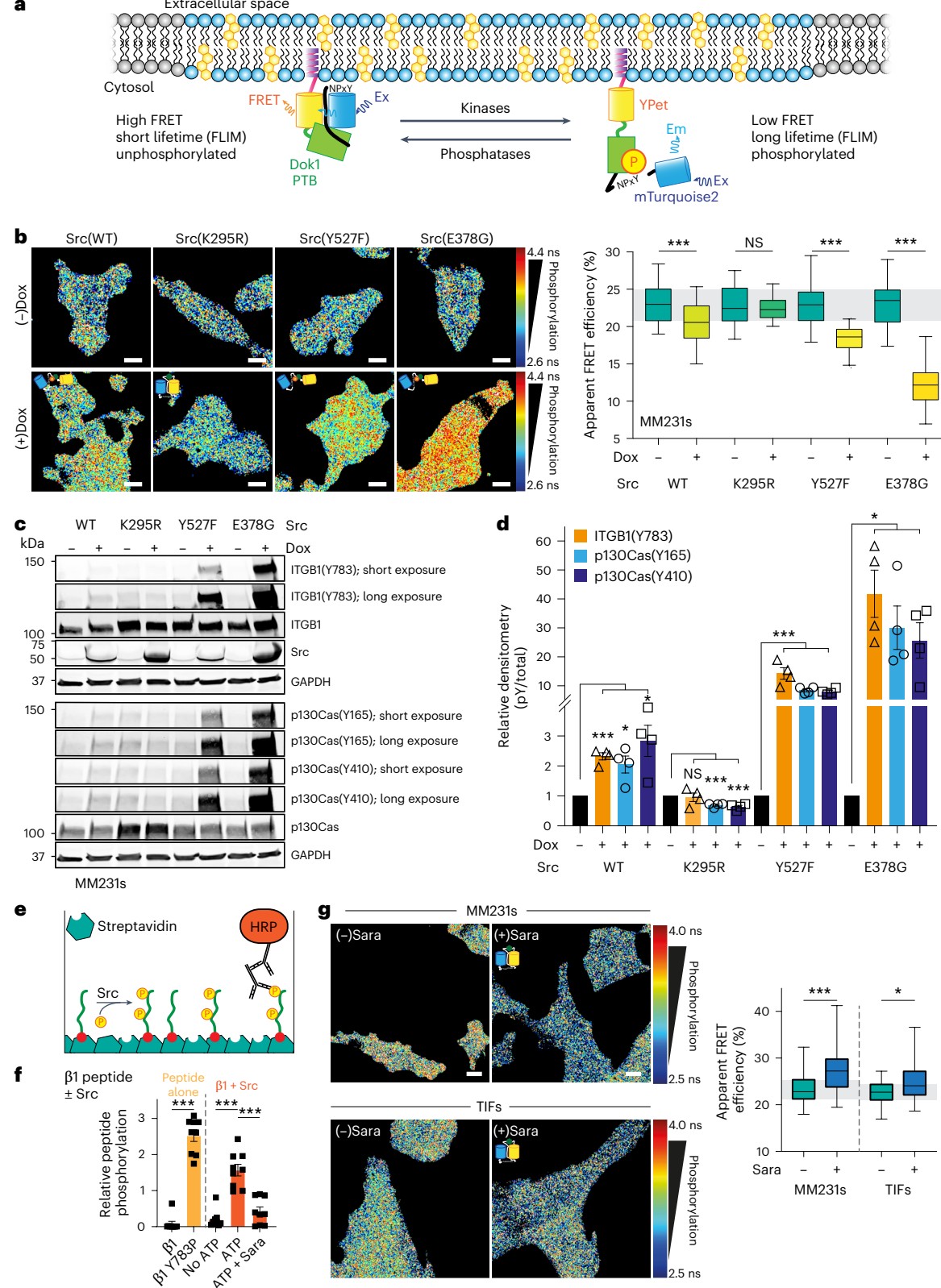

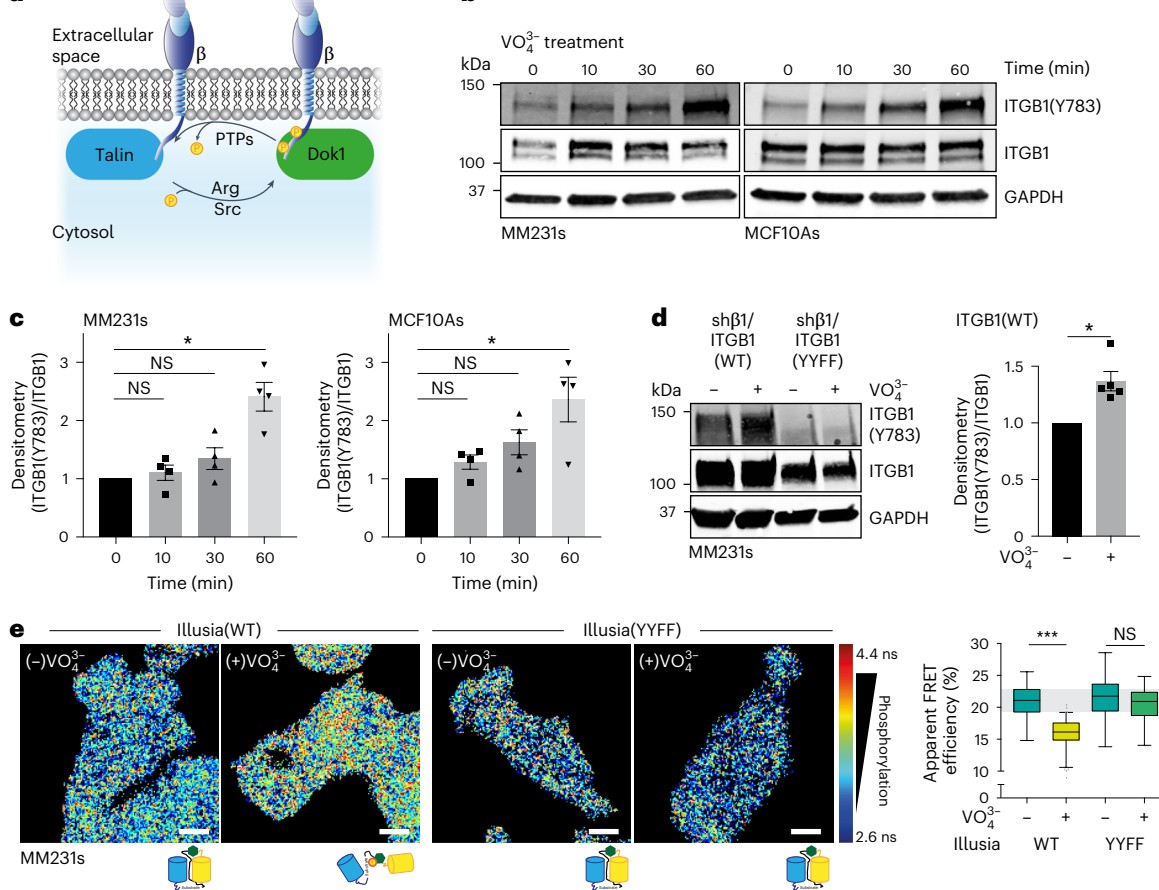

**Fig. 3 | PTPs actively regulate phosphorylation of the ITGB1 NPxY motifs. a**, A schematic of phosphorylation-dependent Dok1 recruitment to ITGB1. **b,c**, Representative western blots (**b**) and densitometry (**c**) of MM231 (left) or MCF10A (right) cells treated with the broad-spectrum PTP inhibitor sodium orthovanadate ($VO_4^{3-}$; 100 μM, 2 h; $n = 4$ biological replicates; one-sample two-tailed $t$-test against the normalized control value without $VO_4^{3-}$). The data are the mean ± s.e.m. **d**, A representative western blot (left) of MM231 ITGB1(WT or YYFF) cells treated for 2 h with $VO_4^{3-}$ (100 μM) and densitometry analysis (right) of MM231 ITGB1(WT) cells ($n = 5$ biological replicates; one-sample two-tailed $t$-test against the normalized control value without $VO_4^{3-}$). The data are the

mean ± s.e.m. **e**, Representative FLIM images (left) and quantification of apparent FRET efficiency (right) after $VO_4^{3-}$ treatment (100 μM, 2 h) of MM231 cells stably expressing Illusia(WT) or a non-phosphorylatable mutant reporter Illusia(YYFF) ($n = 88$ (Illusia(WT) ($-VO_4^{3-}$)), 88 (Illusia(WT) ($+VO_4^{3-}$)), 93 (Illusia(YYFF) ($-VO_4^{3-}$)) and 86 (Illusia(YYFF) ($+VO_4^{3-}$)) cells pooled from four biological replicates; one-way analysis of variance with a Šidák correction for multiple comparisons). n.s., not significant. Scale bars, 10 μm. The boxplot represents the median and IQR. The whiskers extend to the minimum and maximum values. The grey area on the boxplot highlights the IQR of the control condition. NS, not significant, *$P < 0.05$, ***$P < 0.001$.

the phosphorylation dependence of the FRET change and found that the Dok1 PTB predominantly binds to the NPxY(783) site (Extended Data Fig. 3e), as described previously[11]. Importantly, Illusia remained intact upon treatment of MM231 or TIF cells with $VO_4^{3-}$ (Extended Data Fig. 3f) or Sara (Extended Data Fig. 3g). In addition, Illusia did not coimmunoprecipitate ITGB1, suggesting no, weak or transient binding (Extended Data Fig. 3h). Cumulatively, these data support the phosphorylation-sensitive changes of Illusia and that active dephosphorylation of ITGB1 is dynamically balanced with kinase-mediated phosphorylation in cancer and normal cells (Fig. 3a).

**A high-content siRNA screen identifies putative ITGB1 PTPs**
To identify the PTPs responsible for ITGB1 dephosphorylation, MM231 cells stably expressing Illusia were transfected, using an optimized protocol (Extended Data Fig. 4a and quantified in Fig. 4b), with a small interfering RNA (siRNA) library against 96 of the 108 predicted PTPs in the human genome[24]. The RNA interference (RNAi) screen identified 18 PTPs where siRNA KD significantly decreased the FRET efficiency of Illusia, suggesting that they may be direct PTPs for ITGB1 (Fig. 4a). Unexpectedly, 57 PTP siRNAs significantly increased the FRET efficiency of Illusia, suggesting an indirect role of the phosphatase in

upregulating ITGB1 phosphorylation (Fig. 4a and complete screen data in Supplementary Table 1). Notably, the silencing of the 16 most promising hits was efficient (Extended Data Fig. 4c). To further focus on those hits that were most likely to be direct ITGB1 PTPs, we filtered hits based on those that had been previously identified as part of the integrin meta-adhesome, linked to the consensus adhesome[25]. *PTPN11* (Shp2) and *PTPN12* (PTP-PEST) were selected for downstream validation (the representative images from the screen in Fig. 4a are provided in Extended Data Fig. 4d) and changes in ITGB1 phosphorylation confirmed by fluorescence lifetime imaging microscopy (FLIM)–FRET (Fig. 4b (*PTPN11*) and 4c (*PTPN12*)) upon efficient silencing of either PTP (Extended Data Fig. 4e,f and quantified in Fig. 4d,e).

**Shp2 and PTP-PEST are direct PTPs for ITGB1**
The data above demonstrate that loss of either Shp2 (*PTPN11*) or PTP-PEST (*PTPN12*) increases ITGB1 phosphorylation; however, the interaction between ITGB1 and these PTPs had not been assessed. Initially, we validated the ability of recombinant Shp2 and PTP-PEST to dephosphorylate phosphorylated ITGB1 tail peptides, detecting free phosphate released through this interaction (Fig. 5a,b). To confirm this interaction in cellulo, we employed expression constructs

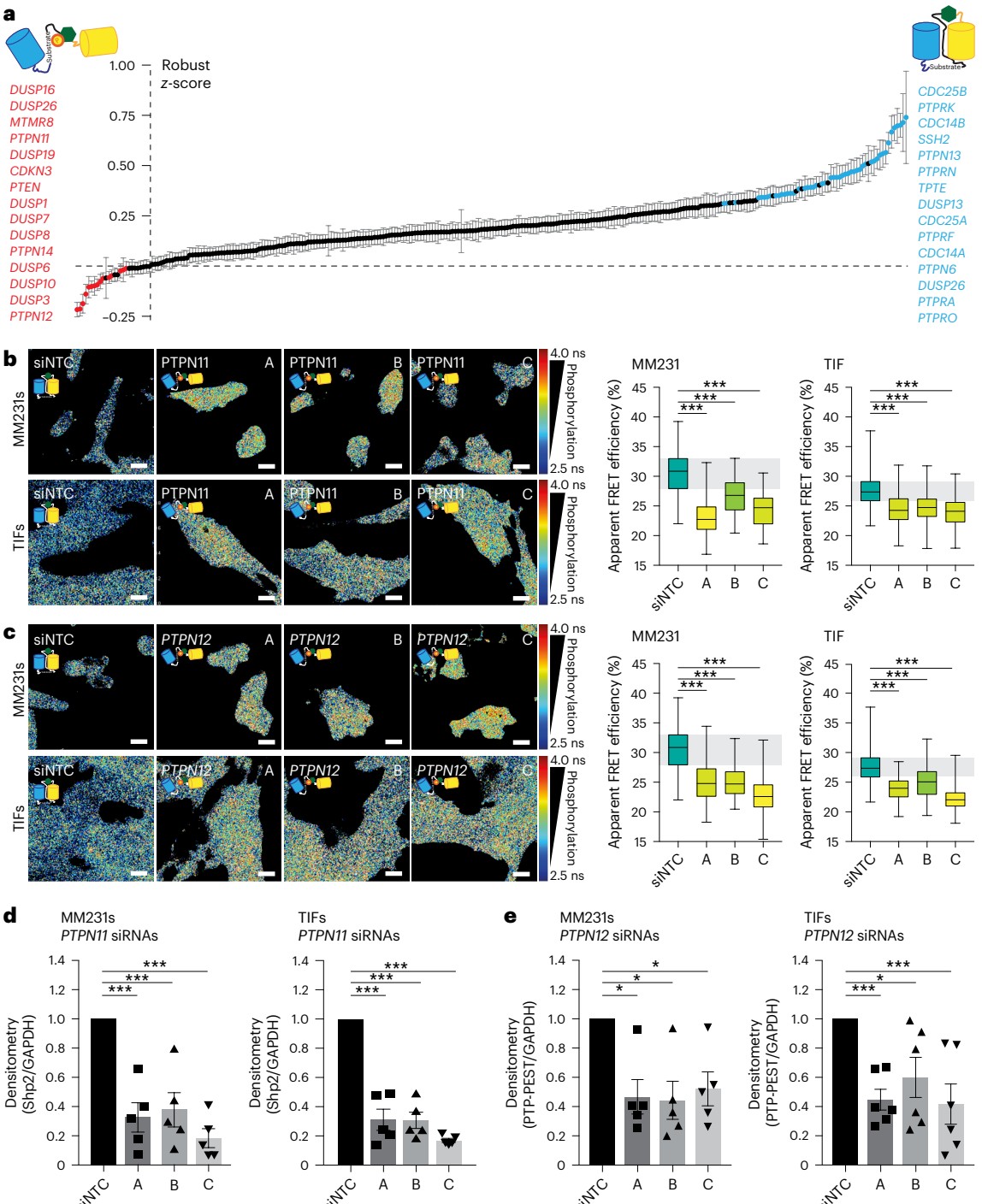

**Fig. 4 | An RNAi FRET screen to identify regulatory PTPs for ITGB1 NPxY sites. a**, A waterfall plot from an RNAi sensitized emission (SE)-FRET screen of MM231 cells stably expressing Illusia to assess changes after KD of 96 out of the 108 PTPs in the human genome, using three siRNAs/target (A, B and C) ($n = 3$; one-way analysis of variance (ANOVA) with a Tukey correction for multiple comparisons after normalizing replicates into robust $z$-scores). PTPs that significantly ($P < 0.01$) increased phosphorylation upon KD are indicated with red text and circles (15 hits), while those that significantly ($P < 0.01$) decreased phosphorylation upon KD are indicated with blue text and circles (15 top hits). The data are the mean ± s.e.m. **b**,**c**, Representative FLIM–FRET images (left) and quantification (right) of MM231 and TIF cells with stable Illusia expression and transfected with siRNAs (A, B or C) against either *PTPN11* (Shp2) (**b**) or *PTPN12* (PTP-PEST) (**c**) (MM231 *PTPN11* silencing, $n = 100$ (siNTC), 98 (siRNA A), 100

(siRNA B) and 100 (siRNA C); *PTPN12* silencing, $n = 99$ (siNTC), 99 (siRNA A), 98 (siRNA B) and 97 (siRNA C) cells pooled from three biological replicates; TIF *PTPN11* silencing, $n = 75$ (siNTC), 76 (siRNA A), 76 (siRNA B) and 78 (siRNA C); *PTPN12* silencing, $n = 74$ (siNTC), 74 (siRNA A), 77 (siRNA B) and 74 (siRNA C) cells pooled from four biological replicates; one-way ANOVA with a Dunnett correction for multiple comparisons). n.s., not significant. Scale bars, 20 μm. The boxplots represent the median and IQR. The whiskers extend to the minimum and maximum values. The grey areas on the boxplots highlight the IQR of the control conditions. siNTC, non-targeting control siRNA. **d**,**e**, Densitometry from western blots of *PTPN11* (representative western blot) (**d**) (Extended Data Fig. 4e) and *PTPN12* (representative western blot) (**e**) (Extended Data Fig. 4f) siRNA KD in MM231 ($n = 5$ biological replicates) and TIF ($n = 5$ (Shp2) or 6 (PTP-PEST) biological replicates) cells. The data are the mean ± s.e.m. *$P < 0.05$, ***$P < 0.001$.

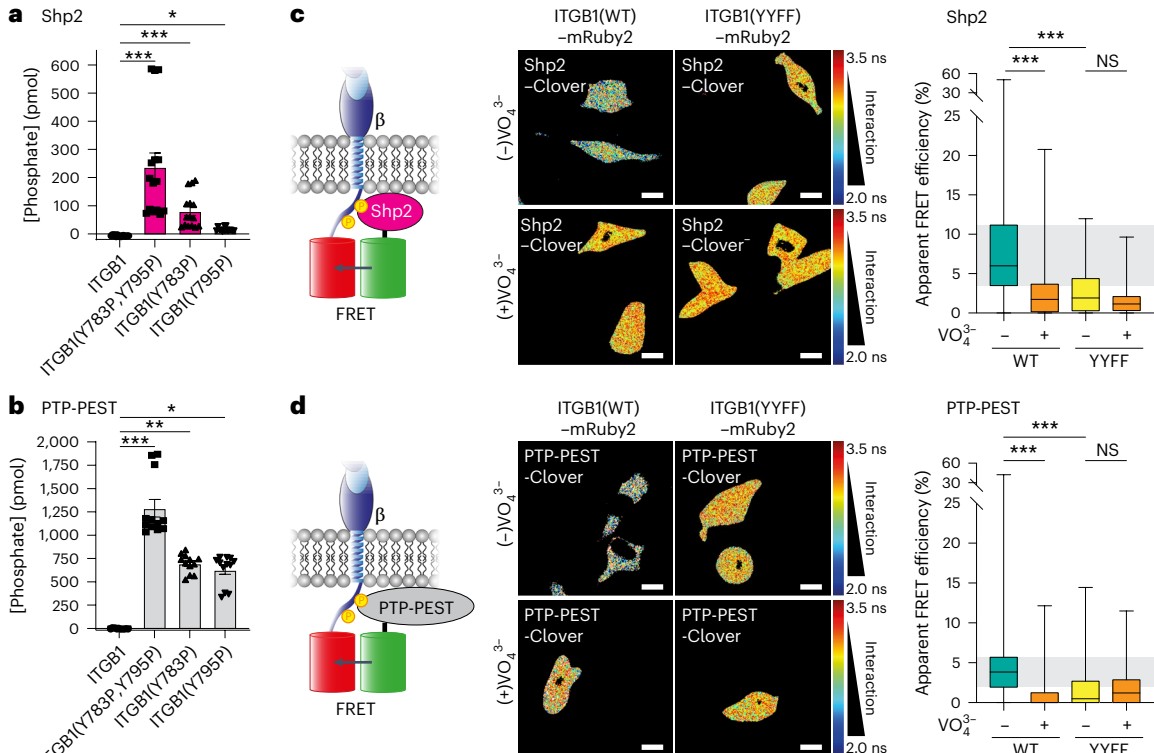

**Fig. 5 | ITGB1 is a substrate for PTP-PEST and Shp2. a,b,** A malachite green assay for free phosphate release after incubation of phosphorylated/non-phosphorylated ITGB1 peptides with recombinant Shp2 (*n* = 5 independent replicates, each performed in triplicate) (**a**) or PTP-PEST (*n* = 4 independent replicates, each performed in triplicate) (**b**). The significance was assessed using a Kruskal–Wallis test with a Dunn's correction for multiple comparisons. The data are presented as the mean ± s.e.m. **c,d,** Schematics of FRET experiments (left) using mRuby2-tagged ITGB1 and Clover-tagged PTPs. Representative FLIM–FRET images (right) and quantification of apparent FRET efficiency of MM231 cells with stable expression of either ITGB1(WT)–mRuby2 or ITGB1(YYFF)–mRuby2

transfected with either Shp2–Clover (**c**) or PTP-PEST-Clover (**d**) and treated with $VO_4^{3-}$ (100 μM, 2 h) (for **c**, *n* = 73 (ITGB1(WT) (−)$VO_4^{3-}$), 62 (ITGB1(WT) (+)$VO_4^{3-}$), 64 (ITGB1(YYFF) (−)$VO_4^{3-}$) and 67 (ITBG1(YYFF) (+)$VO_4^{3-}$) cells pooled from three biological replicates; for **d**, *n* = 75 (ITGB1(WT) (−)$VO_4^{3-}$), 58 (ITGB1(WT) (+)$VO_4^{3-}$), 58 (ITGB1(YYFF) (−)$VO_4^{3-}$) and 65 (ITBG1(YYFF) (+)$VO_4^{3-}$) cells pooled from three biological replicates; one-way analysis of variance with a Tukey correction for multiple comparisons). Scale bars, 20 μm. The boxplots represent the median and IQR. The whiskers extend to the minimum and maximum values. The grey areas highlight the IQR of the control conditions. NS, not significant. *$P$ < 0.05, **$P$ < 0.01, ***$P$ < 0.001.

tagged with fluorescent proteins ideal for FLIM–FRET. This enabled us to observe that both PTPs directly interact with the ITGB1 tail (Fig. 5c,d), which was not observed with the control construct (clover tag alone) (Extended Data Fig. 5a). Furthermore, overexpression of Shp2 WT or a phosphatase-dead mutant in Illusia-expressing MM231 cells demonstrated a clear dependence on the phosphatase activity for the changes observed in ITGB1 phosphorylation by FLIM–FRET (Extended Data Fig. 5b) and western blot (Extended Data Fig. 5c) approaches. This approach yielded similar results for PTP-PEST (Extended Data Fig. 5d–f). Given that Shp2 is under clinical investigation for mutant-KRAS-driven cancers, several small molecules exist for specific inhibition of this PTP[26]. A clinically relevant Shp2 inhibitor, SHP099, significantly upregulated ITGB1 phosphorylation in Illusia-expressing MM231 and TIF cells through both FLIM–FRET (Extended Data Fig. 5g,h) and western blot approaches (Extended Data Fig. 5i), further validating Shp2 as an important regulator of ITGB1 dephosphorylation.

### Disrupting ITGB1 phosphorylation reduces cell invasion

Our initial findings demonstrate that loss of ITGB1 phosphorylation reduces breast cancer invasion in vitro and in vivo (Fig. 1). Breast cancer progression entails increased tissue rigidity impacting many aspects of cancer through integrin-mediated mechanotransduction[27]. This prompted us to investigate whether stiffness influences ITGB1 phosphorylation. We seeded Illusia-expressing cells on hydrogels of different stiffnesses, closer to that found in mammalian tissues and 3D cultures[28,29]. By doing so, we observed increased ITGB1 phosphorylation

at lower stiffnesses (Fig. 6a), occurring in parallel with a reduced cell area (Fig. 6b). This suggests that adhesion in softer environments, such as 3D and mouse models, may have a greater reliance on integrin phosphorylation than typical two-dimensional (2D) cultures on glass. This is in line with previous work demonstrating a continued reliance on Src and FAK for cancer cell survival in conditions with reduced adhesion[30,31]. Indeed, it has been previously suggested that integrin phosphorylation may be increased on soft substrates in response to the lower number of mature adhesions, and our data is in line with that hypothesis[32].

Next, we assessed the relative contribution of the kinase–phosphatase balance in this process by determining the effect of Sara (Src inhibitor) and SHP099 (Shp2 inhibitor) on proliferation of a panel of breast cancer cell lines, including the MM231 cells and another triple-negative breast cancer line, MDA-MB-468 (MM468) (Extended Data Fig. 6a). Here, we observed growth reduction only at high inhibitor concentrations (>5 μM). To unravel the role of Shp2 and Src on invasion, we kept inhibitor concentrations below this threshold and observed a dramatic effect on MM231 invasion in fibroblast-contracted 3D collagen I matrices (Fig. 6c, SHP099, and Extended Data Fig. 6b, Sara). We then explored this effect on invasion further using a basement membrane invasion assay that incorporates decellularized mouse mesentery to recapitulate basement membrane organization and characteristics more faithfully than in vitro polymerized gels, such as Matrigel[33] (Fig. 6d). In this system, both the Src and Shp2 inhibitors provoked a significant reduction in MM231 and MM468 cell invasion (Fig. 6e).

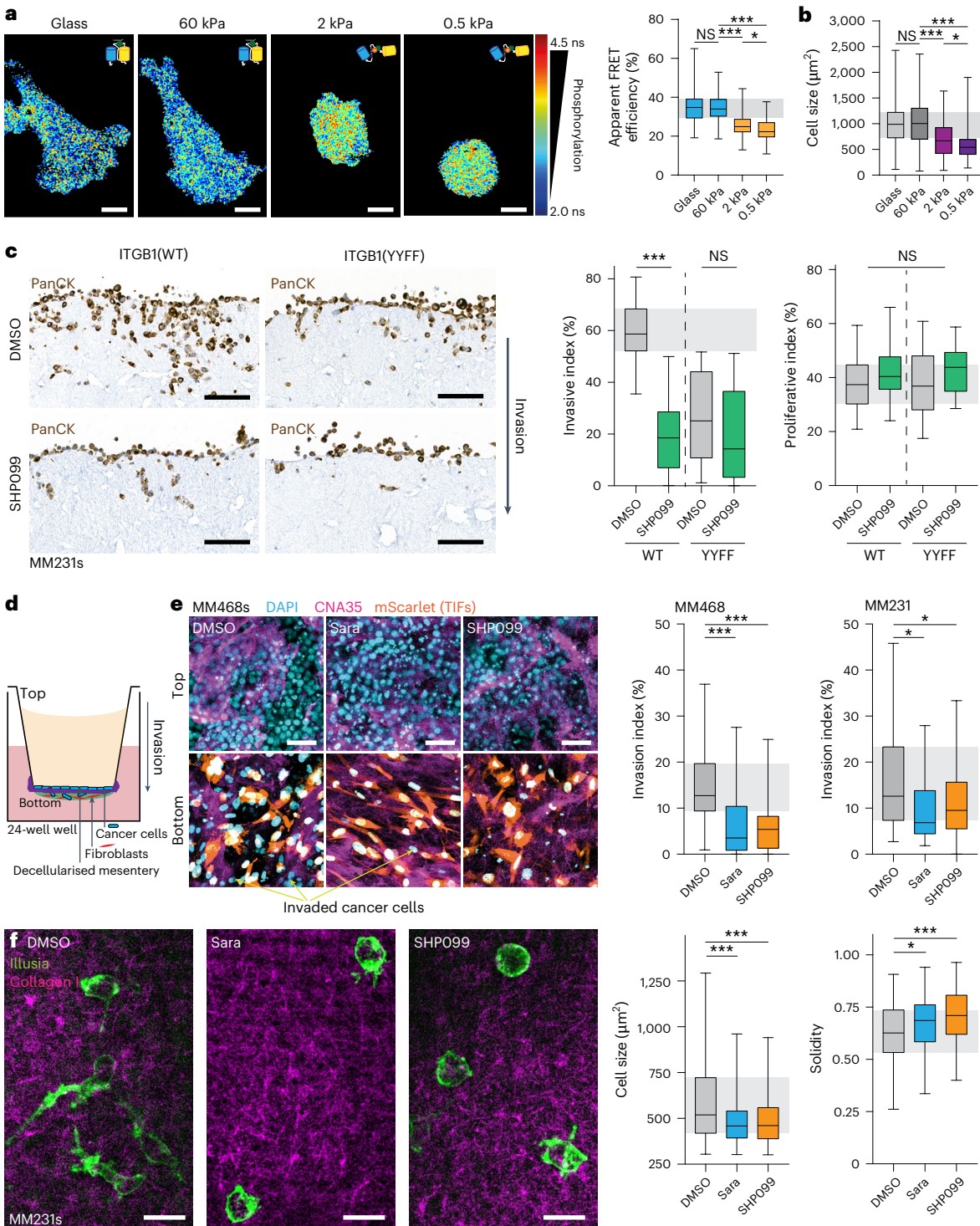

Interestingly, inhibition of either phosphorylation or dephosphorylation resulted in an anti-invasive effect, suggesting that phosphorylation dynamics could be essential for cancer cell movement through a 3D environment. While the number and size of IACs was unchanged between MM231 ITGB1(WT) or ITBG1(YYFF) cells (Extended Data Fig. 1g–i), live imaging revealed that IAC dynamics were significantly reduced in MM231s after SHP099 treatment (Supplementary Video 1 and quantified in Extended Data Fig. 6c). Similarly, MM231 cells embedded in a 3D matrix and treated with Src and Shp2 inhibitors showed a rounded morphology and reduced area (Fig. 6f), suggesting that their ability to adhere and move through the matrix was disrupted by loss of integrin phosphorylation dynamics. Furthermore, live tracking of

Illusia using FRET by sensitized emission (SE-FRET) demonstrated rapid ITGB1 phosphorylation dynamics in cells invading in 3D collagen matrices (Extended Data Fig. 6d and Supplementary Video 2). This dynamic phosphorylation was lost following either SHP099 or Sara treatments where cells exhibited a rounded morphology. In 2D, SE-FRET confirmed a decrease in FRET signal in the SHP099 treatment condition and an increase with Sara, confirming the validity of the SE-FRET approach (Extended Data Fig. 6e). Altogether, we see that disrupting integrin phosphorylation dynamics through mutation of the receptor's NPxY sites or small-molecule inhibition of the identified integrin kinase or phosphatase leads to a reduction in breast cancer invasion in 2D and 3D settings.

**Fig. 6 | Src or Shp2 inhibition of ITGB1 phosphorylation dynamics results in equivalent phenotypes. a,b**, Representative FLIM images (**a**, left) and apparent FRET efficiencies (**a**, right) and cell area (**b**) of MM231 cells stably expressing Illusia and seeded on either glass or hydrogels (60, 2 or 0.5 kPa) (for **a**, n = 110 (glass), 117 (60 kPa), 116 (2 kPa) and 113 (0.5 kPa) cells pooled from four biological replicates; for **b**, n = 121 (glass), 137 (60 kPa), 157 (2 kPa) and 165 (0.5 kPa) cells pooled from four biological replicates; one-way analysis of variance (ANOVA) with a Tukey correction for multiple comparisons). Scale bars, 10 μm. **c**, Representative images (left) of MM231 ITGB1(WT or YYFF) cells invading into 3D fibroblast-contracted collagen I treated with DMSO or SHP099 (100 nM). Pan-cytokeratin (PanCK) staining was used to mark cancer cells and exclude fibroblasts from the analysis. Quantification of invasion beyond 100 μm (right), normalized to the total number of cells/region, or proliferation, normalizing the number of Ki67-positive nuclei to the total number of cells/region (n = 24 regions per cell line pooled from four biological replicates; one-way ANOVA with a Tukey correction for multiple comparisons). Scale bars, 100 μm. **d**, A scheme for the basement membrane invasion assay. **e**, Left: representative images of MM231 and MM468 cells invading into the basement membrane matrix for

4 or 5 days, respectively, in the presence of SHP099 (100 nM) or Sara (1 μM) (collagen I is labelled with HaloTag-CNA35 (magenta), fibroblasts by mScarlet expression (red) and all nuclei stained with DAPI (cyan); the cancer cells are apparent by nuclei staining alone (cyan-positive, mScarlet-negative cells)). Quantification of basement membrane invasion (MM231 cells, n = 31 (from 11 basement membranes; DMSO), 24 (from 9 basement membranes; Sara) and 23 (from 8 basement membranes; SHP099) regions pooled from three biological replicates (right); MM468 cells, n = 31 (from 12 basement membranes; DMSO), 32 (from 12 basement membranes; Sara) and 27 (from 9 basement membranes; SHP099) regions pooled from four biological replicates; one-way ANOVA with a Dunnett's correction for multiple comparisons). Scale bars, 50 μm. **f**, Illusia-expressing MM231s (green) embedded in collagen I (magenta) and treated with SHP099 (100 nM) or Sara (1 μM) for 24 h (cell size, n = 119 (DMSO), 93 (Sara) and 85 (SHP099); solidity, n = 122 (DMSO), 91 (Sara) and 94 (SHP099) cells from four biological replicates; one-way ANOVA with a Dunnett's correction). The boxplots represent the median and IQR. The whiskers extend to the minimum and maximum values. The grey areas highlight the IQR of the control conditions. NS, not significant. *P < 0.05, ***P < 0.001.

## Cofilin is recruited to the phosphorylated ITGB1/Dok1 complex

Several studies have characterized the adhesion complex recruited upon talin binding to the cytoplasmic domain of ITGB1 (refs. 1,12). By contrast, little is known about the composition of the phosphorylated complex. To identify the proteins recruited upon Dok1 binding to phosphorylated ITGB1, we performed bimolecular complementation affinity purification (BiCAP) (Fig. 7a) and mass spectrometry on the immunoprecipitated complex[34] (Fig. 7b and Supplementary Table 2). This identified 25 proteins (including ITGB1 and Dok1) that were significantly enriched in the BiCAP condition. Three of these, Cofilin, VPS35 and annexin A6, were then further validated by western blot using green fluorescent protein (GFP)-trap IP (Fig. 7c). The formation of these complexes within intact MM231 cells was further supported by bimolecular fluorescence complementation (BiFC)–FLIM–FRET (Fig. 7a), performed between the ITGB1/Dok1 complex and red fluorescent protein (RFP)-tagged Cofilin (Fig. 7d), VPS35 or annexin A6 (Extended Data Fig. 7a). Interestingly, the interaction with Cofilin occurred independently of Cofilin phosphorylation state, suggesting that both active and inactive Cofilin could be recruited to the ITGB1/Dok1 complex (Fig. 7d).

Cofilin is an essential actin regulator, implicated in invasion and the generation of matrix-degrading invadopodia[35,36]. Arg and Src are also linked with invadopodia formation through the phosphorylation of cortactin to release Cofilin[37,38]. Therefore, we next assessed the role of Src and the ITGB1/Dok1 complex in invadopodia formation (Extended Data Fig. 7b). Overexpression of constitutively active Src kinase induced invadopodia formation, as expected[39], but this was significantly reduced in cells with non-phosphorylatable ITGB1(YYFF) (Extended Data Fig. 7c). Furthermore, siRNA silencing of *DOK1* resulted in a dramatic loss of Src-induced invadopodia in both ITGB1 WT and YYFF backgrounds (Fig. 7e), together supporting the link between invadopodia formation and the phosphorylated ITGB1 complex. Notably, silencing of Dok1 resulted in a more pronounced phenotype than non-phosphorylatable ITGB1, suggesting that the role of ITGB1 can be compensated by other adhesion receptors but that the scaffolding functions of Dok1 are less dispensable for invadopodia formation. This is further supported by the established phosphorylation-specific binding of Dok1 to other integrin β isoforms[11,40].

## Dynamic ITGB1 phosphorylation supports invadopodia formation

Invadopodia comprise a well-established set of proteins, with the scaffold proteins cortactin (CTTN) and TKS5 among the most explored. Since very few invadopodia components were identified in the BiCAP, we took a more targeted approach using intermolecular FRET to assess the interaction of the phosphorylated ITGB1 complex with these

scaffold proteins. We observed that Dok1 and Cofilin interact with both CTTN and TKS5 (Fig. 8a–d). Furthermore, Dok1 silencing strongly downregulated Cofilin and, to a lesser extent, TKS5 and CTTN protein levels (Extended Data Fig. 8a and quantified in Extended Data Fig. 8b). Dok1 was clearly reduced at the RNA level, consistent with efficient silencing by the siRNAs, while the other invadopodia components showed variable responses to Dok1 silencing (Extended Data Fig. 8c), suggesting that the observed coregulation occurs primarily at the protein, not mRNA level and could involve altered stability of invadopodia components in the absence of Dok1. Taken together, these data demonstrate a novel requirement for the phosphorylated integrin complex and the ensuing interactions with Cofilin, TKS5 and CTTN for invadopodia formation and function. These data are concordant with the requirement for ITGB1 phosphorylation for local invasion away from the primary tumour (Fig. 1).

## Dynamic ITGB1 phosphorylation supports metastasis

To investigate the role of integrin phosphorylation dynamics during later stages of the metastatic cascade, we utilized an animal model of extravasation and metastatic colonization of the lung. To track this process, we engineered MM231 ITGB1(WT or YYFF) cells to express a luciferase/EGFP construct (Extended Data Fig. 8d), observing equivalent luciferase signal in the lung immediately after injection of the cells into the lateral tail vein (Extended Data Fig. 8e and quantified in Extended Data Fig. 8f). The mice injected with MM231 ITGB1(WT) cells showed higher outgrowth of lung metastases compared with the MM231 ITGB1(YYFF) cells (P = 0.0939) (Fig. 8e and quantified in Fig. 8f) along with a greater metastatic area (P = 0.1051) (Extended Data Fig. 8g). Furthermore, treatment of mice with the Shp2 inhibitor, SHP099, reduced the number of metastatic nodules and the overall tumour mass in the lung (P = 0.3622) (Fig. 8g and quantified in Fig. 8h). Interestingly, more granular assessment of the ratio of micro- to macrometastasis demonstrated further the requirement for integrin phosphorylation, observing a slight increase in the ratio towards micrometastasis in ITGB1(YYFF) cells, when compared to the WT controls (Extended Data Fig. 8h). Given the similar sensitivity of both MM231 ITGB1(WT) and ITGB1(YYFF) cells to flow-induced cell death in suspension, these differences are unlikely to reflect differences in cell survival in the circulation (Extended Data Fig. 8i,j). While these data did not reach statistical significance, they show a trend in line with a requirement for integrin phosphorylation dynamics during effective cancer dissemination (Fig. 8i). Together, we propose a mechanism where finely controlled integrin phosphorylation and dephosphorylation regulate cell invasion and cancer dissemination through invadopodia formation and play a key role at distinct steps of the metastatic cascade.

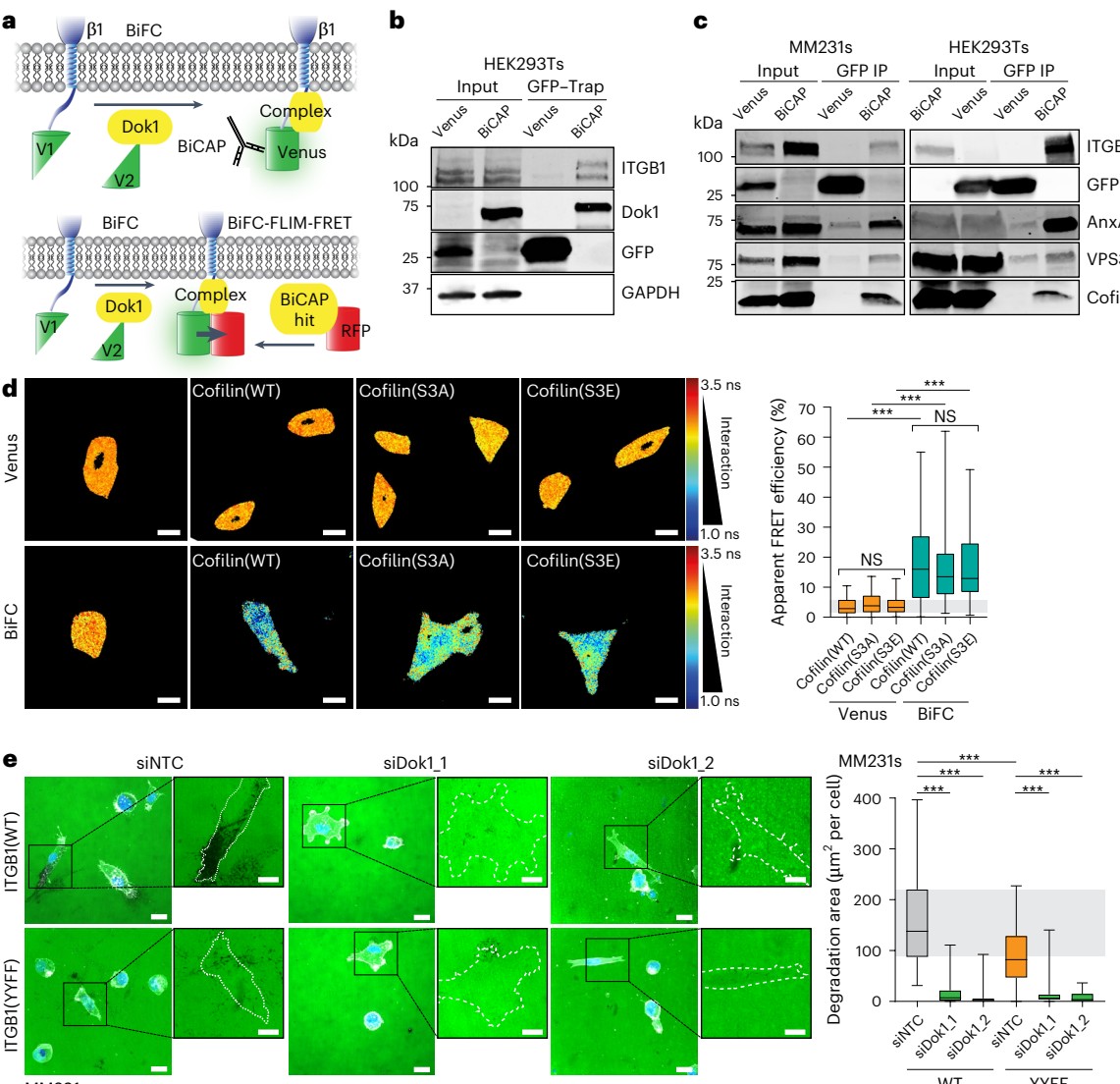

**Fig. 7 | Phosphorylation-sensitive recruitment of Dok1 supports invadopodia formation. a**, A schematic of the interaction between ITGB1-V1 and a V2-tagged adaptor or regulator, highlighting the resulting V1/V2 (Venus) protein complex as the restored epitope for BiCAP (top) or donor for BiFC–FLIM–FRET (bottom). The Venus tag alone was used as a control for the BiCAP and BiFC–FLIM–FRET experiments. **b**, A representative BiCAP immunoblot after HEK293T cell transfection with either Venus or ITGB1-V1/Dok1-V2 ($n$ = 3 biological replicates). **c**, Representative BiCAP immunoblots from MM231 and HEK293T cells where annexin A6 (AnxA6), VPS35 and Cofilin coimmunoprecipitate with the Dok1/ITGB1 complex ($n$ = 3 biological replicates). **d**, Representative BiFC–FLIM–FRET images (left) and quantification of apparent FRET efficiency (right) from MM231 cells transfected with RFP-tagged Cofilin mutants WT, S3A and S3E (Venus, $n$ = 73 (WT), 72 (S3A) and 64 (S3E); BiFC, $n$ = 92 (WT), 64 (S3A) and 64 (S3E) cells from

three biological replicates; one-way analysis of variance (ANOVA) with a Tukey correction for multiple comparisons). Scale bars, 20 μm. **e**, Representative images (left) and quantification of gelatin degradation (right) by MM231 ITGB1(WT or YYFF) cells with Dox-inducible Src(E378G) expression. The MM231 cells were transfected with siRNAs against Dok1 (siDok1_1 and siDok1_2) or a NTC siRNA and treated with Dox for 24 h before being seeded on fluorescent gelatin (green) for 6 h (actin labelled with SiR-actin (white), nuclei with DAPI (blue)) (ITGB1(WT), $n$ = 36 (NTC), 38 (siDok1_1) and 36 (siDok1_2); ITGB1(YYFF), $n$ = 38 (NTC), 34 (siDok1_1) and 37 (siDok1_2) fields of view pooled from three biological replicates; one-way ANOVA with a Šidák correction for multiple comparisons). Scale bars, 20 μm; insets, 10 μm. The boxplots represent the median and IQR. The whiskers extend to the minimum and maximum values. The grey areas highlight the IQR of the control conditions. NS, not significant. ***$P$ < 0.001.

## Discussion

Here, we show that dynamic ITGB1 phosphorylation facilitates the recruitment of a proinvasive Dok1 complex to support invadopodia formation, cancer invasion and metastatic dissemination. Through the development of Illusia, a FRET reporter for ITGB1 phosphorylation, we demonstrate spatiotemporal regulation of this post-translational modification in invading cells. We identify and validate two integrin PTPs, Shp2 and PTP-PEST and, conversely, demonstrate increased integrin phosphorylation by Src and Arg kinases. Strikingly, inhibition of either Src or Shp2 dramatically inhibits cell migration and invasion, demonstrating that dynamic integrin phosphorylation is important for

efficient cancer invasion and metastatic colonization. Furthermore, Dok1 binding to phosphorylated ITGB1 facilitates recruitment of the actin-severing protein Cofilin, along with the invadopodia scaffolds TKS5 and CTTN. Importantly, this invadopodia complex is lost in cells with non-phosphorylatable ITGB1 YYFF, which demonstrate compromised invadopodia formation and invasion.

On the whole-cell level, integrins impact cell migration by regulating front-rear polarity and signalling[41,42]. In our study, live-cell imaging appears to show fluctuations in phosphorylation (Illusia reporter) in invading cells, with flickering and regional differences at the leading and trailing edges (Supplementary Video 2). This implies that integrin

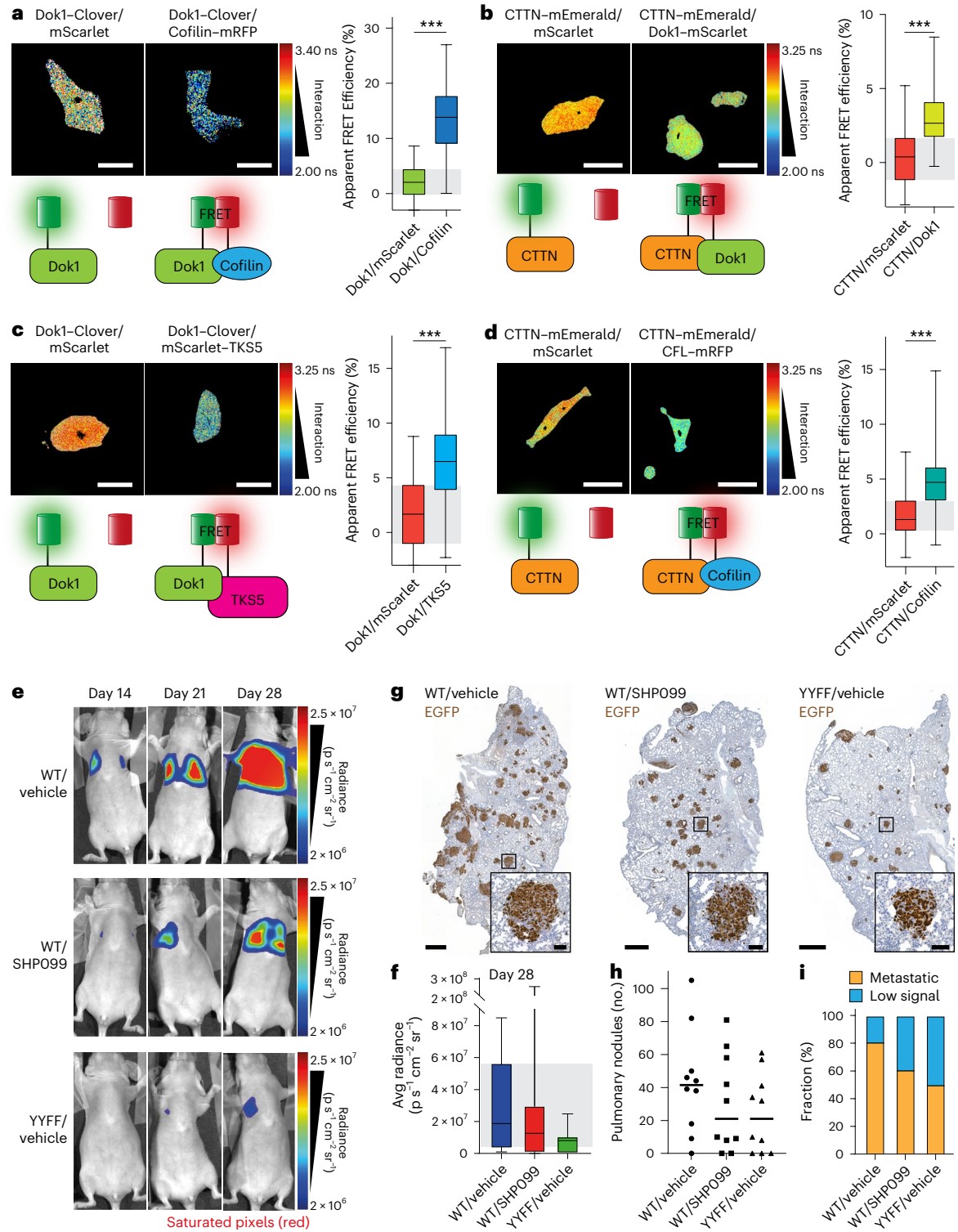

phosphorylation may be polarized in cells, probably coordinated by rapid phosphorylation cycles of individual integrin molecules. The constant switching between talin-bound non-phosphorylated ITGB1 and the phosphorylated integrin/Dok1 multiprotein complex may facilitate tuning of adhesion receptor functions through exchange of proximal interactors and thus contribute to front-rear polarity and cell migration. Future studies are needed to interrogate how this is governed across scales (from single molecules to cells) and fine-tuned by tyrosine kinase/phosphatase networks. One insight comes from a study demonstrating Shp2 regulation of focal adhesion kinase activity at the cell periphery and nascent adhesion formation consistent with

lamellipodia spreading[43]. Similarly, the Shp2 ability to promote IAC maturation through ROCK2 activation indicates that phosphorylation dynamics are important to modulate the activation state of different IAC components to ensure that dynamic processes are not stalled by over/underactivation[44] (further discussed in ref. 45). Notably, a screening study assessing the binding of several PTB-domain-containing adaptor proteins to an NxxY-peptide array identified three potential phosphorylation-dependent binding partners, Appl, Dok2 and Frs2 (ref. 46). Interestingly, loss of Frs2 has recently been shown to sensitize cells to Shp2 inhibition, suggesting a further link between cancer progression and additional phosphorylation-sensitive integrin

**Fig. 8 | The Dok1/ITGB1 complex recruits Cofilin and other invadopodia components to adhesion sites to mediate efficient cancer dissemination.**
**a–d**, Representative images (left) and quantification of apparent FRET efficiency (right) for intermolecular FLIM–FRET of the following tagged protein pairs, Dok1–Clover/Cofilin–mRFP (**a**), CTTN–mEmerald/Dok1–mScarlet (**b**), Dok1–Clover/mScarlet–TKS5 (**c**) and CTTN–mEmerald/Cofilin–mRFP (**d**). FRET between mScarlet and the donor-tagged protein was used as a negative control for all pairs (for **a**, n = 85 (Dok1/mScarlet) and 95 (Dok1/Cofilin) cells pooled from five biological replicates; for **b**, n = 65 (CTTN/mScarlet) and 68 (CTTN/Dok1) cells pooled from three biological replicates; for **c**, n = 62 cells for each condition pooled from three biological replicates; for **d**, n = 65 (CTTN/mScarlet) and 70 (CTTN/Cofilin) cells pooled from three biological replicates; unpaired two-tailed Student's t-test with a Welch's correction). Scale bars, 20 μm. **e**, Representative images of mice with MM231 ITGB1 (WT or YYFF) cells stably expressing the luciferase/EGFP construct. Oral gavage of Vehicle or SHP099

(100 mg kg⁻¹) proceeded for 5 days from the day of injection. **f**, A box and whisker plot highlighting the endpoint metastatic burden as an average (Avg) radiance value from the luciferase signal of the MM231 cells in **e** (n = 9 mice tracked per group). **g**, Representative lung sections stained for EGFP-positive MM231 cells. Scale bars, 2 mm; insets: 200 μm. **h**, Quantification of pulmonary nodule number (that is, clusters of greater than ten cells) in lungs from EGFP-positive MM231 cells (n = 10 mice per group). **i**, Quantitative real-time PCR of the RNA samples collected from the MM231 ITGB1(WT or YYFF) cells stably expressing the luciferase/EGFP construct. The mice were designated as either 'metastatic' or 'low signal' after setting a threshold for 'metastatic' as having an expression fold change >1 compared with the mean of the WT/vehicle control with human GAPDH normalized to mouse/human GAPDH (n = 10 mice/group). The boxplots represent the median and IQR. The whiskers extend to min and max values. The grey areas highlight the IQR of the control conditions. ***P < 0.001.

complexes[47]. Such studies highlight the diversity of adhesion complexes and the role of post-translational modifications in modulating their functions within the cell.

There are no obvious cell migration defects reported for mice carrying the ITGB1 YYFF mutant[5,9]. However, in organisms with fewer integrin isoforms, YYFF mutation of the *ITGB1* orthologs leads to defects in distal tip cell migration and ovulation (*Caenorhabditis elegans*) and disruption of the myotendinous junctions (*Drosophila melanogaster*)[48,49]. It is important to note that the two conserved cytoplasmic NxxY motifs are present in multiple integrin β-subunits and Dok1 can interact with them[50]. The significant decrease in invadopodia observed after Dok1 KD (Fig. 7e) suggests that Dok1 could be a core component for invasion mediated by multiple integrins and implies a wider impact of our findings. Furthermore, the finding that ITGB1 phosphorylation is increased on softer substrates (Fig. 6a,b) was initially surprising given previous findings that global tyrosine phosphorylation is reduced in softer environments[51]. Yet, primary human fibroblasts have been shown to increase their invadopodia formation on softer hydrogels[52], and the Src activation state is maintained in both soft and stiff conditions in both cancer and fibroblast cell lines[52,53]. Taken together, these data affirm the significance of ITGB1 phosphorylation dynamics for cell invasiveness and highlight the need for continued investigation into the role of adhesion signalling in different stiffness environments.

While several studies have explored the potential of ablating ITGB1 with genetic approaches, demonstrating reduced breast tumour growth and metastasis in xenograft and genetically engineered mouse models[14,54,55], targeting integrin heterodimers, has so far failed to show efficacy in a cancer setting[13,56]. Given the central role of integrins in cancer progression, the identification of new regulatory vulnerabilities, such as the one described herein, provides new avenues for therapeutic development. Src inhibition has already been demonstrated as an effective antimetastatic therapy in MM231 cells[57,58], and recent work has found similar efficacy for SHP099 in reducing lung colonization by mouse 4T1 breast cancer cells[59]. This is in line with reduced extravasation when invadopodia components, namely CTTN, LPP and MT1-MMP, are removed from breast cancer cells[37,60,61]. The opposite is also true for overexpression of the invadopodia scaffold TKS5, which increases lung cancer[62] and melanoma[63] metastasis. Here, we describe integrin phosphorylation as an apparent 'switch on signal' for invadopodia formation, leading to Dok1 recruitment to the integrin cytoplasmic tail, followed by Dok1-mediated complex formation with CTTN, Cofilin and TKS5. These data highlight the potential of inhibiting integrin phosphorylation regulators to reduce invadopodia formation and ultimately reduce metastatic dissemination.

Another interesting, unexplored finding is the range of PTPs identified in our FRET screen as having the potential to increase ITGB1 phosphorylation. These PTPs probably regulate integrin phosphorylation indirectly by converging on kinase activity and signalling pathways, providing additional complexity to integrin phosphorylation

dynamics. This level of control emphasizes the central role of integrin adhesion receptors and their dynamic phosphorylation in regulating cell behaviour.

## Online content

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

## Methods

All animal experiments were performed in accordance with The Finnish Act on Animal Experimentation (Animal licence number ESAVI/12558/2021). All experiments respected the maximum tumour diameter (15 mm) permitted by the authorization bodies.

### Animal experiments

Subcutaneous xenografts were generated in the flank of 7–8-week-old female athymic nude mice (Foxn1[nu], Envigo) by injecting $3 \times 10^6$ MM231 ITGB1(WT or YYFF) cells in phosphate-buffered saline (PBS). The tumours were then tracked by palpation with callipers until the tumour volume ($\frac{\text{length} \times \text{width}^2}{2}$) was >300 mm³, at which point the mice were sacrificed and tumours were collected. Quantification of local invasion and Ki67-positive nuclei was performed in QuPath[64], as described previously[65,66], on xenografts with a smaller cross-sectional area than 300,000 μm² that showed local invasion.

Colonization of the lung was assessed through lateral tail vein injection of 100 μl PBS with $7.5 \times 10^5$ MM231 ITGB1(WT or YYFF) cells expressing luciferase/EGFP into 7–8-week-old female athymic nude mice (Foxn1[nu], Envigo). The cells were treated with dimethyl sulfoxide (DMSO) or SHP099 (100 nM) 24 h before injection. Oral gavage of the vehicle (0.5% methylcellulose, 0.1% Tween80 in sterile water) or SHP099 (100 mg kg⁻¹) was then performed daily for 5 days, starting from day 0, 2–3 h before tail vein injections. To reduce oesophageal irritation from repeated gavage, 24% sucrose in sterile water was applied to the needles for each gavage[67]. The metastatic growth was tracked using D-luciferin (150 mg kg⁻¹; PerkinElmer, 122799) through intraperitoneal injection and imaging on an IVIS spectrum (PerkinElmer). Lungs were collected 28 days later, taking the large lobe for paraffin embedding after formalin fixation and sending the remaining lobes for RNA isolation.

The mice were housed in standard conditions (12-h light–dark cycle; ambient temperature, 21 °C; 50% ± 8% humidity) with food and water available ad libitum.

### Anoikis assays

To measure the resistance of cells to anoikis and shear forces, a custom-made flow system was developed. This was composed of a peristaltic pump for flow induction (Ismatec, Reglo digital, MS-4/12) and microfluidic tubing with a closed-loop circuit, composed of 50 cm silicon tubing (Ibidi, 10840) and a three-stop connector compatible with the Ismatec pump (Tygon 0.38 ID; Ismatec, SCE0398) linked together with 23G needles. The cells were perfused into this system after resuspending in 1 ml of complete media and treating with SHP099, DMSO or a mixture of cell-death-inducing compounds as a positive control (doxorubicin (10 μg ml⁻¹; Selleckchem, E2516), $VO_4^{3-}$ (50 μM), gemcitabine (10 μM; Selleckchem, S1149)), as indicated. For the 'no flow' controls, the cells were seeded in ultralow attachment plates (Corning, 3471) for 24 h. Whereas the 'flow' samples were taken from these ultralow attachment plates after 22 h and aspirated into the microfluidic system before closing the circuit. A capillary-like flow of 400 μm s⁻¹ was used as pump input for 2 h, as described previously[68]. The cells from all conditions were then collected, and annexinV staining was performed following the manufacturer's instructions (Abcam, ab219919). Flow cytometry data acquisition was performed on a BD LSRFortessa Cell Analyzer (BD Biosciences), using FlowJo software (version 10.8.2, BD Biosciences) for the analysis.

### Basement membrane invasion assays

The mesenteric basement membrane was surgically extracted from mouse intestines, then adhered to the bottom of a plastic transwell insert (Greiner, Thincerts, 8 μm pore size, 662638) after first removing the filter with a scalpel, as described previously[33]. The mesentery was then decellularized using $NH_4OH$ (1 M) in PBS with 1% Triton X-100 for 60 min. These were then washed 4× with PBS, treated with DNAse

I (10 μg ml⁻¹, Roche) in PBS for 30 min at 37 °C, washed 1× with PBS and left overnight at 4 °C in PBS containing calcium and magnesium. Next, mesenteries were seeded with $3 \times 10^5$ mScarlet-overexpressing TIFs/membrane in 50 μl of a 1:4 collagen:FBS mix, neutralizing the collagen I as described previously[16,65], allowing this to set at 37 °C before adding complete growth medium and incubating overnight at 37 °C with 5% $CO_2$. MM468 or MM231 cells (final concentration of $1 \times 10^6$ cells per membrane) overexpressing EGFP were then seeded in the top of these mesenteric membranes and a chemoattractive gradient was established with complete media in the bottom of the Transwell and serum free media in the top. MM231 or MM468 cells were then treated with Sara (SelleckChem, S1006) and SHP099 (SelleckChem, S8278) for 4 or 5 days, respectively, refreshing the media every 2 days. At the end of the invasion period, the mesenteries were treated with CNA35-HaloTag recombinant protein (270 μg ml⁻¹), as well as HaloTag far-red ligand (10 nM; Promega, JaneliaFluor 646, GA1120) before fixation and staining with 4′,6-diamidino-2-phenylindole (DAPI) (dihydrochloride; 1 μg ml⁻¹, Life Technologies, D1306) and then imaged on a Zeiss LSM880 with Airyscan and the 40× Zeiss LD LCI Plan-Apochromat objective.

### BiCAP mass spectrometry

BiCAP was performed as described previously[34], using MM231 or HEK293T cells transfected with either pDEST-ITGB1-V1/pDEST-Dok1-V2 or control pEF.DEST51-mVenus using Lipofectamine 3000 (Thermo Fisher) in OptiMEM (Gibco, 31985070), as per the manufacturer's protocol. After an overnight incubation, the cells were lysed in IP lysis buffer (40 mM HEPES buffer, 75 mM NaCl, 2 mM EDTA, 1% NP-40, along with protease (cOmplete Mini, EDTA-free, Roche) and phosphatase (PhosSTOP, Roche) inhibitor cocktails) and subjected to IP using GFP-trap beads (30 μl, Chromotek, gfa), at 4 °C with gentle agitation for 1 h. The samples were then spun down at 300g for 5 min and the beads washed three times with PBS and once with 50 mM Tris–HCl, pH 8.0, 150 mM NaCl. The on-bead digestion and liquid chromatography–tandem mass spectrometry were performed as described previously[65], before the assignment of peptides and label-free quantification of abundance ratios (normalized to total peptide amount) in Proteome Discoverer 2.5 (Thermo Fisher) using intensity values from the precursor ions.

### Cell line models

TIFs (a kind gift from J. C. Norman, Beatson Institute)[69], HEK293FT (Thermo Fisher, R70007), HEK293T (ATCC, CRL-3216), MM231 (ATCC, HTB-26), MM468 (ATCC, HTB-132), HCC1937 (ATCC, CRL-2336) and MDA-MB-361 (ATCC, HTB-27) cells were all cultured in Dulbecco's modified Eagle medium (Sigma, D2429) supplemented with 10% FBS (20% FBS for the MDA-MB-361 cells) and L-glutamine (100 mM). MCF10A (ATCC, CRL-10317) cells were cultured in Dulbecco's modified Eagle medium/F12 (Invitrogen, 11330-032) supplemented with 5% horse serum, EGF (20 ng ml⁻¹; Peprotech, AF-100-15), hydrocortisone (0.5 mg ml⁻¹; Sigma, H0888), cholera toxin (100 ng ml⁻¹; Sigma, C8052) and insulin (10 μg ml⁻¹; Sigma, I9278). All cell lines were regularly tested for mycoplasma and were found to be negative. The MM231s were authenticated by STR profiling using the services of the Leibniz Institute DSMZ. BT-20 (ATCC, HTB-19) cells were cultured in minimum essential medium (Thermo Fisher, 31095052) supplemented with 10% FBS and L-glutamine (100 mM).

### FLIM–FRET microscopy

Frequency-domain FLIM–FRET was performed using a LIFA fast frequency-domain FLIM system (Lambert Instruments) attached to an inverted microscope (Zeiss AXIO Observer.D1) with sinusoidally modulated (40 MHz) epi-illumination (1 W for 405 nm or 3 W for 470 nm) from a temperature-stabilized multi-light-emitting diode (LED) system (Lambert Instruments) and a ×63/1.15 objective (Zeiss, Objective LD C-Apochromat ×63/1.15 W Corr M27). Atto425 (1 μM; Sigma, 56759) or

fluorescein (20 µM; Sigma, F7505) in PBS/0.1 M Tris, pH 7.5 were used as lifetime reference standards. An appropriate filter set for mTurquoise2 (FT 455, no excitation filter//475/20) or Clover/EGFP (FT 480, 450/50//510/50) was used to measure the phase and modulation fluorescence lifetimes per pixel from images of cells acquired at 12 phase settings, using the manufacturer's software. The apparent FRET efficiency ($E_{app}$) was calculated using the measured lifetimes per cell of each donor–acceptor pair ($\tau_{DA}$) and the average lifetime of the donor-only ($\tau_D$) samples,

$$E_{app} = (1 - \tau_{DA}/\tau_D) \times 100.$$

### Flow cytometry
MM231 ITGB1(WT or YYFF) cell lines were trypsinized, fixed in 2% PFA for 10 min at room temperature and washed with PBS before being stained with primary antibodies to assess surface integrin levels (1:100 dilution in Tyrode's buffer (ITGB1 (clone P5D2, in-house production from DSHB hybridoma), inactive ITGB1 (clone mAb13, in-house production from DSHB hybridoma) and β3 (MCA728, AbD Serotech))) for 1 h at 4 °C with gentle agitation. The samples were then washed 2× with PBS and incubated with secondary antibodies (ALEXA-488 conjugated anti-mouse/anti-rat Invitrogen; 1:300 dilution in Tyrode's buffer) for a 1 h at 4 °C with gentle agitation. The cells were then washed again with PBS before being resuspended in 200 µl of PBS and loaded into a 96-well plate. Cytometry was then performed on an LSRFortessa Cell Analyzer using the High Throughput Sampler (BD Biosciences). Up to 10,000 single cell events were collected per condition. Gating and statistical analysis of the cell population were performed in FlowJo (version 10.8.2, BD Biosciences).

### Gene silencing
Gene silencing was performed by transfecting 10 nM siRNA using RNAiMAX, as per the manufacturer's instructions. The non-target control (NTC) (AllStars negative control), Dok1_1 (Hs_Dok1_2, SI00372799, AAGGATCCCAATTCGGGTAA) and Dok1_2 (Hs_Dok1_3, SI00372792, CCGCCTGGACTGCAAAGTGAT) siRNAs were all purchased from QIAGEN. For the siRNA screen, two negative controls (Silencer Select negative control 1 (s813; 4390843) and Silencer Select negative control 2 (s814; 4390846)) and the Silencer Select Human Phosphatase library (4397919) were used (Thermo Fisher Scientific, Ambion). GFP silencing was performed with a Silencer Select anti-GFP siRNA (Thermo Fisher Scientific, Ambion, s229077, 4399665). The RNAi library was resuspended and reformatted into the plates with a Biomek NXP pipetting robot (Beckman Coulter) and ATS100 acoustic nanodispenser (EDC biosystems) under laminar flow (Kojair).

### Generation of stable cell lines
For stable expression of the mRuby2-tagged ITGB1 WT and YYFF and the Illusia and Illusia(YYFF) intramolecular FRET biosensors, MM231 cells were cotransfected with the respective pPB.DEST vectors and pCMV-hyPBase (Supplementary Table 3), using Lipofectamine 3000, as per the manufacturer's protocol (Thermo Fisher Scientific). The stable fluorescent cells were then isolated by FACS within a narrow fluorescence range.

The lentiviral particles for the generation of stable cell lines were packaged in HEK293FT packaging cells by cotransfecting the third-generation lentiviral packaging system, pMDLg/pRRE, pRSV-Rev and pMD2.G, along with one of several transfer plasmids (that is, pLKO.1-shBeta1, pLV430g-ofl_T2A_EGFP, pLenti6.3/TO/V5-DEST-Paxillin-EGFP, pLenti6.3/TO/V5-DEST-EGFP, pLenti6.3/TO/V5-DEST-mScarlet, pINDUCER20-Src(WT), pINDUCER20-Src(E378G), pINDUCER20-Src(K295R), pINDUCER20-Src(Y527F), pINDUCER20-ABL2(WT), pINDUCER20-ABL2(K281M), pINDUCER20-PTPN12 or pINDUCER20-PTPN12(D199A,C231S); Supplementary Table 5) using Lipofectamine 3000 (Thermo Fisher)

in OptiMEM (Gibco, 31985070), as per the manufacturer's protocol. A total of 24 h after transfection, the medium was replaced with complete growth medium for a further 24 h, at which point the medium was collected and filtered through a 0.45-µm syringe filter. MM231s or TIFs were then transduced with the lentiviral particles for 48 h, in the presence of Polybrene (8 µg ml⁻¹; Sigma, TR-1003-G), before washing and selecting stable positive cells using puromycin (1 µg ml⁻¹). Where fluorescent constructs were used, stable cells were then isolated by FACS within a narrow fluorescence range.

### IP
The cells were collected in 300 µl IP lysis buffer (40 mM HEPES, 75 mM NaCl, 2 mM EDTA, 1% NP-40, along with protease (cOmpleteTM Mini, EDTA-free, Roche) and phosphatase (PhosSTOP, Roche) inhibitor cocktails) using a plastic cell lifter. They were then incubated in the IP lysis buffer for 1 h with vigorous rotation at 4 °C. The lysates were then centrifuged at 20,000*g* at 4 °C for 10 min, and the cleared lysate was transferred to a new Eppendorf tube; 30 µl was added to 7 µl of 8× sample buffer (SB; 0.66 M Tris-HCl, pH 6.8, 33% glycerol, 398 µM bromphenol blue, 266 mM dithiothreitol (DTT; Sigma, D0632), 740 mM sodium dodecyl sulfate) to use as an input control. The remainder of the cleared lysate was incubated for 1 h at 4 °C with anti-pY affinity beads (30 µl per sample, APY03-Beads, Cytoskeleton) or GFP-trap beads (30 µl, Chromotek, gfa), as appropriate. These were prewashed with ice-cold wash buffer (150 mM NaCl, 1% NP-40 in water). The beads were then washed 3× with ice-cold wash buffer by centrifugation at 800*g* for 1 min. The wash buffer was then removed and 4× SB with 100 mM DTT was added to the beads, and the samples were boiled for 10 min at 95 °C to denature the bead–protein interactions and recover the proteins. For pY IPs, the cells were treated with sodium orthovanadate ($VO_4^{3-}$; activated, Calbiochem, 5 ml, 5086050004) or water for 2 h at 37 °C before collection. Notably, all IP experiments were performed using ~90% of the sample as input and keeping 10% for the loading control.

### Immunohistochemistry
Fibroblast-contracted 3D collagen I matrices and xenograft tumours were embedded in paraffin after fixation in 10% neutral buffered formalin, as described previously[16,65]. The sections, 4-µm thick, were then stained with either haematoxylin and eosin (H&E) or primary antibodies against Ki67 (Dako, M7240, 1:500), EGFP (Invitrogen, A-11122, 1:800) or pan-cytokeratin (Invitrogen, MA5-13203, 1:25). The slides were scanned on a Pannoramic P1000 (3DHISTECH, 20×/0.8 objective).

### Invadopodia imaging
Eight-well dishes (µ-Slide 8 well ibiTreat; iBidi, 80826) for invadopodia assessment were coated with fluorescent gelatin, as described previously[70]. Briefly, Oregon green-488-conjugated gelatin (Thermo Fisher, G13186) and 5% (w/w) unlabelled gelatin (Sigma, G2500-100G)/sucrose (Sigma, S9378) solution were prewarmed to 37 °C. The wells were coated with 50 µg ml⁻¹ poly-ʟ-lysine (Sigma, P9155) and incubated at room temperature for 20 min. The solution was aspirated, and the wells were washed three times with PBS. A freshly made 0.5% glutaraldehyde solution (Sigma, 340855) was added to each well and incubated on ice for 15 min, followed by three washes with cold PBS. The Oregon green-488-conjugated gelatin was diluted with the unlabelled 5% stock gelatin in a 1:8 ratio and added to the wells. This gelatin mixture was then incubated for 30 min at 37 °C, in the dark. The excess mixture was removed by vacuum aspiration, and the wells were incubated in the dark for 10 min at room temperature, then washed three times with PBS. A freshly made 5 mg ml⁻¹ sodium borohydride (Sigma, 452882) solution was then added to the wells and incubated for 15 min at room temperature. The sodium borohydride was removed, wells washed three times with PBS and incubated with 70% ethanol for 30 min at room temperature. The dishes were then transferred to cell culture

laminar flow hoods and washed three times with sterile PBS. Before each experiment, the dishes were equilibrated with cell culture medium for >1 h at 37 °C.

To assess invadopodia formation, MM231 cells with Dox-inducible Src mutants in stably incorporated lentiviral cassettes were transfected with either siNTC or siDok1 siRNAs and treated with or without Dox (1 μg ml⁻¹) for 24 h before plating on fluorescent gelatin-coated iBidi dishes for 6 h. The cells were then fixed with 4% PFA (Thermo Fisher, 28908)/0.25% Triton X-100 for 10 min at 37 °C, before washing with PBS and blocking overnight with 2% bovine serum albumin (BSA) (Sigma, A8022)/1 M glycine (ITW Reagents, A1067)/PBS. The blocked samples were then incubated with SiR-Actin for 1 h at room temperature followed by staining with DAPI (Life Technologies, D1306). The imaging was performed using a spinning disk confocal (3i Marianas CSU-W1) microscope with a Zeiss Plan-Apochromat ×63/1.4 oil immersion objective.

Gelatin degradation was quantified by creating a mask for degraded areas per field of view. This was then normalized to the number of cells in each field measured using DAPI. The resultant 'degradation area per cell (units μm² per cell)' for each field of view was plotted for further statistical analysis using Prism 7 (GraphPad Software).

## Invasion into fibroblast-contracted 3D collagen I matrices

Fibroblast-contracted 'organotypic' 3D collagen I matrices for invasion assays were generated as described previously[16,65], using TIFs to contract the matrices. Quantification of the MM231 ITGB1(WT or YYFF) proliferation or invasion indices was performed in QuPath[64], using the positive cell detection algorithm over eight regions of interest. For invasion assessment, the fraction of cells in a region of interest that had invaded beyond 100 μm was compared with the total number of cells in that region.

## Malachite green phosphate detection

Detection of free phosphate associated with phosphatase activity was achieved through the use of a Malachite Green Phosphate Detection Kit (Cell Signaling, 12776) using the following recombinant biotinylated peptide fragments of ITGB1 (Genscript; Bio-ITGB1_27aa_tail: NAKWDTGENPIYKSAVTTVVNPKYEGK, Bio-ITGB1Y783P_27aa_tail: NAKWDTGENPI[pY]KSAVTTVVNPKYEGK, Bio-ITGB1Y795P_27aa_tail: NAKWDTGENPIYKSAVTTVVNPK[pY]EGK or Bio-ITGB1Y783PY795P_27aa_tail: NAKWDTGENPI[pY]KSAVTTVVNPK[pY]EGK. Each fragment (2,400 pmol per peptide per reaction) was incubated separately for 1 h at 37 °C with recombinant Shp2 (0.05 μg ml⁻¹; R&D Systems, 1894-SH-100) or PTP-PEST (0.05 μg ml⁻¹; SignalChem, P39-21G-10) in phosphatase buffer (HEPES buffer, pH 7.5 (50 mM)/EDTA (0.2 mM)/DTT (5 mM)/Triton X-100 (0.01%)), before incubation with Malachite Green Reagent (100 μl per reaction). The colour was allowed to develop for 15 min at room temperature, then absorbance read at 640 nm. The absorbance from the blank solution was then subtracted from the sample wells, and the free phosphate was calculated from a standard curve generated through titration of a phosphate standard and detection using the Malachite Green Reagent, as above.

## Molecular cloning

Illusia was initially synthesized (Supplementary Table 3) and cloned into the pDONR221 plasmid (Thermo Fisher) by GeneArt (Thermo Fisher). To generate the phosphorylation-defective pENTR221-Illusia(YYFF), pENTR221-Illusia(Y783F) or pENTR221-Illusia(Y795F) constructs or the phospho-mimetic pENTR221-Illusia(YYEE) construct, the YYFF_top/YYFF_bottom, Y783F_top/Y783F_bottom, Y795F_top/Y795F_bottom or YYEE_top/YYEE_bottom oligonucleotides (Supplementary Table 4) were annealed respectively and ligated into the NotI/BspEI digested pENTR221-Illusia backbone.

To generate the ITGB1 mutants, WT ITGB1 was PCR amplified from mCherry-Integrin-Beta1-N-18 using ITGB1_F and ITGB1_R

(Supplementary Table 4), before ligation into pENTR2b after digestion with KpnI/XhoI. The silent mutations were then introduced between base pairs 2731–2751 from gccttgcattactgctgatat to gccttgcGttactCctCatTt, to create an shRNA-resistant construct against the shRNA expressed by the pLKO.1-shBeta1 vector. The YYFF ITGB1 variant was then PCR amplified and religated back into pENTR2b with KpnI/XhoI, using the PCR primers ITGB1_F, ITGB1(YYFF)_R (Supplementary Table 4). These shRNA-resistant constructs were then tagged with mRuby2 using overlap-extension PCR on the pcDNA3-mRuby2, pENTR2b-ITGB1(WT) or -ITGB1(YYFF) templates with the ITGB1_OEC_F, ITGB1(WT)_OEC_R, ITGB1(YYFF)_OEC_R, mRuby2_OEC_F and mRuby2_OEC_R (Supplementary Table 4), ligating into the pENTR2b backbone after digestion with KpnI/XhoI.

Src(WT) and Src(Y527F) were PCR amplified from pBABE-Src-Rescue using the Src_F and Src_R or Src_Y527F_R primers respectively (Supplementary Table 4), then ligated into pENTR2b after digestion with KpnI/XhoI. To generate Src(K295R) and Src(E378G), the pENTR2b-Src(WT) plasmid underwent site-directed mutagenesis at Gene Universal. For Arg kinase, the pDONR223-ABL2 construct was used as a template for site-directed mutagenesis (Gene Universal) to first remove the K115M mutation present in the Addgene construct to yield the pENTR223-ABL2(WT) construct. This then underwent further site-directed mutagenesis to yield the pENTR223-K281M (Gene Universal).

The pENTR2b-Paxillin-EGFP was generated through PCR amplification of the Paxillin–EGFP cassette from the pEGFP–C2 backbone[71] using the primers Paxillin–EGFP_F and Paxillin–EGFP_R (Supplementary Table 4). The PCR fragment was then digested with XhoI/NotI and ligated into the similarly digested pENTR2b. Similarly, for pENTR2b–EGFP, the EGFP fragment was PCR amplified from the pEGFP-N1 vector using EGFP_F and EGFP_R. An acceptor-only FRET control vector was generated for YPet through PCR amplification using pENTR221-Illusia and the PCR primers YPet_F and YPet_R. Then pENTR2b backbone and PCR fragment were both digested with XhoI/KpnI and ligated together to yield pENTR2b-YPet.

Intermolecular FRET constructs were generated for Dok1 by overlap-extension PCR using the pDONR223-Dok1(no stop) and pcDNA3–Clover plasmids as templates and the Dok1_OEC_F, Dok1_OEC_R, Clover_OEC_F and Clover_OEC_R primers, ligating into pENTR2b after digestion with NcoI/EcoRV to generate pENTR2b-Dok1-Clover. For VPS35 and AnnexinA6, C-terminally tagged mScarlet constructs were synthesized at BioCat in the pENTR221 vector backbone.

Shp2 and PTP-PEST were PCR amplified from pCMV6-PTPN11 and pCMV-PTPN12, respectively, using the PTPN11_F/PTPN11_R1 or PTPN12_F/PTPN12_R1 primer pairs, respectively (Supplementary Table 4). These were then digested and ligated into the pENTR2b backbone to generate pENTR2b-PTPN11 and pENTR2b-PTPN12. These then underwent site-directed mutagenesis at Gene Universal to generate pENTR2b-PTPN12(D199A,C231S) and -PTPN11(D425A,C459S). To generate Clover-tagged constructs, Shp2 and PTP-PEST were PCR amplified from the WT and mutant constructs using the PTPN11_F/PTPN11_R2 or PTPN12_F/PTPN12_R2 primer pairs, respectively (Supplementary Table 4), followed by ligation of the digested fragments into pENTR2b-Dok1-Clover after excision of Dok1 using NotI/SalI.

For the BiFC/BiCAP, V1- or V2-tagged constructs were generated by LR reactions with the pDEST-ORF-V1 or pDEST-ORF-V2 plasmids and either pDONR223-Dok1(no stop) or pENTR221-ITGB1(no stop). The pENTR221-ITGB1(no stop) was generated by PCR of ITGB1 from pENTR2b-ITGB1(WT) using ITGB1_attB1_F and ITGB1_attB2_R, then a BP reaction (BP clonase II, Thermo Fisher Scientific) with pDONR221.

The above shuttle vectors were then subcloned into pEF.DEST51, pINDUCER20, pPB.DEST and pLenti6.3/TO/V5-DEST by performing an LR reaction (LR clonase II, Thermo Fisher Scientific) with the pENTR221 or pENTR2b constructs described above, as necessary (Supplementary Table 5). LR reactions were also performed as indicated into the

custom pDEST-ORF-mScarlet and pDEST-mScarlet-ORF gateway destination vectors, which were ligated into the pDEST-V2-ORF or pDEST-ORF-V2 backbones after PCR amplification (N-terminal primers: mScarlet_Nterm_F and mScarlet_Nterm_R, C-terminal primers: mScarlet_Cterm_F and mScarlet_Cterm_R) (Supplementary Table 4) and digested with ClaI/KpnI and ClaI/XbaI, respectively. All PCR reactions were performed using a high-fidelity DNA polymerase (Phusion II HS, F549S, Thermo Fisher), and all constructs were validated by analytical digests and sequencing and are available through Addgene (Supplementary Table 5).

For the pET28a-HaloTag-CNA35 construct, a HaloTag gene block (IDT) was digested with NheI/EcoRI and ligated into the similarly digested pET28a-EGFP-CNA35 backbone.

### Phosphopeptide ELISA

Wells of a 96-well Nunc-Immuno plate (Nunc, 442404) were coated with streptavidin (overnight at 4 °C), then washed with 0.1% TBS-T before blocking with 5% BSA (1 h at 37 °C). Recombinant biotinylated peptide fragments of ITGB1 (Genscript; Bio-ITGB1_27aa_tail: NAK-WDTGENPIYKSAVTTVVNPKYEGK or Bio-ITGB1Y783P_27aa_tail: NAKWDTGENPI[pY]KSAVTTVVNPKYEGK) were then added to wells and allowed to bind to the streptavidin coating for 1 h at room temperature. The plates were then washed 5× with 0.1% TBS-T before being treated for 30 min at 37 °C with recombinant Src (0.1 µg ml$^{-1}$; Signal-Chem, S19-10G-10), with/without ATP (20 µM), in kinase buffer (HEPES, pH 7.5 (60 mM)/MgCl$_2$ (5 mM)/DTT (1.25 mM)/MnCl$_2$ (5 mM)/Na$_3$VO$_4$ (3 µM)) as indicated. The phosphorylated tyrosine was then detected by first incubating with a pY primary antibody (PY20; 1:1,000, BD Biosciences, 610000) at room temperature and then with a horse radish peroxidase-conjugated anti-mouse secondary (1:2,000, Cell Signaling Technology, 7076) for 1 h at room temperature on a plate shaker (45 rpm). The plates were then washed three times with 0.1% TBS-T, 1-Step TMB (3,3',5,5' tetramethylbenzidine) and ELISA Substrate Solution (Thermo Fisher, 34028) was then added for 4 min before stopping the reaction with hydrogen peroxide (2 M) for 5 s per well and reading absorbance at 490 nm.

### Polyacrylamide hydrogels

The in-house stiffness hydrogels were prepared as described previously[72]. Briefly, glass-bottom dishes (D35-14-1N, Cellvis) were treated with bind silane solution (71.4 µl bind silane ((3-(methacryloyloxy) propyl)trimethoxysilane; Sigma, M6514), Glacial acetic acid (71.4 µl; Sigma, 33209) up to 1 ml in ethanol (96%)) for 30 min at room temperature and then washed twice with ethanol (96%). A total of 12 µl of gel mixture (0.5 kPa: 63 µl 40% acrylamide solution (Sigma, A4058), 10 µl 2% Bis acrylamide solution (Sigma, M1533), 399 µl PBS, 2.5 µl 20% ammonium persulfate (Sigma, A3678, diluted in MilliQ water) and 1 µl TEMED (Sigma, T9281); 60 kPa: 225 µl 40% acrylamide solution, 100 µl 2% Bis acrylamide solution, 175 µl PBS, 2.5 µl 20% ammonium persulfate and 1 µl TEMED) was applied to the dry dish(es), gently placing 13 mm glass coverslips on top of the gel mixture, allowing it to set for 1 h at room temperature, as described previously[72,73]. A sufficient amount of PBS was then added to completely cover the coverslip, before carefully removing the coverslip. Surface activation was performed with Sulfo-Sanpah (sulfosuccinimidyl 6-(4ammonium persulfate-azido-2ammonium persulfate-nitrophenylamino) hexanoate; 0.2 mg ml$^{-1}$ in 50 mM HEPES; Thermo Fisher Scientific, 22589) and $N$-ethyl-$N'$-(3-dimethylaminopropyl)carbodiimide hydrochloride (EDC; 2 mg ml$^{-1}$ in 50 mM HEPES; Sigma, 03450), applied to the gel surface for 30 min at room temperature with gentle agitation. The gels were then incubated in an ultraviolet oven (ultraviolet oven cleaner 342-220, Jelight Company) at 5 cm distance for 10 min, before washing three times with PBS and coating with 10 µg ml$^{-1}$ fibronectin (Millipore, 341631) and Collagen I (Millipore, 08-115), for a further 60 min at 37 °C.

### Proliferation assays

To assess MM231 ITGB1(WT or YYFF) cancer cell proliferation, 2,000 cells per well were seeded in 3× parallel 96-well plates, and a single plate was analysed on each day of the assay using the cell counting kit-8 (Sigma, 96992), as per the manufacturer's instructions. The relative cell density was measured from four wells per treatment condition by reading the absorbance at 450 nm after a 4 h incubation with the cell counting kit-8 reagent at 37 °C. Doubling times were obtained by fitting an exponential growth equation to the data using Prism 7 (GraphPad Software). Similarly, the IC$_{50}$ curves for Sara (SelleckChem, S1006) or SHP099 (SelleckChem, S8278) were performed for a panel of breast cancer cell lines by seeding 2,000 cells per 96-well plate and treating with serial dilutions of the inhibitors for 3 days, before using the cell counting kit-8 (Sigma, 96992) to detect relative cell density, as per the manufacturer's instructions. The IC$_{50}$ curves were then calculated from normalized dose response curves in Prism 7 (GraphPad Software).

### Recombinant protein purification

As described previously[65], BL21 competent *Escherichia coli* (NEB, C2530H) were transformed with the pET28a-HaloTag-CNA35 plasmid and grown overnight on a shaker at 37 °C to yield a 250 ml culture with an OD$_{600}$ = 0.6. This culture was then incubated with IPTG (500 µM; Thermo Fisher, R0392) on a shaker overnight at 30 °C. The bacteria were then pelleted by centrifugation at 6,000$g$ for 15 min at 4 °C before discarding the supernatant and resuspending the pellet in 9 ml TBS with protease inhibitors (cOmplete Mini, EDTA-free, Roche). To this solution, 1 ml Bugbuster (Millipore, 10× protein extraction reagent, 70921-50ML), 1 µl Benzonase Nuclease (Sigma, E1014-5KU), 4.5 µl DNAse I (Sigma, 11284932001) and lysozyme from chicken egg white (Sigma, 62970) were added. This mixture was then rotated for 30 min at 4 °C before centrifugation at 6,000$g$ for 1 h. The supernatant was then purified using a kit for His-tagged proteins (Macherey-Nagel, Protino Ni-TED2000 packed columns, 745120.25). The elution buffer was then exchanged against 4× changes of PBS using centrifugal filters (Millipore, Amicon Ultra-4, 10K UFC801024).

### RNA isolation, complementary DNA generation and quantitative real-time PCR

The RNA from cultured cells was collected and isolated using the NucleoSpin RNA kit (Macherey-Nagel). The colonized mouse lungs, snap-frozen in liquid nitrogen, were homogenized with ceramic beads (Omni, SKU 19-627) and a bead Ruptor (Omni, SKU 25-010). RNA was isolated using a phenol-chloroform extraction method according to the manufacturer's instructions (TRIsure, Bioline, BIO-32033). For cDNA synthesis, 1 µg of the extracted RNA was then used as a template for the high-capacity cDNA reverse transcription kit (Applied Biosystems). Each PCR reaction was performed using 100 ng cDNA and the appropriate Taqman gene expression assays (FAM; Thermo Fisher) for each gene, according to the manufacturer's instructions (Thermo Fisher, TaqMan Fast Advanced Master Mix, 4444557). The following Taqman gene expression assays were used: Colfilin-2/*CFL2* (Hs01071313_g1), Cofilin-1/*CFL1* (Hs02621564_g1), Cortactin/*CTTN* (Hs01124232_m1), Tks5 (*SH3PXD2A*, (Hs01046307_m1), Dok1 (*DOK1*, Hs00796733_s1), EGFP (Mr04329676_mr), receptor-type tyrosine-protein phosphatase-like N/*PTPRN* (Hs00160947_m1), PTP-PEST/*PTPN12* (Hs00184747_m1), myotubularin-related protein 8/*MTMR8* (Hs00250307_m1), cell division cycle 14B/*CDC14B* (Hs00269351_m1), transmembrane phosphatase with tensin homology/*TPTE* (Hs00276201_m1), myotubularin/*MTM1* (Hs00896975_m1), PTP non-receptor-type 20/*PTPN20* (Hs00944181_m1), receptor-type tyrosine-protein phosphatase O/*PTPRO* (Hs00958177_m1), Slingshot homologue 2/*SSH2* (Hs00987189_m1), dual-specificity phosphatase 8/*DUSP8* (Hs01014943_m1), dual-specificity phosphatase 16/*DUSP16* (Hs01015508_m1), dual-specificity phosphatase 3/*DUSP3* (Hs01115776_m1) and glyceraldehyde-3-phosphate dehydrogenase (GAPDH)/*GAPDH*

(Hs02786624_g1). The relative mRNA expression levels were normalized to *GAPDH*, and quantification was performed using the ΔΔCt method[74]. To ascertain relative human cancer cell number in the mouse lung, human-specific GAPDH (Hs04420566_g1) levels, were normalized to GAPDH signal from primers with cross-reactivity to both mouse and human transcripts (Hs02786624_g1), as achieved previously[75].

## SE-FRET microscopy

For the siRNA screen, MM231 cells with stable Illusia expression were seeded in 96-well plates, in which the PTP siRNA library (including three independent siRNAs/target) were first aliquoted in OptiMEM with RNAiMAX to a final concentration of 5 nM per siRNA per well. Incubating overnight, siRNA-transfected cells were split into two 96-well plates, where one was used for collecting RNA (as described above), and the other had a glass-bottom (high performance no. 1.5 cover glass, Cellviz, P96-1.5H-N) suitable for imaging applications. After 72 h, the cells in the glass-bottom plate were fixed with 4% PFA at 37 °C for 10 min. The plates were loaded by a plate handler (Twister II, Caliper) from ambient storage into a BD pathway 855 High-Content Analyzer (Beckton Dickinson) customized with high-power LEDs (Thorlabs SOLIS 445 and 565D), where they were sequentially imaged under control of iLinkPro 1.1 software. Images were acquired using a 20× N.A. 0.75 air objective (Olympus) together with 438/24 nm brightline (Semrock) or HQ500/20 nm (Chroma) excitation filters, 458 nm (Semrock) or 515 nm (Chroma 72100) dichroics and 483/32 nm (Semrock) or 542/27 nm emission filters. The LEDs were electronically shuttered through an optocoupler/double-inverter TTL circuit for precise exposure times and synchronization with camera acquisition[76]. To calculate normalized FRET (NFRET), spectral bleed-through coefficients ($BT_{donor}$ and $BT_{acceptor}$) were calculated using the FRET and Colocalization analyser[77] in Fiji with MM231 cells transfected with donor- (pEF.DEST51-mTurquoise) or acceptor-only (pEF.DEST51-YPet) controls using Lipofectamine 3000, as per the manufacturer's instructions. These coefficients were then fed into the equation below to give the NFRET per cell[78], which were normalized for each replicate through conversion to robust $z$-scores using the stats package (v3.6.2) in RStudio (v2022.12.10). Outliers were removed using ROUT ($Q = 0.1\%$) in Prism 7 (GraphPad Software),

$$NFRET = \frac{I_{FRET} - (I_{donor} \times BT_{donor}) - (I_{acceptor} \times BT_{acceptor})}{\sqrt{I_{donor} \times I_{acceptor}}}.$$

For the live imaging of cancer invasion, MM231 cells with stable Illusia expression were embedded inside 3D collagen I matrices with non-fluorescent TIFs, as described above. After 14 days of fibroblast-driven contraction, the matrices were cut to fit into eight-well Ibidi wells and imaged on a Zeiss LSM880 inverted confocal microscope using a 63×/0.75 Zeiss LD Plan-NEOFLUAR water immersion objective with 405 and 514 nm excitation, detecting simultaneously on PMT detectors for the donor (460–490 nm) and FRET/acceptor (520–540 nm) channels. NFRET was calculated as described above for each pixel of these images.

## Statistics and reproducibility

The sample size for studies was chosen according to previous studies in the same area of research[79]. Data collection and analysis were not performed blind to the conditions of the experiments. Prism 7 (GraphPad Software) was used for all statistical analyses, as indicated in the respective figure legends and the Statistical Source Data file. *P* values less than 0.05 were considered to be statistically significant. Individual data points per condition per replicate are shown, and *n* numbers are indicated in all figure legends, as appropriate. All micrographs (western blots and microscopy images) are representative of three or more independent experiments (*n* numbers are shown in the accompanying

analyses for each micrograph). The original, uncropped western blots can be found in the Source data file for unprocessed western blots.

## 3D cell morphology assays

Illusia-expressing MM231 cells were embedded in a mixture of neutralized collagen I (2.5 ml collagen I acid-extracted rat tail homemade as described previously[16,65], 300 μl 10× minimum essential medium (Gibco, 21430020), 300 μl NaOH (0.22 mM) with 300 μl FBS containing the cells), incubated at 37 °C until set, then overlayed with growth medium containing DMSO, Sara (1 μM; SelleckChem, S1006) or SHP099 (100 nM; SelleckChem, S8278). The matrices were also spiked with Alexa647-labelled collagen I (1:1,000), labelled as described previously[72]. After 24 h, matrices were fixed in 4% PFA/PBS for 10 min at 37 °C and washed three times with PBS. The imaging was performed on a Zeiss 3i CSU-W1 spinning disk confocal microscope using SlideBook 6 acquisition software with a 63×/0.75 Zeiss LD Plan-NEOFLUAR objective with water immersion.

## TIRF microscopy

Total internal reflection fluorescence (TIRF) imaging was performed using a Deltavision OMX SR microscope, with a 60× Olympus APO N TIRF objective after seeding cells in MatTek dishes (part no. P35G-1.5-7-C) for 6 h before fixation for 10 min at 37 °C with 4% PFA in PEM buffer (EGTA (10 mM; VWR Chemicals, 0732), $MgSO_4$ (1 mM; Fluka Analytical, 00627), PIPES (pH 6.9; 100 mM; Sigma, P6757), sucrose (75 mM; Sigma, S9378), Triton X-100 (0.2%; Sigma, T8787) in $H_2O$). The samples were then blocked (2% BSA and glycine (1 M; PanReac AppliChem, A1067) in PBS) for 1 h at room temperature. The blocked samples were then stained with SiR-Actin (1 μM, Spirochrome, sc001), in parallel with primary antibodies against Paxillin ((Y113); 1:100, Abcam, ab32084) or active ITGB1 (clone 12G10; 1:25 from 0.25 mg ml⁻¹ stock; in-house production) in PBS overnight at 4 °C. The samples were then washed twice with PBS before staining with appropriate secondary antibodies for 1 h at room temperature, and a further two washes with PBS. Analysis of microscopy data was performed in Fiji (NIH), assessing colocalization using Pearson's coefficients from the Coloc2 plugin. For live imaging of paxillin, MM231 cells with paxillin–EGFP were seeded in MatTek dishes overnight, treated with SHP099 (100 nM) or DMSO for 1 h.

## Western blot

The protein lysates from cultured cell lines were prepared in TXLB lysis buffer (50 mM HEPES, 0.5% sodium deoxycholate, 0.1% SDS, 10 mM $Na_3VO_4$, 1% Triton X-100, 0.5 mM EDTA, 50 mM NaF, along with protease (cOmpleteTM Mini, EDTA-free, Roche) and phosphatase (PhosSTOP, Roche) inhibitor cocktails), adjusting volumes according to protein concentration (DC protein assay kit, Bio-Rad, 5000111). Gel electrophoresis was performed to separate proteins (Mini-PROTEAN TGX Precast Gels 4–20%, Bio-Rad, 4561096) that were then transferred onto a nitrocellulose membrane (Trans-Blot Turbo Transfer System, Bio-Rad) and blocked with AdvanBlock-Fluor (Advansta, R-03729-E10). The primary antibodies against ITGB1 (1:1,000, Abcam, ab52971), ITGB1(Y783) (1:500, Abcam, ab62337), PTP-PEST (1:1,000, Cell Signalling, 14735), Shp2 (1:1,000, Cell Signalling, 3397), Src (1:1,000, Cell Signalling, 2108), Src(Y416) (1:500, Cell Signalling, 2101), Arg (1:1,000, Abcam, ab134134), p130Cas (1:1,000, Santa Cruz, sc-20029), p130Cas(Y165) (1:1,000, Cell Signalling, 4015), p130Cas(Y410) (1:1,000, Cell Signalling, 4011), pY (1:1,000, BD Biosciences, 610000), GFP (1:1,000, Thermo Fisher, A11122), RFP (1:1,000; Chromotek, 6g6-100), Dok1 (1:1,000, Abcam, ab8112), VPS35 (1:1,000, Abcam, ab10099), Cofilin (D3F9; 1:1,000, Cell Signalling, 5175), Tks5 (1:500, Millipore, MABT336), β-Actin (1:10,000, Sigma, A1978), Cortactin (CTTN; clone 4F11; 1:500, Millipore, 05-180), annexin A6 (1:500, Abcam, ab31026) and GAPDH (1:10,000; Hytest, 5G4MAB6C5) were incubated overnight at 4 °C in AdvanBlock-Fluor. The membranes were washed between primary and secondary antibody treatments with TBS-T. IRDye

secondary antibodies (Li-Cor, 1:5,000 diluted in TBS-T) were incubated for at least 1 h at room temperature, before detection on an Odyssey fluorescence imager CLx (Li-Cor). To assess broad inhibition of PTPs, sodium orthovanadate ($VO_4^{3-}$; activated, Calbiochem, 5 ml, 5086050004) was used. Densitometry analysis was performed in Fiji (NIH) by normalizing the signal to the GAPDH loading control.

## Reporting summary

Further information on research design is available in the Nature Portfolio Reporting Summary linked to this article.

## Data availability

All data are available in the main text or the Supplementary Information. In addition, the mass spectrometry proteomics data have been deposited to the ProteomeXchange Consortium via the PRIDE[79] partner repository with the dataset identifier PXD048646. The plasmids generated in this study are available through Addgene, as indicated in Supplementary Table 5, or by contacting one of the corresponding authors. All other data supporting the findings of this study are available from the corresponding author on reasonable request. Source data are provided with this study.

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

## Acknowledgements

We thank P. Laasola, J. Siivonen and R. Mahran for technical assistance and the Ivaska lab for scientific discussion and feedback on the manuscript. We also thank N. Pasquier, A. Isomursu and M. Mathieu for providing necessary reagents for the completion of the study, and for critical reading, we thank H. Hamidi. For support with applying the basement membrane invasion assay, we thank R. Staneva and D. Vignjevic (Institut Curie, Paris, France). For services, instrumentation and expertise at Turku Bioscience (University of Turku, Turku, Finland), we thank the Cell Imaging and Cytometry Core, the Turku Proteomics Facility and the Genome Editing Core, all supported by Biocenter Finland. We also thank the Turku Screening Unit lab automation site, part of the DDCB platform supported by Biocenter Finland, for services, instrumentation and expertise. The clone pENTR223-SH3PXD2A was from the ORFeome library at the Genome Biology Unit core facility, supported by HiLIFE and the Faculty of Medicine, University of Helsinki, and Biocenter Finland. The histological methods were performed by the Histology core facility of the Institute of Biomedicine, University of Turku, Finland. This work was supported by the Finnish Cancer Institute (K. Albin Johansson Professorship to J.I.); a Research Council of Finland research project (grant no. 325464 to J.I.) and Centre of Excellence programme (grant nos. 346131 and 364182 to J.I.); the Cancer Foundation Finland (to J.I.); the Sigrid Juselius Foundation (to J.I.); and the Research Council of Finland InFLAMES Flagship Programme (grant nos. 337530 and 357910). J.R.W.C. was supported by the European Union's Horizon 2020 research and innovation programme under the Marie Sklodowska-Curie grant agreement (grant no. 841973), a Research Council of Finland postdoctoral research grant (grant no. 338585) and Research Fellowship (grant no. 360775). G.F. was supported by a Research Council of Finland postdoctoral research grant (grant no. 332402) and the Turku Collegium for Science, Medicine and Technology. M.G. was supported by a Worldwide Cancer Research grant (no. 23-0123 to J.I.).

## Author contributions

J.R.W.C., J.K., O.J., J.K., G.F., M.G. and D.K. performed the experiments and analysed the data. J.R.W.C. and J.I. wrote the paper. All authors commented on the manuscript and approved the final version.

## Funding

## Competing interests

The authors declare no competing interests.

## Additional information

**Extended data** are available for this paper at https://doi.org/10.1038/s41556-025-01663-4.

**Correspondence and requests for materials** should be addressed to James R. W. Conway or Johanna Ivaska.

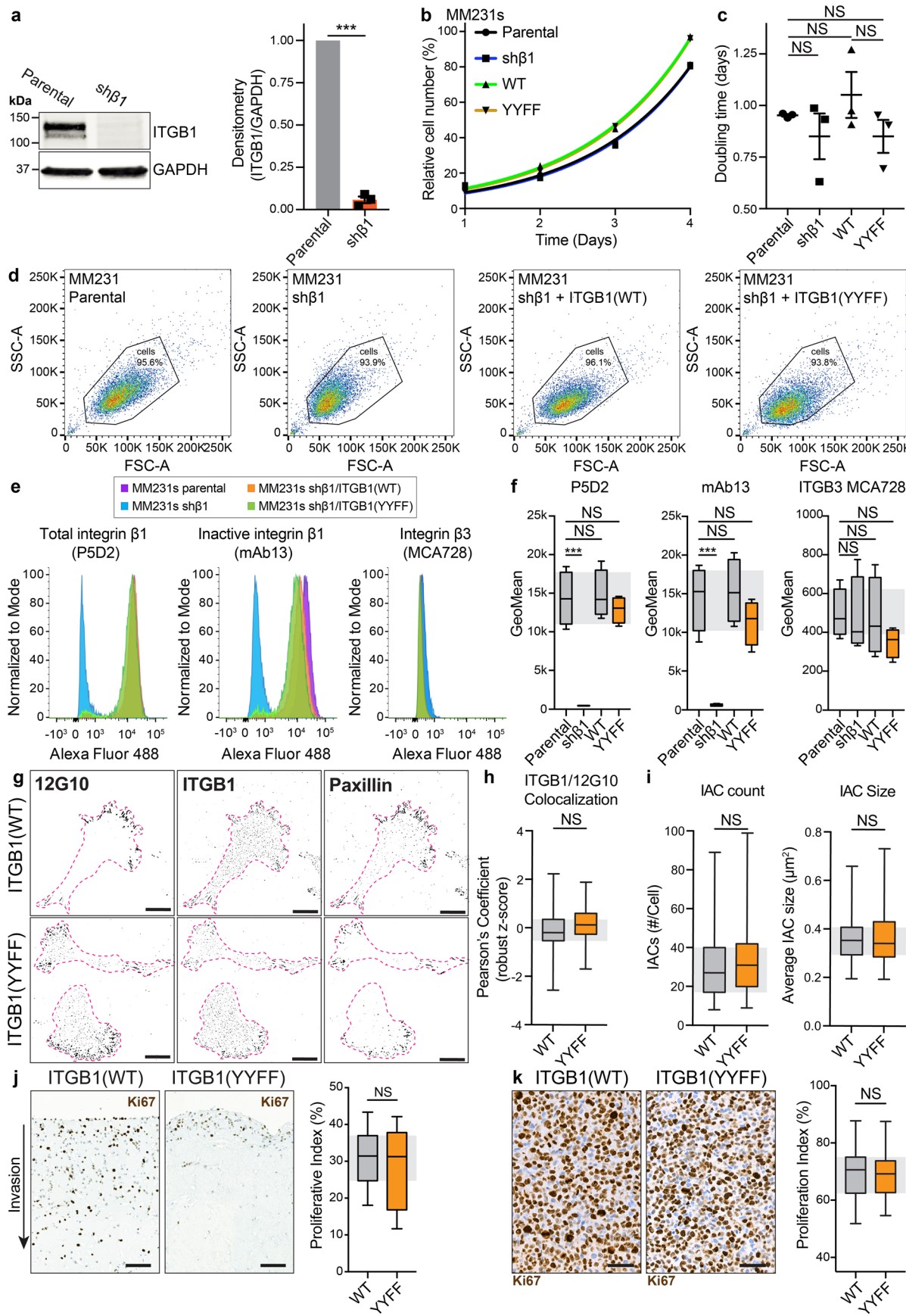

**Extended Data Fig. 1 | See next page for caption.**

**Extended Data Fig. 1 | Mutation of the ITGB1 NPxY tyrosine (Y) residues to non-phosphorylatable phenylalanine (F) does not change surface integrin levels, or cancer cell proliferation. a**, Representative western blot (left) and densitometry (right) of ITGB1 levels after shRNA-mediated KD in MM231 cells (shβ1; n = 3 biological replicates; significance assessed using a one-sample two-tailed *t*-test against the normalised control value; ***p < 0.001). Data are mean ± s.e.m. **b** & **c**, Representative curves (**b**) and doubling times (**c**) from the relative cell density of parental and shβ1 MM231 cells, and shβ1 MM231s with stable ITGB1(WT or YYFF) reexpression (ITGB1(WT or YYFF)-mRuby2 transposon vectors, described in methods; n = 3 biological replicates; significance assessed using a one-way ANOVA with a Šidák correction for multiple comparisons; NS, not significant). **d**, Gating strategy for the flow cytometry data presented in **e** & **f**. **e** & **f**, Representative histograms (**e**) and quantification (**f**) of the surface expression of total ITGB1 (P5D2), inactive ITGB1 (mAb13) and total integrin β3 (MCA728) using flow cytometry in shβ1 MM231s with or without ITGB1 reexpression as indicated (n = 4 biological replicates; significance assessed using a one-way ANOVA with a Tukey correction for multiple comparisons; NS, not significant; ***p < 0.001). **g** & **h**, TIRF images (**g**) of MM231 shβ1 cells with ITGB1 WT or YYFF reexpression (cells outlined with pink dashed lines) and quantification of colocalization (**h**) between active integrin staining (12G10 antibody) and either mRuby2-tagged ITGB1 WT (average Pearson's r of 0.4853) or YYFF (average Pearson's r of 0.5012; n = 65 [ITGB1 (WT)] and 63 [ITGB1(YYFF)] cells pooled from

three biological replicates; significance assessed using an unpaired two-tailed Student's *t*-test with a Welch's correction; NS, not significant). **i**, Analysis of paxillin staining in MM231 cells from (**g**) to compare IAC average number and size per cell (n = 87 [ITGB1 (WT)] and 85 [ITGB1 (YYFF)] cells pooled from four biological replicates; NS, not significant). **j**, Representative images of invading MM231 ITGB1(WT or YYFF) cells (left) and quantification of their proliferation (right) from the fibroblast-contracted 3D collagen I invasion assays in Fig. 1c. Cells are stained with the proliferation marker Ki67. Quantification was achieved by normalising the number of Ki67-positive nuclei to the total number of cells/region (n = 24 regions for each cell line pooled from three biological replicates; significance assessed using an unpaired two-tailed Student's *t*-test with a Welch's correction; NS, not significant). Scale bars, 100 μm. **k**, Representative images (left) and quantification (right) of Ki67-stained subcutaneous xenografts of MM231 cells with either ITGB1(WT or YYFF) from Fig. 1e. Quantification of positive (brown) to negative (blue) staining of Ki67 in 400 μm² regions of interest from subcutaneous xenografts is shown (n = 9 mice (WT) or 11 mice (YYFF); 5 regions/mouse/condition); significance assessed using an unpaired two-tailed Student's *t*-test with a Welch's correction; NS, not significant). Scale bars, 50 μm. Source data and exact p-values are provided in the statistical source data file. Boxplots represent median and interquartile range. Whiskers extend to min and max values. Grey areas on boxplots highlight the IQR of the control conditions.

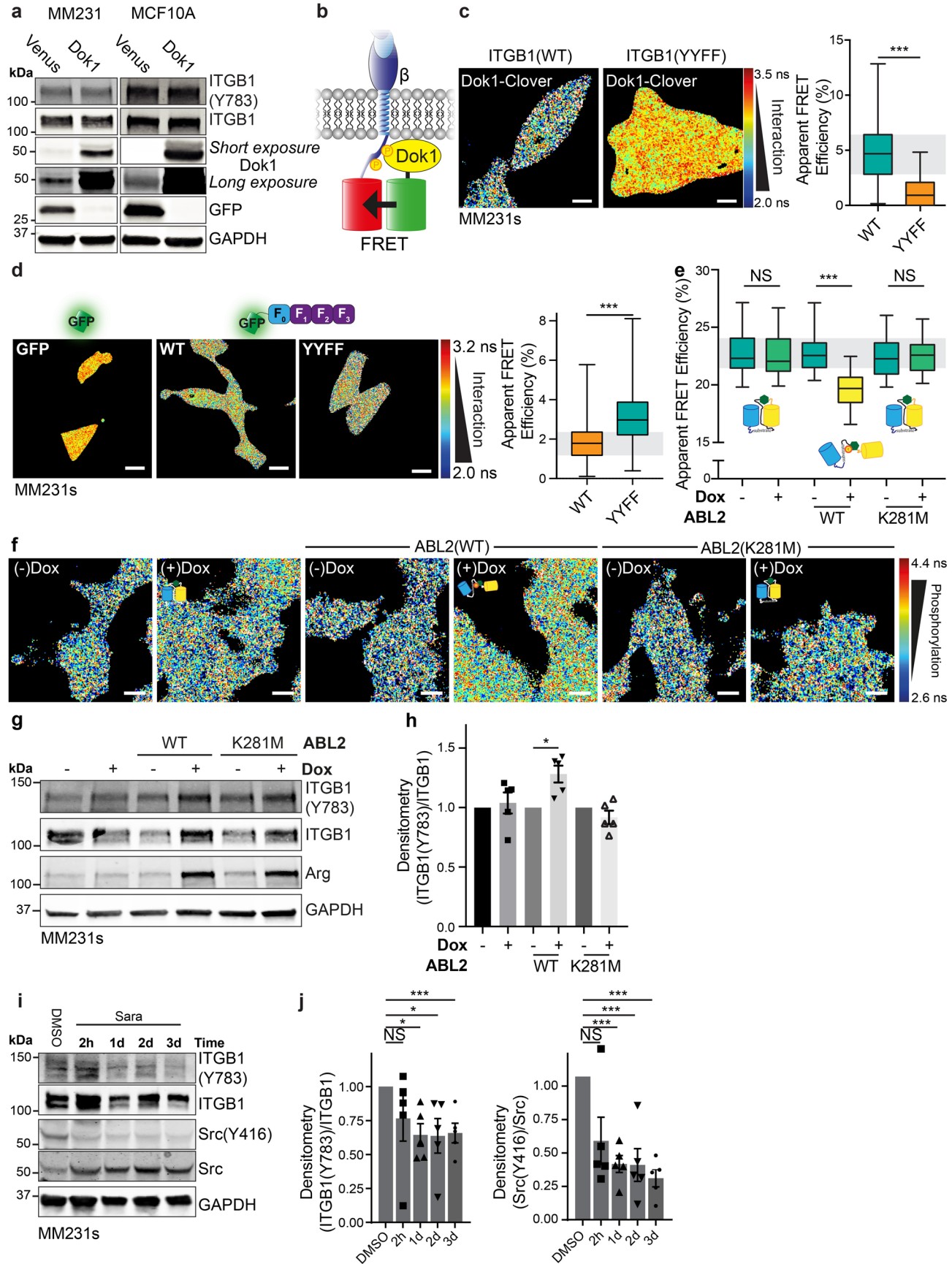

Extended Data Fig. 2 | See next page for caption.

**Extended Data Fig. 2 | Arg (ABL2) phosphorylates the ITGB1 NPxY sites. a**, Representative western blots of MM231 and MCF10A cells expressing either Venus or Dok1 (n = 4 biological replicates). **b**, A schematic of the intermolecular FRET approach between Dok1 and ITGB1 (WT or YYFF). **c**, Representative FLIM-FRET images (left) and quantification of apparent FRET efficiency (right) from MM231 shβ1 cells reexpressing mRuby2-tagged ITGB1(WT or YYFF) and transfected with Dok1-Clover (n = 60 cells in each condition pooled from three biological replicates; significance assessed using an unpaired two-tailed Student's $t$-test with a Welch's correction; ***p < 0.001). Scale bars, 10 μm. **d**, Representative FLIM-FRET images (left) and quantification of apparent FRET efficiency (right) from MM231 shβ1 cells reexpressing mRuby2-tagged ITGB1(WT or YYFF), and transfected with GFP-$F_0F_3$ talin head domain fragment (n = 62 cells in each condition pooled from three biological replicates; significance assessed using an unpaired two-tailed Student's $t$-test with a Welch's correction; ***p < 0.001). Scale bars, 20 μm. **e** & **f**, Apparent FRET efficiencies (**e**) and representative images (**f**) of MM231 cells stably expressing Illusia and Dox-inducible ABL2(WT) or kinase-dead ABL2(K281M) ± Dox treatment. Parental cells treated with Dox are used as an additional control (n = 60 cells in each condition pooled from three biological replicates; significance assessed using a one-way ANOVA with a Šidák correction for multiple comparisons; NS, not significant, ***p < 0.001). Scale bars, 10 μm. **g** & **h**, Representative western blot (**g**) and densitometry (**h**) analysis of ITGB1 phosphorylation levels in MM231 cells with Dox-induced ABL2(WT) or ABL2(K281M) overexpression (n = 5 biological replicates; significance assessed using a one-sample two-tailed $t$-test against the normalised control values for each cell line without Dox; NS, not significant, *p < 0.05). Data are mean ± SEM. **i** & **j**, Representative western blot (**i**) and densitometry (**j**) of MM231 cells treated with saracatenib (Sara) for 0, 2, 24, 48 and 72 h (1 μM; n = 5 biological replicates; significance assessed using a one-sample two-tailed $t$-test against the normalised DMSO control). Source data and exact p-values are provided in the statistical source data file. Boxplots represent median and interquartile range. Whiskers extend to min and max values. Grey areas on boxplots highlight the interquartile range of the control conditions.

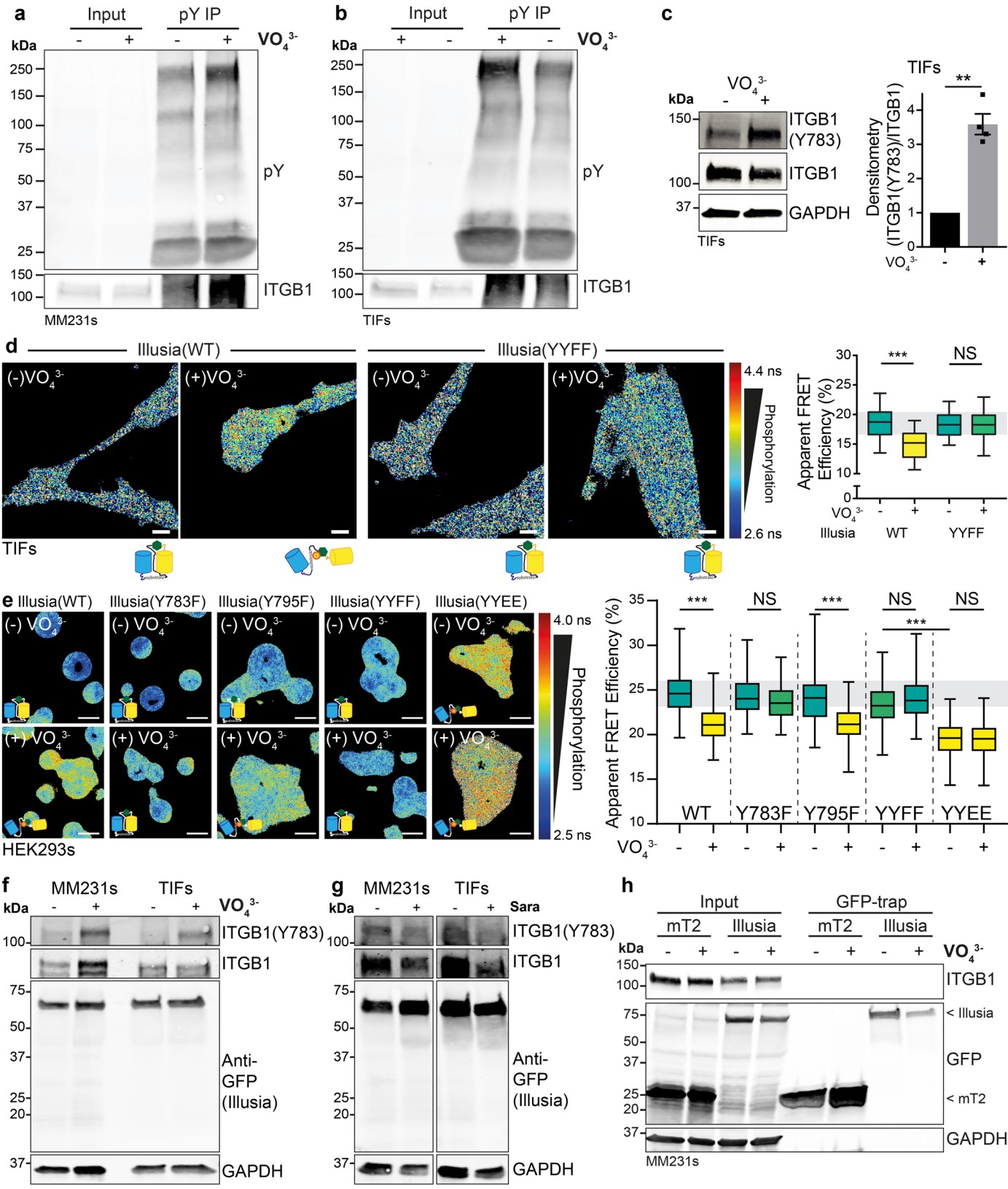

**Extended Data Fig. 3 | See next page for caption.**

**Extended Data Fig. 3 | Validation of the phosphorylation-dependent changes of Illusia and the active dephosphorylation of ITGB1 in cancer and normal cells. a & b,** Representative immunoprecipitation (IP) of phosphorylated ITGB1 in MM231 (**a**) and TIF (**b**) cells using anti-pY beads after $VO_4^{3-}$ treatment (100 mM, 2 h; n = 3 biological replicates). **c,** Representative western blot and densitometry analysis of ITGB1 phosphorylation levels in TIF cells after $VO_4^{3-}$ treatment (100 mM, 2 h; n = 4 biological replicates; significance assessed using a one-sample two-tailed $t$-test against the normalised control value without $VO_4^{3-}$; **p < 0.01). Data are mean ± SEM. **d,** Representative images (left) and quantification of apparent FRET efficiency (right) of $VO_4^{3-}$ (100 mM, 2 h) treated TIF cells stably expressing Illusia (WT), or a non-phosphorylatable mutant (YYFF) (n = 60 cells in each condition pooled from three biological replicates; significance assessed using a one-way ANOVA with a Šidák correction for multiple comparisons; NS, not significant, ***p < 0.001). Scale bars, 10 μm. **e,** Representative FLIM-FRET images (left) and apparent FRET efficiencies (right)

from HEK293 cells transfected with different Illusia variants (WT, Y783F, Y795F, YYFF and YYEE; n = 127 [WT (−)$VO_4^{3-}$], 152 [WT (+)$VO_4^{3-}$], 113 [Y783F (−)$VO_4^{3-}$], 120 [Y783F (+)$VO_4^{3-}$], 109 [Y795F (−)$VO_4^{3-}$], 117 [Y795F (+)$VO_4^{3-}$], 115 [YYFF (−)$VO_4^{3-}$], 121 [YYFF (+)$VO_4^{3-}$], 106 [YYEE (−)$VO_4^{3-}$], and 109 [YYEE (+)$VO_4^{3-}$] cells pooled from four biological replicates; significance assessed using one-way ANOVA with a Šidák correction for multiple comparisons; NS, not significant, ***p < 0.001). Scale bars, 20 μm. **f & g,** Representative western blots of MM231 and TIF cells stably expressing Illusia and treated with $VO_4^{3-}$ (**f**; 100 mM, 2 h; n = 3 biological replicates) or Sara for 24 h (**g**; 1 μM; n = 4 biological replicates). **h,** Representative GFP-trap IP of mT2 or Illusia from MM231 cells stably expressing the reported constructs (n = 3 biological replicates). Source data and exact p-values are provided in the statistical source data file. Boxplots represent median and interquartile range. Whiskers extend to min and max values. Grey areas on boxplots highlight the interquartile range of the control conditions.

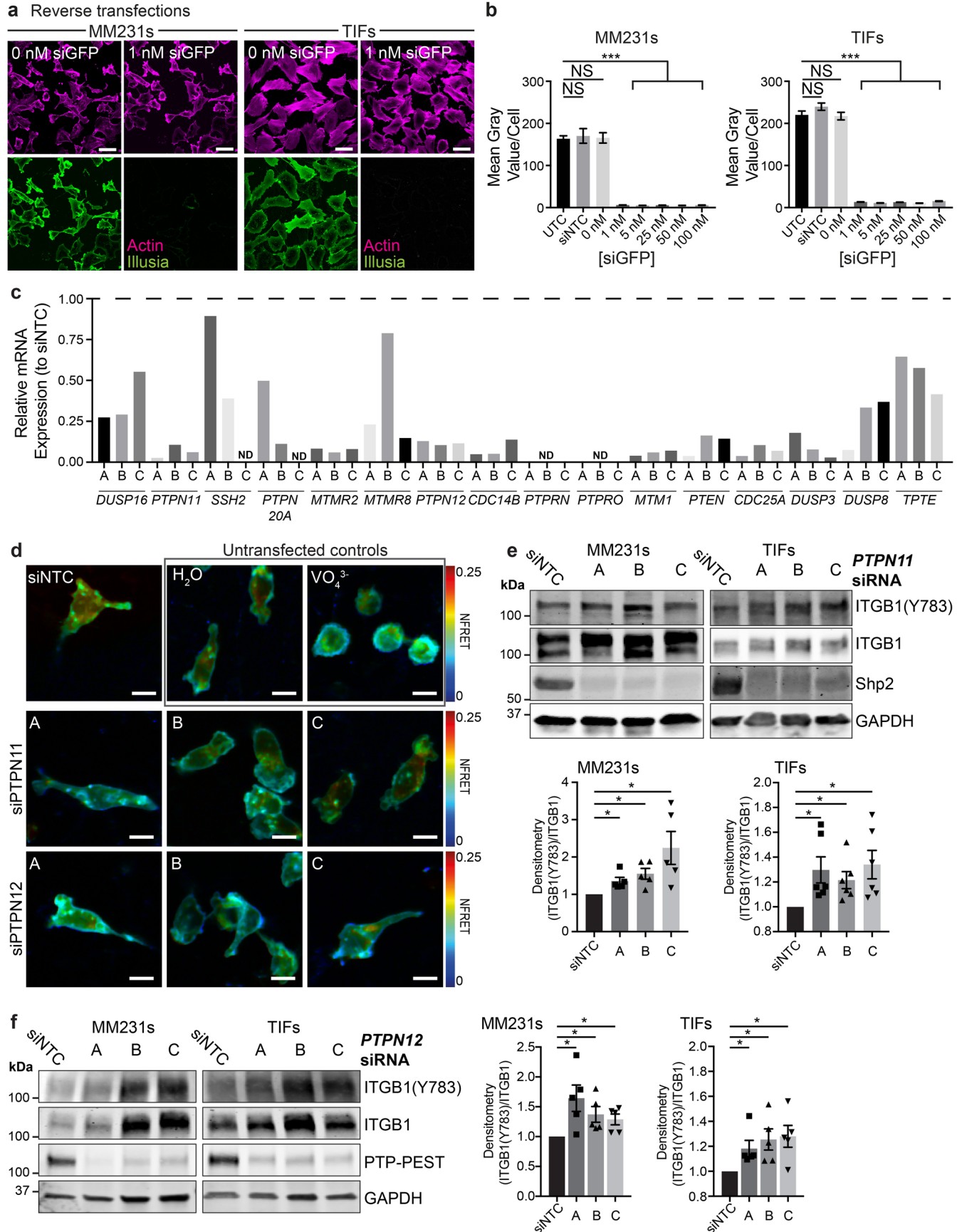

**Extended Data Fig. 4 | See next page for caption.**

**Extended Data Fig. 4 | Validation of siRNA knock-down of a subset of hits that showed significant changes in the Illusia FRET screen. a & b**, Representative images (**a**; Scale bars, 50 µm) and quantification of reverse transfections in MM231 and TIF cells using an siRNA against GFP; actin staining (SiR-Actin, magenta) and Illusia (Green) (UTC, untransfected control; siNTC, non-targeting control siRNA) (MM231s, n = 220 (UTC), 70 (siNTC), 122 (0 nM), 140 (1 nM), 79 (5 nM), 89 (25 nM), 117 (50 nM) and 98 (100 nM) | TIFs, n = 179 (UTC), 155 (siNTC), 174 (0 nM), 183 (1 nM), 224 (5 nM), 199 (25 nM), 118 (50 nM) and 198 (100 nM) cells pooled from three biological replicates; significance assessed using a one-way ANOVA with a Dunnett's correction for multiple comparisons; NS, not significant, ***p < 0.001). Data are mean ± SEM. **c**, Parallel qRT-PCR from RNA samples collected from the RNAi FRET screen cells targeting the top 16 hits (n = 1 experiment, assessed in triplicate). ND, Not detected. **d**, Representative SE-FRET images of MM231s stably-expressing Illusia showing the normalised FRET (NFRET) from the untransfected controls with or without $VO_4^{3-}$, and siRNAs targeting *PTPN11* and *PTPN12*, against the NTC siRNA. Scale bars, 20 µm. **e & f**, Representative western blots validating siRNA KD efficiency and ITGB1 phosphorylation levels in MM231 (left; n = 5 biological replicates; densitometry for ITGB1(Y783)/ITGB1 is displayed here. Densitometry for Shp2/GAPDH is in Fig. 4e) and TIF (right; n = 6 biological replicates; densitometry for ITGB1(Y783)/ITGB1 is displayed here. Densitometry for Shp2/GAPDH is in Fig. 4f) cells transfected with three different siRNAs (A, B and C) against *PTPN11* (**e**) or *PTPN12* (**f**). Source data and exact p-values are provided in the statistical source data file.

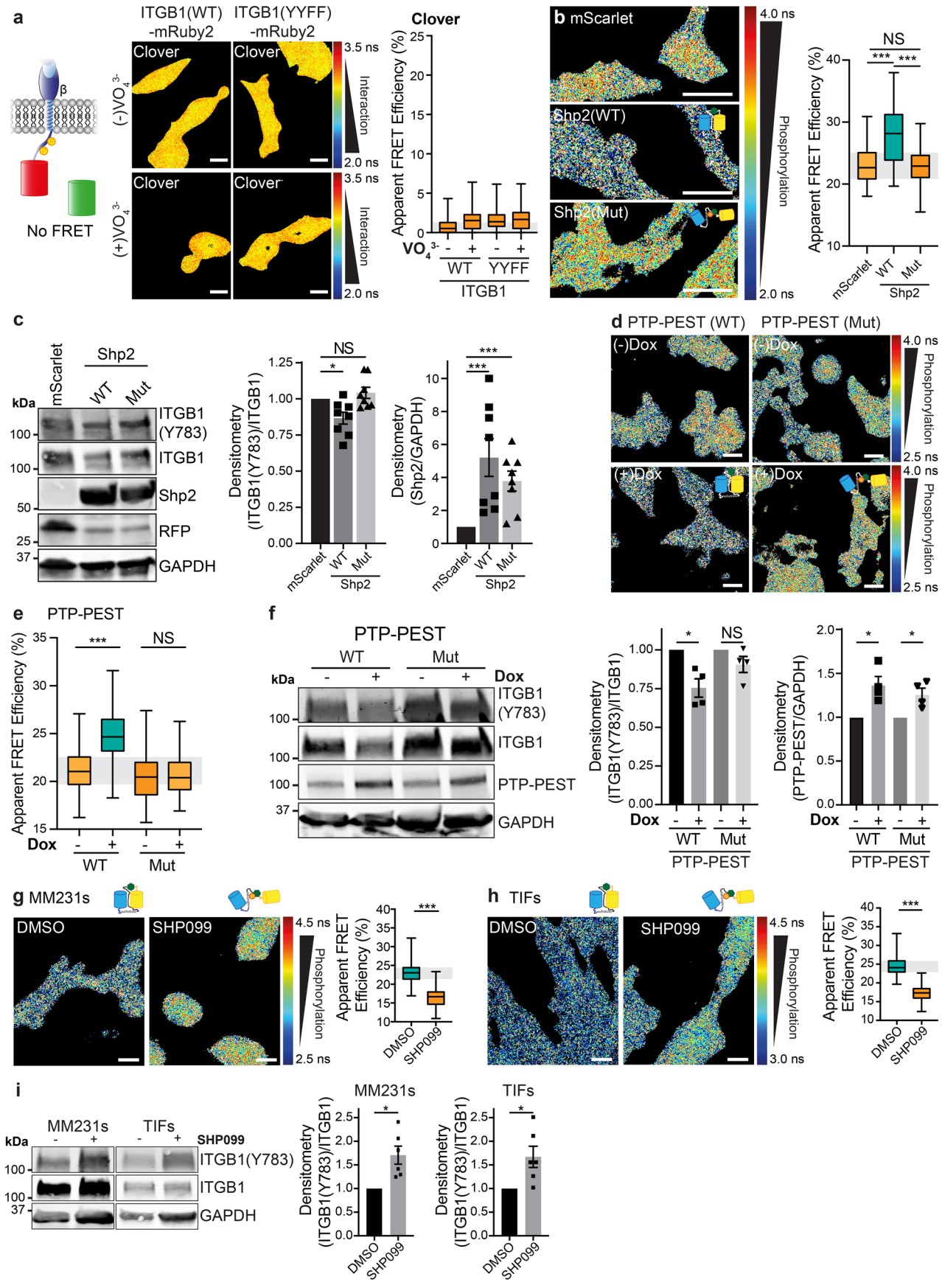

**Extended Data Fig. 5 | See next page for caption.**

**Extended Data Fig. 5 | Inhibition or overexpression of Shp2 or PTP-PEST modulates ITGB1 phosphorylation. a**, Schematic of the FRET experiment (left), with representative FLIM-FRET images (middle) and quantification of apparent FRET efficiency (right) of MM231 cells with stable expression of either ITGB1(WT)-mRuby2 or ITGB1(YYFF)-mRuby2 transfected with Clover and treated with VO$_4$$^{3-}$ (n = 65 [ITGB1(WT) (−)VO$_4$$^{3-}$], 70 [ITGB1(WT) (+)VO$_4$$^{3-}$], 72 [ITGB1(YYFF) (−)VO$_4$$^{3-}$], and 66 [ITGB1(YYFF) (+)VO$_4$$^{3-}$]) cells pooled from three biological replicates. Scale bars, 20 μm. **b**, Representative FLIM-FRET images (left) and quantification of apparent FRET efficiency (right) in MM231 cells with stable Illusia expression and constitutive overexpression of PTPN11 (Shp2, WT) or a phosphatase-dead mutant (Shp2, Mut; PTPN11(D425A, C459S)) (n = 97 (mScarlet), 96 (WT) and 96 (Mut) cells pooled from four biological replicates; significance assessed using a one-way ANOVA with a Tukey correction for multiple comparisons; NS, not significant, ***p < 0.001). Scale bars, 20 μm. **c**, Representative western blot (left) and densitometry (right) of MM231 cells with stable Illusia expression and constitutive overexpression of PTPN11 (Shp2, WT) or a phosphatase-dead mutant (Shp2, Mut; PTPN11(D425A, C459S)); n = 8 biological replicates; significance assessed using a one-sample two-tailed *t*-test against the normalised control value; *p < 0.05, ***p < 0.001; NS, not significant). **d-e**, Representative FLIM-FRET images (**d**) and quantification of apparent FRET efficiency (**e**) of MM231 cells with stable Illusia expression treated overnight with Dox to induce overexpression of PTP-PEST WT or a phosphatase-dead mutant (Mut, PTPN12(D199A, C231S) (n = 100 cells in each condition pooled from four biological replicates; significance assessed using a one-way ANOVA with a Šidák correction for multiple comparisons; NS, not significant, ***p < 0.001). Scale bars, 20 μm. **f**, Western blot (left) and densitometry (right) from parallel data in (**d**; n = 4 biological replicates; significance assessed using a one-sample two-tailed *t*-test against the normalised control value; *p < 0.05; NS, not significant). **g** & **h**, Representative FLIM-FRET images (left) and quantification of apparent FRET efficiency (right) of MM231 (**g**) and TIF (**h**) cells with stable Illusia expression and treated with SHP099 for 2 h (100 nM) (MM231, n = 96 (DMSO) and 95 (SHP099) | TIFs, n = 100 (DMSO) and 99 (SHP099) cells pooled from four biological replicates; significance assessed using an unpaired two-tailed Student's *t*-test with a Welch's correction (NS, not significant, ***p < 0.001). Scale bars, 20 μm. **i**, Representative western blots (left) and densitometry (right) of MM231 and TIF cells with stable Illusia expression and treated with SHP099 for 2 h (100 nM; n = 6 biological replicates; significance assessed using a one-sample two-tailed *t*-test against the normalised control value; *p < 0.05). Data are mean ± SEM. Source data and exact p-values are provided in the statistical source data file. Boxplots represent median and interquartile range. Whiskers extend to min and max values. Grey areas on boxplots highlight the interquartile range of the control conditions.

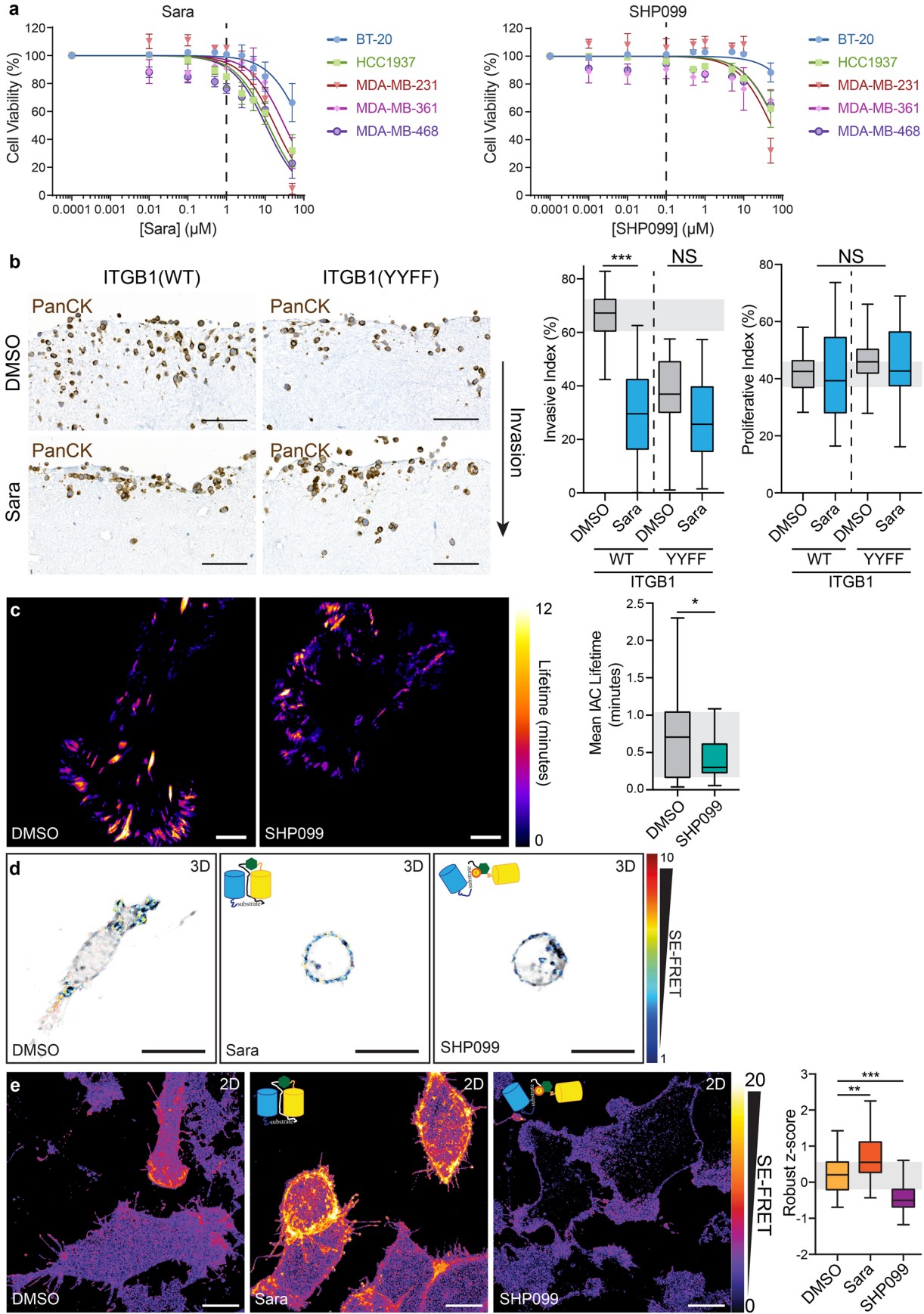

**Extended Data Fig. 6 | See next page for caption.**

**Extended Data Fig. 6 | Inhibition of Src or Shp2 reduces breast cancer invasion and IAC lifetime. a**, IC50 curves for human HCC1937, MM231, MDA-MB-361, MM468 and BT-20 cells treated with increasing concentrations of either Sara or SHP099 (n = 3 biological replicates; 6 wells analysed per treatment per cell line for each replicate). Dotted lines indicate concentrations used for later experiments in the article. Data are presented as mean values +/- SEM. **b**, Representative images (left) of MM231 ITGB1(WT or YYFF) cell invasion into 3D fibroblast-contracted collagen I in the presence of DMSO or Sara (1 µM), stained with pan-cytokeratin (PanCK) to exclude fibroblasts from the analysis. Quantification (right) of invasion beyond 100 µm, normalised to the total number of cells/region, or proliferation, normalising the number of Ki67-positive nuclei to the total number of cells/region (ITGB1(WT), n = 32 (DMSO) and 32 (Sara) | ITGB1(YYFF), n = 32 (DMSO) and 30 (Sara) regions pooled from four biological replicates; significance assessed using an unpaired two-tailed Student's *t*-test with Welch's correction; NS, not significant, ***p < 0.001). Scale bars, 100 µm. **c**, MM231 cells stably expressing paxillin-EGFP imaged on OMX TIRF for 30 min in the presence of SHP099 (100 nM) or DMSO control (n = 26 (DMSO) and 21 (SHP099) cells pooled from three biological replicates; significance assessed using a one-way ANOVA with a Tukey correction for multiple comparisons; *p < 0.05). **d**, Representative images of Illusia-expressing MM231 cells embedded in 3D collagen matrices and treated with DMSO or either SHP099 (100 nM) or Sara (1 µM; n = 3 biological replicates; 3-8 movies/replicate/condition). Scale bars, 20 µm. **e**, Illusia-expressing MM231 cells treated overnight with Sara (1 µM) or SHP099 (100 nM) and imaged for changes in FRET by sensitised emission (SE-FRET) on a confocal microscope (n = 30 (DMSO), 29 (Sara) and 27 (SHP099) fields of view pooled from three biological replicates; significance assessed using a one-way ANOVA with a Dunnett's correction for multiple comparisons; **p < 0.01, not significant, ***p < 0.001). Scale bars, 10 µm. Source data and exact p-values are provided in the statistical source data file. Boxplots represent median and interquartile range. Whiskers extend to min and max values. Grey areas on boxplots highlight the interquartile range of the control conditions.

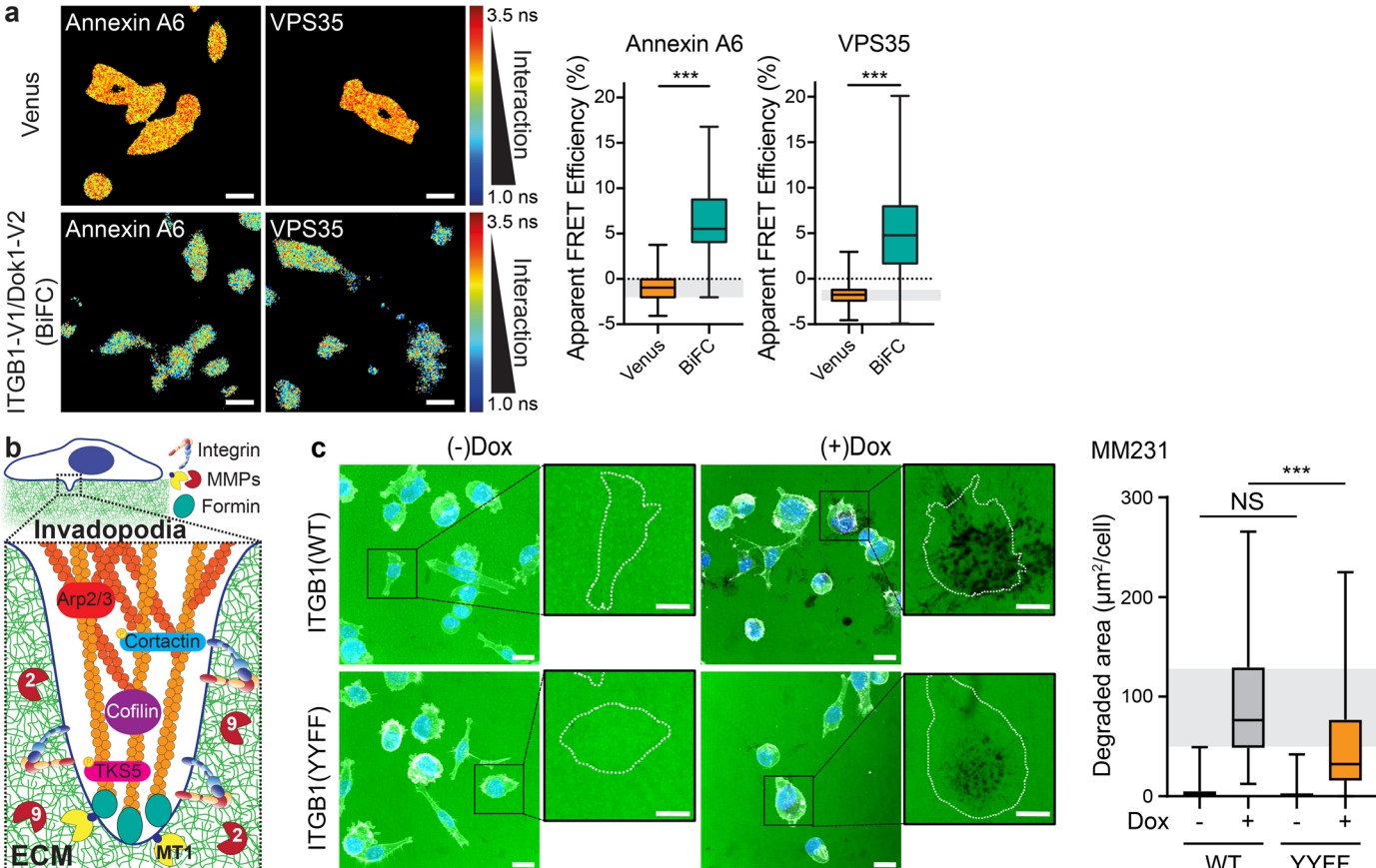

**Extended Data Fig. 7 | Cofilin, VPS35 and Annexin A6 are recruited to the phosphorylated Dok1/ITGB1 complex. a**, Representative BiFC-FLIM-FRET images (left) and quantification of apparent FRET efficiency (right) from MM231 cells transfected with Venus or ITGB1-V1/ Dok1-V2 and either mScarlet-tagged Annexin A6 or VPS35 (n = 61 for each Annexin A6 condition, 58 (VPS35, Venus) and 60 (VPS35, BiFC) cells pooled from three biological replicates; significance assessed using an unpaired two-tailed Student's *t*-test with a Welch's correction; ***p < 0.001). Scale bars, 20 μm. **b**, Schematic of an invadopodium degrading the ECM. **c**, Representative images (left) and quantification (right) of MM231 ITGB1(WT or YYFF) cells with doxycycline(Dox)-inducible Src(E378G), treated

(+/-)Dox overnight and then seeded on fluorescent gelatin (green) for 6 h (white, SiR-Actin stain; blue, DAPI nuclear stain; n = 30 [ITGB1 (WT) (-)Dox], 41 [ITGB1 (WT) ( + )Dox], 34 [ITGB1(YYFF) (-)Dox] and 41 [ITGB1(YYFF) ( + )Dox] fields of view pooled from three biological replicates; significance assessed using a one-way ANOVA with a Šidák correction for multiple comparisons; NS, not significant, ***p < 0.001). Scale bars, 20 μm. Source data and exact p-values are provided in the statistical source data file. Boxplots represent median and interquartile range. Whiskers extend to min and max values. Grey areas on boxplots highlight the interquartile range of the control conditions.

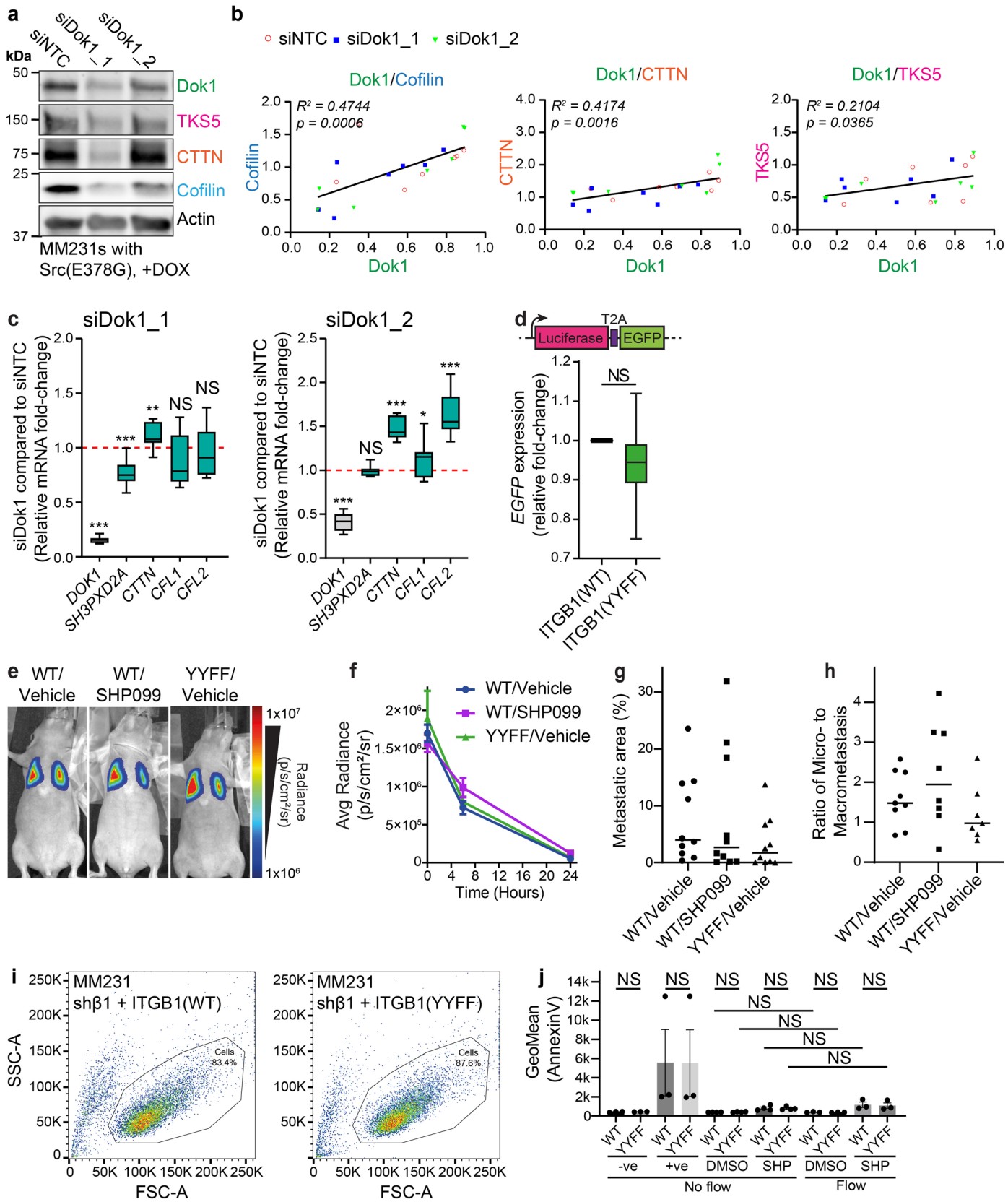

Extended Data Fig. 8 | See next page for caption.

**Extended Data Fig. 8 | Expression of invadopodia components is co-regulated at the protein level. a**, Representative immunoblot from MM231 ITGB1 (WT or YYFF) cells and doxycycline (Dox)-inducible Src(E378G) transfected with siRNAs against Dok1, treated with Dox for 24 h and blotted for Cofilin, TKS5, CTTN and Dok1. **b**, X-Y correlation plots from densitometric analysis of western blots in (**a**), normalising to the actin loading control for Cofilin, TKS5, CTTN and Dok1. Plotted are graphs showing correlations between Dok1-Cofilin (left), Dok1-TKS5 (middle) and Dok1-CTTN (right; n = 7 biological replicates; significance assessed using an X-Y linear regression). **c**, qPCR data from MM231 cells transfected with siRNAs against *DOK1* (siDok1_1 and siDok1_2), or a NTC (siNTC), and treated with Dox for 24 h before processing for qRT-PCR. Plotted is the relative mRNA fold-change, normalised to *GAPDH* (n = 4 biological replicates, performed in duplicate or triplicate; significance assessed using a one sample *t*-test against the normalised siNTC control values; NS, not significant, $p < 0.001$, $p < 0.01$, $p < 0.05$). **d**, Schematic of luciferase/EGFP construct (top) and qRT-PCR (bottom) for *EGFP* fold-change between the MM231 ITGB1(WT or YYFF) cells stably-expressing the luciferase/EGFP construct, normalised to *GAPDH* (n = 4 biological replicates, performed in triplicate; significance assessed using a one-sample two-tailed *t*-test against the normalised control values; NS, not significant). **e**, Representative images of mice immediately after lateral tail vein injection with MM231 ITGB1(WT or YYFF) cells stably-expressing the luciferase/EGFP construct and pre-treated with either DMSO or SHP099 (100 nM). Oral gavage of Vehicle or

SHP099 (100 mg/kg) proceeded for 5 days from the day of injection. **f**, Quantification of the average (avg) radiance from the luciferase signal during the colonisation stages of the MM231 cells (n = 9 mice tracked/group; mean ± SEM). **g**, Quantification of metastatic area (%) in lungs from EGFP-positive MM231 cells (n = 10 mice/group; scatter plot with a line at the median value). **h**, Assessment of the ratio of micro- to macrometastasis in lungs from clusters of EGFP-positive MM231 cells lesser or greater than 3,000 μm² respectively; excluding mice where no micro- or macrometastasis were detected (n = 9, 8 & 7 mice for the WT/vehicle, WT/SHP099 and YYFF/vehicle groups respectively; scatter plot with a line at the median value). **i** & **j**, Gating strategy (**i**) and flow cytometric analysis (**j**) of annexinV-stained MM231s with ITGB1(WT or YYFF) and treated with either DMSO or SHP099 (SHP; 100 nM) while grown overnight in suspension on ultra-low attachment plates and either subjected to flow and/or treated with SHP099 (100 nM) or DMSO control (negative (-ve): unstained negative control samples; positive ( + ve): MM231s treated with a cell-death-inducing cocktail of doxorubicin (10 μg/ml), $VO_4^{3-}$ (50 μM), gemcitabine (10 μM); n = 4 biological replicates/cell line for DMSO and SHP099 "No flow" and n = 3 biological replicates for all other conditions; mean ± SEM; Statistics from a one-way ANOVA with a Tukey correction for multiple comparisons; NS, not significant). Source data and exact p-values are provided in the statistical source data file. Boxplots represent median and interquartile range. Whiskers extend to min and max values.

# Reporting Summary

## Statistics

For all statistical analyses, confirm that the following items are present in the figure legend, table legend, main text, or Methods section.

| n/a | Confirmed | |
|---|---|---|
| ☐ | ☒ | The exact sample size (*n*) for each experimental group/condition, given as a discrete number and unit of measurement |
| ☐ | ☒ | A statement on whether measurements were taken from distinct samples or whether the same sample was measured repeatedly |
| ☐ | ☒ | The statistical test(s) used AND whether they are one- or two-sided<br>*Only common tests should be described solely by name; describe more complex techniques in the Methods section.* |
| ☒ | ☐ | A description of all covariates tested |
| ☐ | ☒ | A description of any assumptions or corrections, such as tests of normality and adjustment for multiple comparisons |
| ☐ | ☒ | A full description of the statistical parameters including central tendency (e.g. means) or other basic estimates (e.g. regression coefficient) AND variation (e.g. standard deviation) or associated estimates of uncertainty (e.g. confidence intervals) |
| ☐ | ☒ | For null hypothesis testing, the test statistic (e.g. *F*, *t*, *r*) with confidence intervals, effect sizes, degrees of freedom and *P* value noted<br>*Give P values as exact values whenever suitable.* |
| ☒ | ☐ | For Bayesian analysis, information on the choice of priors and Markov chain Monte Carlo settings |
| ☒ | ☐ | For hierarchical and complex designs, identification of the appropriate level for tests and full reporting of outcomes |
| ☐ | ☒ | Estimates of effect sizes (e.g. Cohen's *d*, Pearson's *r*), indicating how they were calculated |

*Our web collection on statistics for biologists contains articles on many of the points above.*

## Software and code

Policy information about availability of computer code

| | |
|---|---|
| Data collection | Commercially available microscopes and mass spectrometers and their corresponding software were used to collect data and are indicated in the appropriate methods sections. |
| Data analysis | Label-free quantification of mass spectrometry data was performed in Proteome Discoverer 2.5 (ThermoFisher); Flow cytometry data were analysed in FlowJo (v.10.8.2, BD Biosciences); QuPath, an open source software for Bioimage Analysis (https://qupath.github.io/) was used to quantify invasion and Ki-67 staining of xenografts and in collagen matrices; Fiji (NIH), an open-source platform for biological-image analysis, was used to analyze densitometry data (western blots) and microscopy images. The specific plug-ins used for each analysis are indicated in the methods and include the FRET and Colocalization analyser and Coloc2. Statistical analysis was performed in Graphpad Prism v.7 or Rstudio (v2022.12.10; robust z-scores calculated using stats package (v3.6.2)) as indicated in the methods. |

For manuscripts utilizing custom algorithms or software that are central to the research but not yet described in published literature, software must be made available to editors and reviewers. We strongly encourage code deposition in a community repository (e.g. GitHub). See the Nature Portfolio guidelines for submitting code & software for further information.

# Data

Policy information about availability of data

All manuscripts must include a data availability statement. This statement should provide the following information, where applicable:
- Accession codes, unique identifiers, or web links for publicly available datasets
- A description of any restrictions on data availability
- For clinical datasets or third party data, please ensure that the statement adheres to our policy

> All data are available in the main text or the supplementary materials. Additionally, the mass spectrometry proteomics data have been deposited to the ProteomeXchange Consortium via the PRIDE partner repository with the dataset identifier PXD048646. Plasmids generated in this study are available through Addgene, as indicated in Supplementary Table 5, or by contacting one of the corresponding authors.

# Research involving human participants, their data, or biological material

Policy information about studies with human participants or human data. See also policy information about sex, gender (identity/presentation), and sexual orientation and race, ethnicity and racism.

| | |
|---|---|
| Reporting on sex and gender | N/A |
| Reporting on race, ethnicity, or other socially relevant groupings | N/A |
| Population characteristics | N/A |
| Recruitment | N/A |
| Ethics oversight | N/A |

Note that full information on the approval of the study protocol must also be provided in the manuscript.

# Field-specific reporting

Please select the one below that is the best fit for your research. If you are not sure, read the appropriate sections before making your selection.

☒ Life sciences        ☐ Behavioural & social sciences        ☐ Ecological, evolutionary & environmental sciences

For a reference copy of the document with all sections, see nature.com/documents/nr-reporting-summary-flat.pdf

# Life sciences study design

All studies must disclose on these points even when the disclosure is negative.

| | |
|---|---|
| Sample size | No sample size calculations were performed. Samples sizes were chosen based on previous experiments in the field to ensure that statistical testing was both robust and reliable. For example, for microscopy data, n=10-50 cells/replicate from three or more biological replicates followed the success of our own previous work (PMID: 37844244). This was similarly the case for the animal experiments, where the number of mice for the local invasion assessment and lung colonization experiments were performed as previously (PMID: 37436978; PMID: 27336951). Sample sizes are indicated in figure legends. |
| Data exclusions | Outlier removal for the SE-FRET screen (Figure 4a) was performed using GraphPad Prism 7, ROUT (Q = 0.1%). |
| Replication | Experiments were repeated at least three times (number of replicates are indicated in the figure legends). The only exception being the qRT-PCR for the FRET screen, performed once with three technical replicates (Supplementary Figure 4c). All attempts at replication were successful. |
| Randomization | Animals were randomly assigned to cages (equal number of animals per cage) by animal facility staff. Cages were chosen at random for experimentation. All animals were maintained under the same condition and were at the same developmental stage. |
| Blinding | Experiments were not performed in a blinded fashion. Analysis softwares/statistical packages were used as detailed in the methods for robust data analysis, removing user bias. In addition, appropriate controls were included in experiments and control versus treated samples were analysed in the same fashion. |

# Behavioural & social sciences study design

All studies must disclose on these points even when the disclosure is negative.

| | |
|---|---|
| Study description | Briefly describe the study type including whether data are quantitative, qualitative, or mixed-methods (e.g. qualitative cross-sectional, quantitative experimental, mixed-methods case study). |
| Research sample | State the research sample (e.g. Harvard university undergraduates, villagers in rural India) and provide relevant demographic information (e.g. age, sex) and indicate whether the sample is representative. Provide a rationale for the study sample chosen. For studies involving existing datasets, please describe the dataset and source. |
| Sampling strategy | Describe the sampling procedure (e.g. random, snowball, stratified, convenience). Describe the statistical methods that were used to predetermine sample size OR if no sample-size calculation was performed, describe how sample sizes were chosen and provide a rationale for why these sample sizes are sufficient. For qualitative data, please indicate whether data saturation was considered, and what criteria were used to decide that no further sampling was needed. |
| Data collection | Provide details about the data collection procedure, including the instruments or devices used to record the data (e.g. pen and paper, computer, eye tracker, video or audio equipment) whether anyone was present besides the participant(s) and the researcher, and whether the researcher was blind to experimental condition and/or the study hypothesis during data collection. |
| Timing | Indicate the start and stop dates of data collection. If there is a gap between collection periods, state the dates for each sample cohort. |
| Data exclusions | If no data were excluded from the analyses, state so OR if data were excluded, provide the exact number of exclusions and the rationale behind them, indicating whether exclusion criteria were pre-established. |
| Non-participation | State how many participants dropped out/declined participation and the reason(s) given OR provide response rate OR state that no participants dropped out/declined participation. |
| Randomization | If participants were not allocated into experimental groups, state so OR describe how participants were allocated to groups, and if allocation was not random, describe how covariates were controlled. |

# Ecological, evolutionary & environmental sciences study design

All studies must disclose on these points even when the disclosure is negative.

| | |
|---|---|
| Study description | Briefly describe the study. For quantitative data include treatment factors and interactions, design structure (e.g. factorial, nested, hierarchical), nature and number of experimental units and replicates. |
| Research sample | Describe the research sample (e.g. a group of tagged Passer domesticus, all Stenocereus thurberi within Organ Pipe Cactus National Monument), and provide a rationale for the sample choice. When relevant, describe the organism taxa, source, sex, age range and any manipulations. State what population the sample is meant to represent when applicable. For studies involving existing datasets, describe the data and its source. |
| Sampling strategy | Note the sampling procedure. Describe the statistical methods that were used to predetermine sample size OR if no sample-size calculation was performed, describe how sample sizes were chosen and provide a rationale for why these sample sizes are sufficient. |
| Data collection | Describe the data collection procedure, including who recorded the data and how. |
| Timing and spatial scale | Indicate the start and stop dates of data collection, noting the frequency and periodicity of sampling and providing a rationale for these choices. If there is a gap between collection periods, state the dates for each sample cohort. Specify the spatial scale from which the data are taken |
| Data exclusions | If no data were excluded from the analyses, state so OR if data were excluded, describe the exclusions and the rationale behind them, indicating whether exclusion criteria were pre-established. |
| Reproducibility | Describe the measures taken to verify the reproducibility of experimental findings. For each experiment, note whether any attempts to repeat the experiment failed OR state that all attempts to repeat the experiment were successful. |
| Randomization | Describe how samples/organisms/participants were allocated into groups. If allocation was not random, describe how covariates were controlled. If this is not relevant to your study, explain why. |
| Blinding | Describe the extent of blinding used during data acquisition and analysis. If blinding was not possible, describe why OR explain why blinding was not relevant to your study. |

Did the study involve field work? ☐ Yes ☐ No

## Field work, collection and transport

| | |
|---|---|
| Field conditions | *Describe the study conditions for field work, providing relevant parameters (e.g. temperature, rainfall).* |
| Location | *State the location of the sampling or experiment, providing relevant parameters (e.g. latitude and longitude, elevation, water depth).* |
| Access & import/export | *Describe the efforts you have made to access habitats and to collect and import/export your samples in a responsible manner and in compliance with local, national and international laws, noting any permits that were obtained (give the name of the issuing authority, the date of issue, and any identifying information).* |
| Disturbance | *Describe any disturbance caused by the study and how it was minimized.* |

# Reporting for specific materials, systems and methods

We require information from authors about some types of materials, experimental systems and methods used in many studies. Here, indicate whether each material, system or method listed is relevant to your study. If you are not sure if a list item applies to your research, read the appropriate section before selecting a response.

### Materials & experimental systems

| n/a | Involved in the study |
|---|---|
| ☐ | ☒ Antibodies |
| ☐ | ☒ Eukaryotic cell lines |
| ☒ | ☐ Palaeontology and archaeology |
| ☐ | ☒ Animals and other organisms |
| ☒ | ☐ Clinical data |
| ☒ | ☐ Dual use research of concern |
| ☒ | ☐ Plants |

### Methods

| n/a | Involved in the study |
|---|---|
| ☒ | ☐ ChIP-seq |
| ☐ | ☒ Flow cytometry |
| ☒ | ☐ MRI-based neuroimaging |

## Antibodies

| | |
|---|---|
| Antibodies used | All antibodies are described in the methods and are commercially available. These include: primary antibodies against Ki67 (Dako, 905 M7240, 1:500), EGFP (Invitrogen, A-11122, 1:800), pan-cytokeratin (Invitrogen, 906 MA5-13203, 1:25), integrin β1 (clone P5D2, in-house production from DSHB hybridoma), inactive β1 (clone 836 mAb13, in-house production from DSHB hybridoma) and β3 (MCA728, AbD Serotech), ITGB1 (1:1,000, Abcam, ab52971), ITGB1(Y783) (1:500, Abcam, ab62337), PTP-PEST (1:1,000, Cell Signalling, 14735), Shp2 (1:1,000, Cell Signalling, 3397), Src (1:1,000, Cell Signalling, 2108), Src(Y416) (1:500, 2101, Cell Signalling), Arg (1:1,000, ab134134, Abcam), p130Cas (1:1000, sc-20029, Santa Cruz), p130Cas(Y165) (1:1,000, 4015, Cell Signalling), p130Cas(Y410) (1:1000, 4011, Cell Signalling), pY (1:1,000, 610000, BD Biosciences), GFP (1:1,000, A11122, ThermoFisher), Dok1 (1:1,000, ab8112, Abcam), VPS35 (ab10099, Abcam, Goat, 1:1,000), Cofilin (D3F9; 1:1,000, Rabbit, 5175, Cell Signalling), Tks5 (1:500, Millipore, MABT336), Cortactin (CTTN; clone 4F11; 1:500, Millipore, 05-180), Annexin A6 (1:500, Abcam, ab31026), HRP)-conjugated anti-mouse secondary (1:2,000, Cell Signaling Technology, 7076) and GAPDH (1:10,000; Hytest, 5G4MAB6C5) |
| Validation | Antibodies used in this study have been validated by the manufacturer in the specified application (western blotting, IF, IHC, flow) and cross-reactivity with other species indicated in the data sheet. Antibodies were also used in prior studies. |

## Eukaryotic cell lines

Policy information about cell lines and Sex and Gender in Research

| | |
|---|---|
| Cell line source(s) | The cell lines used in the manuscript were TIFs (a kind gift from J.C. Norman (Beatson Institute, Glasgow, Scotland, UK))69, HEK293FT (ThermoFisher, R70007), HEK293T (ATCC, CRL-3216), MDA-MB-231 (ATCC, HTB-26), MDA-MB-468 (ATCC, HTB-132), HCC1937 (ATCC, CRL-2336), BT-20 (ATCC, HTB-19), MDA-MB-361 (ATCC, HTB-27) and MCF10A (ATCC, CRL-10317). |
| Authentication | The MDA-MB-231s were authenticated by STR profiling using the services of the Leibniz Institute DSMZ. Other cell lines were not authenticated separately by the authors. |
| Mycoplasma contamination | All cell lines were tested and confirmed negative for mycoplasma on a regular basis. |
| Commonly misidentified lines (See ICLAC register) | BT-20 cells purchased from ATCC were used in a panel of breast cancer cell lines that were available in the lab. This panel was used to test the effect on proliferation against a range of breast cancer cell lines to Src or Shp2 inhibition. |

# Palaeontology and Archaeology

| | |
|---|---|
| Specimen provenance | *Provide provenance information for specimens and describe permits that were obtained for the work (including the name of the issuing authority, the date of issue, and any identifying information). Permits should encompass collection and, where applicable, export.* |
| Specimen deposition | *Indicate where the specimens have been deposited to permit free access by other researchers.* |
| Dating methods | *If new dates are provided, describe how they were obtained (e.g. collection, storage, sample pretreatment and measurement), where they were obtained (i.e. lab name), the calibration program and the protocol for quality assurance OR state that no new dates are provided.* |

☐ Tick this box to confirm that the raw and calibrated dates are available in the paper or in Supplementary Information.

| | |
|---|---|
| Ethics oversight | *Identify the organization(s) that approved or provided guidance on the study protocol, OR state that no ethical approval or guidance was required and explain why not.* |

Note that full information on the approval of the study protocol must also be provided in the manuscript.

# Animals and other research organisms

Policy information about studies involving animals; ARRIVE guidelines recommended for reporting animal research, and Sex and Gender in Research

| | |
|---|---|
| Laboratory animals | Female, Hsd:AthymicNude-Foxn1nu; 7-8 wks, min 20 gram at the start of the experiment; Mice were housed in standard conditions (12-hour light/dark cycle; ambient temperature, 21 degrees celsius; 50% humidity ± 8%) with food and water available ad libitum. |
| Wild animals | No wild animals were used in the study |
| Reporting on sex | Given the context of breast cancer, female mice were used in all animal experiments |
| Field-collected samples | No field collected samples were used in the study |
| Ethics oversight | All animal experiments were performed in accordance with The Finnish Act on Animal Experimentation (Animal licence number ESAVI/12558/2021). |

Note that full information on the approval of the study protocol must also be provided in the manuscript.

# Clinical data

Policy information about clinical studies
All manuscripts should comply with the ICMJE guidelines for publication of clinical research and a completed CONSORT checklist must be included with all submissions.

| | |
|---|---|
| Clinical trial registration | *Provide the trial registration number from ClinicalTrials.gov or an equivalent agency.* |
| Study protocol | *Note where the full trial protocol can be accessed OR if not available, explain why.* |
| Data collection | *Describe the settings and locales of data collection, noting the time periods of recruitment and data collection.* |
| Outcomes | *Describe how you pre-defined primary and secondary outcome measures and how you assessed these measures.* |

# Dual use research of concern

Policy information about dual use research of concern

## Hazards

Could the accidental, deliberate or reckless misuse of agents or technologies generated in the work, or the application of information presented in the manuscript, pose a threat to:

No | Yes

☐ ☐ Public health

☐ ☐ National security

☐ ☐ Crops and/or livestock

☐ ☐ Ecosystems

☐ ☐ Any other significant area

## Experiments of concern

Does the work involve any of these experiments of concern:

No | Yes

☐ ☐ Demonstrate how to render a vaccine ineffective

☐ ☐ Confer resistance to therapeutically useful antibiotics or antiviral agents

☐ ☐ Enhance the virulence of a pathogen or render a nonpathogen virulent

☐ ☐ Increase transmissibility of a pathogen

☐ ☐ Alter the host range of a pathogen

☐ ☐ Enable evasion of diagnostic/detection modalities

☐ ☐ Enable the weaponization of a biological agent or toxin

☐ ☐ Any other potentially harmful combination of experiments and agents

# Plants

| Seed stocks | N/A |
|---|---|
| Novel plant genotypes | N/A |
| Authentication | N/A |

# ChIP-seq

## Data deposition

☐ Confirm that both raw and final processed data have been deposited in a public database such as GEO.

☐ Confirm that you have deposited or provided access to graph files (e.g. BED files) for the called peaks.

Data access links
*May remain private before publication.*
*For "Initial submission" or "Revised version" documents, provide reviewer access links. For your "Final submission" document, provide a link to the deposited data.*

Files in database submission
*Provide a list of all files available in the database submission.*

Genome browser session
(e.g. UCSC)
*Provide a link to an anonymized genome browser session for "Initial submission" and "Revised version" documents only, to enable peer review. Write "no longer applicable" for "Final submission" documents.*

## Methodology

Replicates
*Describe the experimental replicates, specifying number, type and replicate agreement.*

Sequencing depth
*Describe the sequencing depth for each experiment, providing the total number of reads, uniquely mapped reads, length of reads and whether they were paired- or single-end.*

Antibodies
*Describe the antibodies used for the ChIP-seq experiments; as applicable, provide supplier name, catalog number, clone name, and lot number.*

Peak calling parameters
*Specify the command line program and parameters used for read mapping and peak calling, including the ChIP, control and index files used.*

| Data quality | *Describe the methods used to ensure data quality in full detail, including how many peaks are at FDR 5% and above 5-fold enrichment.* |
|---|---|
| Software | *Describe the software used to collect and analyze the ChIP-seq data. For custom code that has been deposited into a community repository, provide accession details.* |

# Flow Cytometry

## Plots

Confirm that:

☒ The axis labels state the marker and fluorochrome used (e.g. CD4-FITC).

☒ The axis scales are clearly visible. Include numbers along axes only for bottom left plot of group (a 'group' is an analysis of identical markers).

☒ All plots are contour plots with outliers or pseudocolor plots.

☒ A numerical value for number of cells or percentage (with statistics) is provided.

## Methodology

| Sample preparation | Flow cytometry: MM231 ITGB1(WT or YYFF) cell lines were trypsinised, fixed in 2% PFA for 10 min at room temperature, and washed with PBS before being stained with primary antibodies to assess surface integrin levels (1:100 dilution in Tyrode's buffer [β1 (clone P5D2, in-house production from DSHB hybridoma), inactive β1 (clone mAb13, in-house production from DSHB hybridoma) and β3 (MCA728, AbD Serotech)]) for 1 h at 4oC with gentle agitation. Samples were then washed 2x with PBS and incubated with secondary antibodies (ALEXA-488 conjugated Anti-Mouse/Anti-Rat Invitrogen; 1:300 dilution in Tyrode's buffer) for a 1 h at 4oC with gentle agitation. Cells were then washed again with PBS before being resuspended in 200 µl of PBS and loaded into a 96-well plate for analysis. Cell sorting: FACS was used in the generation of stable cell lines. Cells expressing fluorescent constructs were trypsinised, washed and resuspended in PBS + 1% FBS prior to sorting + pen/strep. Fluorescent cells were then isolated within a narrow fluorescence range, collected in FBS and cultured for further experiments in complete medium. |
|---|---|
| Instrument | LSRFortessa cell analyzer using the High Throughput Sampler (BD Biosciences). Up to 10,000 single cell events were collected per condition. |
| Software | Gating and statistical analysis of the cell population were performed in FlowJo (version 10.8.2; BD Biosciences). |
| Cell population abundance | Positive cells from the FACS were expanded post sorting for further experiments. |
| Gating strategy | The gating for flow cytometry was performed in FlowJo (FSC and SSC dot plots) to exclude debris and dead cells from analyses. |

☒ Tick this box to confirm that a figure exemplifying the gating strategy is provided in the Supplementary Information.

# Magnetic resonance imaging

## Experimental design

| Design type | *Indicate task or resting state; event-related or block design.* |
|---|---|
| Design specifications | *Specify the number of blocks, trials or experimental units per session and/or subject, and specify the length of each trial or block (if trials are blocked) and interval between trials.* |
| Behavioral performance measures | *State number and/or type of variables recorded (e.g. correct button press, response time) and what statistics were used to establish that the subjects were performing the task as expected (e.g. mean, range, and/or standard deviation across subjects).* |

## Acquisition

| Imaging type(s) | *Specify: functional, structural, diffusion, perfusion.* |
|---|---|
| Field strength | *Specify in Tesla* |
| Sequence & imaging parameters | *Specify the pulse sequence type (gradient echo, spin echo, etc.), imaging type (EPI, spiral, etc.), field of view, matrix size, slice thickness, orientation and TE/TR/flip angle.* |
| Area of acquisition | *State whether a whole brain scan was used OR define the area of acquisition, describing how the region was determined.* |

Diffusion MRI ☐ Used ☐ Not used

## Preprocessing

**Preprocessing software**
*Provide detail on software version and revision number and on specific parameters (model/functions, brain extraction, segmentation, smoothing kernel size, etc.).*

**Normalization**
*If data were normalized/standardized, describe the approach(es): specify linear or non-linear and define image types used for transformation OR indicate that data were not normalized and explain rationale for lack of normalization.*

**Normalization template**
*Describe the template used for normalization/transformation, specifying subject space or group standardized space (e.g. original Talairach, MNI305, ICBM152) OR indicate that the data were not normalized.*

**Noise and artifact removal**
*Describe your procedure(s) for artifact and structured noise removal, specifying motion parameters, tissue signals and physiological signals (heart rate, respiration).*

**Volume censoring**
*Define your software and/or method and criteria for volume censoring, and state the extent of such censoring.*

## Statistical modeling & inference

**Model type and settings**
*Specify type (mass univariate, multivariate, RSA, predictive, etc.) and describe essential details of the model at the first and second levels (e.g. fixed, random or mixed effects; drift or auto-correlation).*

**Effect(s) tested**
*Define precise effect in terms of the task or stimulus conditions instead of psychological concepts and indicate whether ANOVA or factorial designs were used.*

**Specify type of analysis:** ☐ Whole brain ☐ ROI-based ☐ Both

**Statistic type for inference**
*Specify voxel-wise or cluster-wise and report all relevant parameters for cluster-wise methods.*

(See Eklund et al. 2016)

**Correction**
*Describe the type of correction and how it is obtained for multiple comparisons (e.g. FWE, FDR, permutation or Monte Carlo).*

## Models & analysis

| n/a | Involved in the study |
|-----|----------------------|
| ☐ | ☐ Functional and/or effective connectivity |
| ☐ | ☐ Graph analysis |
| ☐ | ☐ Multivariate modeling or predictive analysis |

**Functional and/or effective connectivity**
*Report the measures of dependence used and the model details (e.g. Pearson correlation, partial correlation, mutual information).*

**Graph analysis**
*Report the dependent variable and connectivity measure, specifying weighted graph or binarized graph, subject- or group-level, and the global and/or node summaries used (e.g. clustering coefficient, efficiency, etc.).*

**Multivariate modeling and predictive analysis**
*Specify independent variables, features extraction and dimension reduction, model, training and evaluation metrics.*

