## [Peer Review File · Nature Cell Biology]

Dynamic regulation of integrin $\beta 1$ phosphorylation supports invasive breast cancer progression

Corresponding Author: Professor Johanna Ivaska

Version 0:

Decision Letter:

*Please delete the link to your author homepage if you wish to forward this email to co-authors.

Dear Professor Ivaska,

Your manuscript, "Dynamic regulation of integrin $\beta 1$ phosphorylation supports invasive breast cancer progression", has now been seen by 3 referees, who are experts in phosphatases in cell biology (referee 1); biosensors (referee 2); integrins and ECM (referee 3). As you will see from their comments (attached below) they find this work of potential interest, but have raised substantial concerns, which in our view would need to be addressed with considerable revisions before we can consider publication in Nature Cell Biology.

Nature Cell Biology editors discuss the referee reports in detail within the editorial team, including the chief editor, to identify key referee points that should be addressed with priority, and requests that are overruled as being beyond the scope of the current study. To guide the scope of the revisions, I have listed these points below. We are committed to providing a fair and constructive peer-review process, so please feel free to contact me if you would like to discuss any of the referee comments further.

I should stress that the referees' concerns point to potential unclear characterization of integrin beta1 phosphorylation and unclear FRET based biosensor, which would need to be addressed with experiments and data, and reconsideration of the study for this journal and re-engagement of referees would depend on strength of these revisions.

In particular, it would be essential to:

A.) Further characterize integrin beta 1 phosphorylation, localization, and interactions (all Reviewers) with additional experimental data

B.) Experimentally assess the YYFF mutation and FRET based biosensor (all Reviewers)

C.) Further measure effects on metastatic potential through measuring circulating tumour cell load, as well as quantify metastasis and metastatic areas (Reviewer #3).

D.) All other referee concerns pertaining to strengthening existing data, providing controls, methodological details, clarifications and textual changes, should also be addressed.

E.) Finally please pay close attention to our guidelines on statistical and methodological reporting (listed below) as failure to do so may delay the reconsideration of the revised manuscript. In particular please provide:

We would be happy to consider a revised manuscript that would satisfactorily address these points, unless a similar paper is published elsewhere, or is accepted for publication in Nature Cell Biology in the meantime.

- ensure that it conforms to our format instructions and publication policies (see below and www.nature.com/nature/authors/).

- provide a point-by-point rebuttal to the full referee reports verbatim, as provided at the end of this letter.

- provide the completed Editorial Policy Checklist (found here <https://www.nature.com/authors/policies/Policy.pdf>), and Reporting Summary (found here <https://www.nature.com/authors/policies/ReportingSummary.pdf>).

This is essential for reconsideration of the manuscript and these documents will be available to editors and referees in the event of peer review. For more information see <http://www.nature.com/authors/policies/availability.html> or contact me.

Nature Cell Biology is committed to improving transparency in authorship. As part of our efforts in this direction, we are now requesting that all authors identified as 'corresponding author' on published papers create and link their Open Researcher and Contributor Identifier (ORCID) with their account on the Manuscript Tracking System (MTS), prior to acceptance. ORCID helps the scientific community achieve unambiguous attribution of all scholarly contributions. You can create and link your ORCID from the home page of the MTS by clicking on 'Modify my Springer Nature account'. For more information please visit <http://www.springernature.com/orcid>.

Link Redacted

We would like to receive a revised submission within six months. We would be happy to consider a revision even after this timeframe, however if the resubmission deadline is missed and the paper is eventually published, the submission date will be the date when the revised manuscript was received.

We hope that you will find our referees' comments, and editorial guidance helpful. Please do not hesitate to contact me if there is anything you would like to discuss.

Best wishes,

Daryl

Daryl Jason Verzosa David, PhD

Senior Editor, Nature Cell Biology
Nature Portfolio
Advisory Editor, npj Biological Physics and Mechanics

Heidelberger Platz 3, 14197 Berlin, Germany
Email: daryl.david@nature.com
ORCID: <https://orcid.org/0000-0002-9253-4805>

Reviewers' Comments:

Reviewer #1:

Remarks to the Author:

Key results

It is known that two tyrosine phosphorylation sites in the integrin $\beta 1$ (ITGB1) tail are key for promoting migration and other cell behaviours through differential recruitment of key effectors. This study investigates the regulation of ITGB1 phosphorylation and its impact on invasion and metastasis. Using a novel FRET reporter based on Dok1 recruitment to the phosphorylated integrin tail, Arg (Abl2) and Src kinases are identified as regulators of phosphorylation of site Y783. Inhibition of these kinases phenocopies an integrin YYFF (non-phosphorylatable) mutant, which shows impaired invasion. They next investigate the counteracting phosphatase using an siRNA screen of the Dok1 recruitment FRET reporter. Of the phosphatases that decrease the FRET signal (i.e. promote phosphorylation and Dok1 binding), they focus on PTP-PEST (PTPN12) and SHP2 (PTPN11), since these are components of the integrin adhesome. The authors suggest that ITGB1 is a direct substrate of SHP2 (followed up on most due to availability of an inhibitor). Surprisingly, both SHP2 and Src inhibition lead to impaired cell invasion, which is initially counterintuitive. This leads the authors to propose that it is phosphorylation dynamics, and not just phosphorylation, that mediate invasion. There is then a characterisation of the Dok1 interactome, which identified cofilin. Cofilin and Dok1 are shown to bind to tks5 and cortactin via FRET reporters, to promote invasion. In a model of metastatic colonisation of the lung, SHP2 inhibition leads to a phenotype similar to the expression of ITGB1 YYFF.

Validity

Overall, the quality of the data is excellent and in large part the interpretation and conclusions are valid. There are a few things that limit the manuscript in terms of reaching definitive conclusions.

First, the YYFF is a great tool for determining the impact of losing integrin phosphorylation. In the end, the conclusion of the manuscript is that impairing phosphorylation dynamics, in either direction, inhibits invasion. In an ideal world, the authors would use an equivalent dephosphorylation defective construct to illustrate this. Furthermore, there are times when a comparison with the YYFF mutant would be useful but is absent. SHP2 as a regulator of invasion is not novel (e.g. 10.2147/OTT.S138833, therefore the link to integrin dephosphorylation is key.

Second, the claim that integrin is a direct substrate for SHP2 and PTP PEST is not fully substantiated. Many PTPs have effects on kinases that could also explain the results. For example, SHP2 has been identified as a FAK phosphatase. There are also other pathways regulated by phosphatases that contribute to invasion.

Third, the most striking result is the similarity of phenotype of a phosphatase and a kinase. I'd like to see more insight into what the

authors think could explain this or how it might work, since a lot of the manuscript focuses on Dok1 (i.e. integrin phosphorylation-dependent) mediated invasion. It is also worth noting that despite similar effects on integrin phosphorylation, PTP-PEST and SHP2 are reported to have opposing effects on invasion (e.g. <https://doi.org/10.1158/0008-5472.CAN-18-0085> and 10.2147/OTT.S138833). In the screening data (Supplementary Table S1), the 3 siRNAs targeting PTPN11 (SHP2) give 3 different results (based on robust z score), therefore the follow up is critical to the conclusion that it is dynamics and not phosphorylation driving invasion.

Fourth, using FRET, the authors identify an interaction between Dok1 and cortactin and Tks5, however, neither were identified in the interactome of Dok1 determined by BiCAP. This is not addressed in the text.

Significance

The manuscript sets out to gain insights into phosphorylation dynamics using elegant imaging approaches. This is a tour de force in cellular interaction studies, and the live cell reporter for phosphorylation events in the cell is really exciting. Although the importance of dynamic regulation of phosphorylation in shaping signalling outcomes is not new (see Gelens and Saurin, *Dev Cell* 2018), the work highlights it in a new and highly dynamic, disease relevant system. Furthermore, the screen of phosphatases highlights the complexity of the upstream regulation of just a single phosphosite and yields many areas for future investigation. In addition, the mechanism by which Dok1 promotes invasion is further clarified by identifying its interactome when bound to integrin. The combination of approaches will be an invaluable addition to the tool kits and knowledge of the phosphorylation signalling and adhesion fields. Finally, the data are also highly significant for the cancer field; understanding invasion processes is key for understanding metastasis, and SHP2 inhibition is shown to prevent metastatic colonisation in the lung. Therefore, SHP2 inhibitors should be considered for prevention of metastasis (intervention is not investigated) as well as inhibition of MAPK signalling.

Suggested improvements to address concerns raised above

1. The ITGB1 YYFF mutant provides compelling evidence that phosphorylation at these sites is essential for invasion. However, an equivalent tool to demonstrate that dephosphorylation is equally essential is missing. I fully acknowledge that this may not be possible to address since glutamates (i.e. a YYEE construct) do not always recapitulate tyrosine phosphorylation. However, the authors have a clean way to evaluate this using their Dok1-integrin FRET reporter (perhaps they already have?). The only other alternatives I can think of are use of a synthetic amino acid, which I think is well beyond the scope of the manuscript. Or, use of constitutively active Src as shown in Figure 2c (although this is somewhat confounded by other functions of Src).

a. Is YYEE a feasible way to “block” dephosphorylation?

b. If not, or in addition, what is the consequence of expressing constitutively active Src for invasion?

c. Does YYFF cause cells to round up like SHP2 and Src inhibitors when imaged in 3D collagen? (like Supplementary video 2). This would further cement the commonality of the phenotypes. Since there is a good chance that Src and SHP2 have functions that impact invasion beyond their effects on ITGB1.

2. The phosphatase screen identified SHP2 and PTP-PEST as negative regulators of Integrin phosphorylation (as read out by Dok1 binding to phosphorylated integrin). The current data do not fully support that the two phosphatases directly dephosphorylate integrin.

a. Figure 5b: Could the FRET reporters reveal indirect phosphorylation-dependent interactions? (i.e. between SHP2 and ITGB1)

b. Add PTP PEST FLIM-FRET image to Figure 5b.

c. The GFP trap pulldowns using wt and “substrate trapping” mutants of each phosphatase show greater binding of integrin to the mutant. However, it looks as though if this were quantified the binding might actually be equal since the wildtype phosphatases are significantly less well expressed. This appears to be a tag issue since untagged SHP2 is not sensitive to mutational status (Supp figure 5c). To increase the confidence in these assays, find a tag that shows similar expression levels. In addition, one would expect that phosphorylated integrin would bind to the trapping mutant and any bound to the wildtype would be dephosphorylated. (i.e. blot for pY783). This would rule out an association mediated by the SHP2 SH2 domains, which is another possible explanation for the data in Figure 5b. Similarly, a co-IP with WT and YYFF ITGB1 could address the phosphorylation dependence of the “trapping”.

d. Since vanadate is used to “block” the association of SHP2 and integrin by FRET (Figure 5B), this could also be used to compete ITGB1 off in the pull downs (Figure 5C).

e. Use a direct dephosphorylation assay. Overexpression in cells can lead to indirect effects. This could be done using a similar approach taken to demonstrate Src phosphorylation (figure 2e) of ITGB1 tails. Or take an approach using alkylated lysates to carry out dephosphorylation with recombinant protein and western blot for the Y783 phosphosite, or enrich tyrosine phosphorylated proteins with phosphotyrosine IPs.

f. Does double knockdown of SHP2 and PTP-PEST show an additive effect on ITGB1 phosphorylation?

Minor comments related to dephosphorylation:

g. Supplementary figures 5c and e both require quantification since the lanes where the active phosphatases are present seem to be underloaded which could explain the lower levels of ITGB1 phosphorylation.

h. What is the % input for fig. 5c? This is important since the SHP2 mutant enriches ITGB1 to higher levels than the input. Is this consistent with the stoichiometry of phosphorylated ITGB1 in these cells? I.e. if 50% of ITGB1 is pulled out – is it feasible that 50% of ITGB1 was phosphorylated in the cell lysates?

i. Check molecular weights in Figure 5c. SHP2 should be ~70kDa, so adding 20+ kDa via clover should increase its MW above the 75 kDa marker (I acknowledge MW ladders are not perfect – but it looks a bit odd). I could not see a method for the GFP trap experiment, please include.

3. The dynamic phosphorylation of ITGB1 is used as a potential explanation for the similar phenotypes of inhibiting SHP2 (a phosphatase) and Src (a kinase).

a. Line 428. The title of this paragraph undermines the point made using the Shp2 inhibitor. Consider adding “dynamics” since one would expect ITGB1 to be phosphorylated upon SHP2 inhibition (according to the Illusia reporter data), yet this impairs invasion.

b. I would like to read more about how the authors think this concept could work, the discussion is very brief. For example, it could be that Src and SHP2 elicit a “flickering” effect on ITGB1 phosphorylation, an effect at the single molecule scale. Alternatively, one could imagine that this occurs on a population scale, perhaps even setting up gradients of phosphorylation, where either Src or SHP2 predominates (but what acts upstream of them?). The video S2 appears to show front-rear differences in phosphorylation (via FRET reporter) on a whole cell scale. And then at the leading edge there are regional differences. Both SHP2 and Src inhibition lead to an effective loss of polarity (rounding up). Is this cause or consequence? On one hand SHP2 and Src could have other impacts on polarization and this is indirect.

Alternatively, the inability to establish ITGB1 phosphorylation differentials may impair polarity. This could be addressed as suggested in point 1c using ITGB1 YYFF and/or YVEE in 3D culture.

c. Does Src inhibition have the same effect as SHP099 and ITGB1 YYFF on tumour colonization? (Figure 8e) Why was it not included?

Minor points:

1. Define ITGB1 in introduction (mostly integrin β 1 is used, but occasionally ITGB1 is used. E.g. line 80 vs line 88).
2. Include quantification of Western blots e.g. Supp Figures 2i, 3e and f, 4e and f, 5c and e
3. Supp figure 5 b – no image for mScarlet control (control images are otherwise present)
4. Supp Figure 7c. The study implicates SHP2 in invadopodia formation. Could it be incorporated into this diagram?
5. Supplementary video 2. Switch order of Sara and SHP099 in description to correspond to order in video. Could also include time frames in video legend for ease of interpretation.
6. Typos in Figure 8f YYYFF and Supp video 2: Illusia
7. Figure 7e. Loss of Dok1 has a greater impact than ITGB1 YYFF. Are there other integrins that bind to Dok1 that could explain this? (i.e. Dok1 is downstream and therefore a core component for invasion mediated by multiple integrins)
8. Include an acknowledgement in the text that cortactin and Tks5 were not identified in the BICAP experiment. This could well reflect the different approaches used.

Clarity

The text is very clear and accessible to a broad audience. There are some areas where previous knowledge, especially on phosphatases, is not provided. The functions of both Src and SHP2 in focal adhesions have been described and this should be acknowledged as it clearly has a bearing on the conclusions drawn here. The flow is logical, but I have the feeling that this study initially set out to characterise the phosphorylated integrin-Dok1 complex and then sought to understand its regulation. The initially counterintuitive role of the phosphatases requires further elaboration, since it is contradictory to the pro-invasion role of the phosphorylated complex. The authors are attempting to address a challenging question with innovative and clearly described tools. I have added a few references below that could add to the discussion.

References

The references are appropriate. The authors might consider adding references related to SHP2 in focal adhesions and polarity (e.g. 10.26508/lsa.202201557 and 10.1158/1541-7786.MCR-12-0578). Interestingly, it was recently shown that ITGB1 and FRS2 (mentioned in discussion) knockdown sensitise cells to Shp2 inhibition (<https://doi.org/10.1158/0008-5472.CAN-23-1127>). As mentioned above, dynamic phosphorylation regulation has been discussed previously and might help shape the discussion on the topic: <https://doi.org/10.1016/j.devcel.2018.03.002>

Reviewer #2:

Remarks to the Author:

Summary:

Authors report the regulatory mechanisms governing integrin phosphorylation and their efforts to understand the role of integrin phosphorylation in the progression of cancer. They started with cell line experiments that using β 1 integrins with non-phosphorylatable residues at the cytoplasmic domain (YYFF) and showed reduction in invasion through the collagen matrix. This observation suggests the importance of β 1 integrin phosphorylation at the NPxY motifs. Then they developed a FRET-based biosensor, termed Illusia, that utilizes fluorescence proteins, a portion of β 1 cytoplasmic tail (aa772-798), and Dok1 protein that preferentially binds to a phosphorylated tyrosine, and uses it as a proxy of the phosphorylation status of the integrin β 1 tail. Authors showed that the integrin β 1 phosphorylation is related to cancer cell motility and identified tyrosine phosphatases and regulators for β 1 phosphorylation. The manuscript is informative and identifying direct involvement of other components in the integrin/Dok1 complex using FRET sensors is exciting. A suitably revised version of the manuscript will be a good candidate for Nature Cell Biology. Please consider the following questions and comments.

Major comments:

1. Authors reported the downregulated invasion index of the YYFF mutant using 3D collagen matrix invasion assay in fig.1. Their claim, I believe, that this invasion phenotype difference with the mutations is responsible for cancer cell invasion. However, this may be a general motility difference that is not specific to cancer cells. So they need to test if the YYFF effect is cancer cell specific, and if normal cells do the same, perhaps the invasion index is just the cell motility index in the 3D collagen matrix?
2. To my understanding, Illusia reflects the phosphorylation activities that can be targeted to the tail motif (i.e. kinase/phosphatase activity for the freely diffusing β 1 tail-like substrate), rather than directly reporting integrin β 1 phosphorylation. The sensor may not properly reflect the status of 'active' integrins connected to cytoskeletal structure via adaptor proteins. Please make this clear and discuss the limitations and validity of the Illusia.
3. It is unclear if Illusia requires phosphorylation of the both NPxY motifs within the sensor β 1 tail region for FRET change. What could be the FRET change or lifetime change when only one of them is phosphorylated? Another question is, can Illusia also bind to a phosphorylated integrin in cell membrane.
4. Illusia uses the phosphotyrosine binding domain of Dok1. How does overexpressed Illusia, localized on the membrane, affect the Dok1 binding proteins (Cofilin, CTTN, and TKS5 in fig.8)?
5. The structure of unphosphorylated Illusia is not clear (in main text and figure). Why does it have the well-folded structure with high FRET? What is the expected distance between donor and acceptor in the unphosphorylated and phosphorylated states?
6. It would be nice if 'FRET efficiency (%)' is better defined in the main text and figure. In FRET efficiency box plots, the y-axis may be 'apparent FRET efficiency' defined in method ($E = (1 - TDA/TD) \times 100$). If so, please make that explicit. Is a single data point in the box plots the apparent FRET efficiency from each pixel (the average of molecules in a diffraction limit) or the average value of a single cell?

7. Authors used the siRNA library to identify specific PTPs responsible for $\beta 1$ dephosphorylation and found PTPN11 and PTPN12. Also, 57 of them showed an unexpected increase of FRET in Illusia. It would be beneficial if authors check any compensatory effects of other genes (upregulation or downregulation). Although it might be out of the scope for the current work to elucidate the indirect role of the phosphatase in upregulating integrin $\beta 1$ phosphorylation, authors can provide expression level of a few phosphatases using western blot to make sure their expression level is similar to WT cells.

8. Co-IP results in Fig. 5C are not clear on how GFP Trap was done and unclear why there are stronger bands for mutants and why only Shp2 CoIP showed anti-GFP band. Why was this not done in MM231 cells.

9. I am not sure if 'dynamic regulation' in the title clearly describes the ambivalent effect of phosphorylation and dephosphorylation shown in this manuscript. What's a little confusing about the data is that it is not the absolute level of phosphorylation but the ability to change phosphorylation state that is important, and perhaps that is what they mean by dynamic regulation?

Minor comments:

1. It is up to the author to decide, but we recommend using 'fluorescence resonance energy transfer' instead of 'Förster resonance energy transfer' honoring Theodor Förster who was a longtime member of the Nazi party and chaired a department in occupied Poland. Even though he contributed to the understanding of FRET, we can name the phenomena without honoring someone associated with inhumane atrocities.

2. Fig 1d quantification shows the significance symbol (*), but it is not described. According to the main text the p-value would be higher than >0.05 . Please specify p-value and let readers interpret the data and its statistical significance. Using the symbol without the description will mislead readers. In Fig. 5C caption, PTPN11 and PTPN12 would need to change to Shp2 and PTP-PEGT, respectively for consistency.

3. I cannot find the main text describing the data of Fig. 3d.

4. Author may use one of VO3- or Vanadate (Fig. 3e) for consistency of figures.

5. 'NTC' should be defined in the figure legend. Please check all acronyms.

6. Both 'FRET' and 'Phosphorylation' are used for the guide of lifetime color code randomly. Please be consistent.

Reviewer #3:

Remarks to the Author:

Overview: The manuscript 'Dynamic regulation of integrin $\beta 1$ phosphorylation supports invasive breast cancer progression' focuses on the mechanism and implications of integrin phosphorylation at the NPxY (783/795) sites using Förster resonance energy transfer and proteomics approaches.

It is well recognized that integrins play an important role in various extracellular matrix (ECM) dependent cellular functions such as adhesion formation, migration, invasion, rigidity sensing, and mechanotransduction. However, the control and modulation of these various functions by integrins is not fully understood. In this manuscript, the authors investigate the phosphorylation of integrin $\beta 1$ at Y783 and Y795 as a functional requirement for invadopodia formation, and its potential role in cancer cell invasion and metastasis. The delicate balance of this phosphorylation state mediated by kinases and phosphatases explored in this manuscript helps to clarify one of the intrinsic mechanisms by which a cell can modulate invadopodia formation and invasion to regulate cancer metastasis through phosphorylation of key integrins that control tumor cell adhesion activity.

The role of proteins such as integrin $\beta 1$, Dok1 and Cortactin in invadopodia formation has been extensively studied and previously reported. Nevertheless, how these proteins are recruited to the invadopodia complex remains an open question. The authors present comprehensive cell biological studies to show that phosphorylated integrin $\beta 1$ can serve as a base for the recruitment of proteins involved in invadopodia formation. Mechanistically the authors determined that the regulation of integrin $\beta 1$ phosphorylation is mediated through a balanced combination of the activity of Src and Arg kinases, and PTP-PEST and Shp2 phosphatases. Details were presented demonstrating how inhibiting either these phosphatases or kinases regulates integrin phosphorylation, integrin-Dok1 interactions, cancer cell invasion, and invadopodia formation. Quantification and biochemical evidence for these adhesion interactions was presented using a novel FRET sensor for integrin phosphorylation- Illusia that was coupled with FLIM imaging, which was inspired by previous kinase FRET sensors. The authors also used a variety of additional FRET pairs, pulldown assays and western blots to verify key interactions in this phosphorylated integrin-Dok1 interactome. These cell biology studies overall were well executed and comprehensive although some critical details and controls are recommended, as outlined below.

However, despite the compelling in vitro studies and intriguing evidence for a kinase/phosphatase regulated phosphorylation adhesion complex regulating tumor cell invadopodia, evidence in support of the impact of this interactome on tumor cell phenotype and specifically metastasis in vivo is rudimentary. To begin with unfortunately the authors failed to conduct critical analysis of key metrics that would argue for any impact on tumor cell dissemination in vivo, and the in vivo models used for their experiments are less than ideal. It remains unclear which step during metastasis they maintain is critically modulated by integrin phosphorylation. Metastasis is a multi step process that depends upon efficient extravasation into the circulation, intravasation into the metastatic site and most importantly survival and expansion at the metastatic site to form a viable metastatic lesion. To be clear, tumor cell invasion into the parenchyma does not equate with metastasis and merely defines what is considered to be a malignant lesion (i.e. all malignant lesions by definition invade into the associated stroma but not all malignant lesions in fact generate metastatic tumors). It remains unclear what if anything informative the subcutaneous tumor xenograft studies presented in the first part of this manuscript reveal that wasn't already shown by the data presented in the in vitro collagen studies which clearly demonstrate an impact on tumor cell invasion. At the very least the authors should have quantified circulating tumor cells. Such a metric would have made a good case if there were quantifiable differences for an impact on potential tumor metastasis. In addition, the rationale for the tail vein studies remains ill-defined as this manipulation essentially monitors for metastatic outgrowth in the lungs and to a smaller extent to intravasation into the metastatic tissue, which are different criteria to that expounded upon in the earlier tumor in vivo studies and argued by the invadopodia studies. It is also somewhat disappointing that

the authors confined the in vivo metastasis studies only to subcutaneous xenografts and immunocompromised mice, especially given the large body of impressive data that has accumulated regarding the role of the immune system in cancer metastasis and specifically to dissemination and metastatic outgrowth. Finally, and importantly, while the cell biology work is well executed, and the authors should be commended, the conceptual advance remains modest so that while the results are certainly interesting, the novelty is not evident. A number of clarifications and questions need to be addressed by the authors.

Major concerns:

1. Reconsideration of murine models to assess metastasis:

The authors conducted subcutaneous injection studies into immune-compromised mice and quantified invasion into the surrounding parenchyma as an indication of impact on tumor cell invasion in vivo. Subcutaneous injection does not reconstitute the stromal microenvironment of tumors (unless studying skin cancer) with much fidelity so it is unclear how this manipulation would reflect impact on tumor invasion in the mammary gland. It is recommended that the authors consider conducting fat pad injection studies which would better assess impact on tumor cell invasion into the breast tissue stroma. Moreover, as the data are currently presented it remains unclear what new information is obtained by merely measuring invasion in these tissues that was not already demonstrated using the collagen matrices in vitro? A malignant tumor is an invasive tumor! There is no correlation between extent of invasion of a primary tumor and metastatic potential. Tumor size yes! But tumor invasion that describes whether it is a benign tumor or a malignant tumor. A better metric to measure impact on metastatic potential would be measuring circulating tumor cell load. This is easily done and at the very least should be done. Furthermore, while not essential, clearly the use of immune competent mouse models is a much better system to use to assess impact on metastasis rather than these human lines into immune compromised mice.

Regarding the tail vein injection in Fig. 8e, the rationale for performing this experiment is confusing. There is no convincing indication as to whether the integrin YYFF mutation actually impact tumor cell metastasis and if this is mediated via its effects on invadopodia formation. The tail vein injection studies measure the ability of the tumor cells to survive and grow in the lungs of the mice and to a lesser extent some level of intravasation. The impact on invadopodia could impact intravasation - however these assays over load the system with a bolus of cells and essentially swamp the lungs with cells. Even nonmalignant cells will get into the lungs with this type of manipulation. The difference is that nonmalignant cells won't grow to form viable metastatic lesions. What would be very constructive is if the authors were to quantify micro and macro metastasis as well as metastatic area. This analysis would reveal whether or not the cells are competent to gain access (intravasate) into the lungs but fail to grow out (micro mets) or are able to gain access and grow out (macro mets). Metastatic area would also be reflected in this analysis. In addition, conducting these studies in an immunocompetent mouse would provide much needed evidence in support of the impact of this phosphorylated interactome on metastatic potential of tumor cells. It is very possible that the impact of integrin phosphorylation extends well beyond classic invadopodia formation!

2. It is recommended that the author improve the characterization of the shRNA mediated integrin beta1 knockdown:

The authors show an ~90% reduction in integrin beta1 expression with the shB1. However, no significant reduction in proliferation was seen, which is different from previous studies (Hou et al, Scientific Reports, 2016; Grzesiak et al, Cancer Therapy, 2011). The authors are advised to check proliferation and adhesion formation on different beta1 specific substrates such as Type 1 collagen, Fibronectin and Laminin to better understand the effect of beta1 knockdown and mutant on basic cell functions such as proliferation, adhesion formation, and 2D cell migration.

3. Spatial localization of the integrin beta1 YYFF:

Activated integrins have been shown to co-localize with talin, while the non-phosphorylatable mutant YYFF has a lower affinity towards talin. However, in Fig. S1f,g the authors show a high colocalization between integrin YYFF and activated integrin beta1. They also show a higher staining of YYFF at the adhesions (marked by paxillin). It is strongly recommended that the authors address this disparity. The authors should also quantify the Illusia FRET efficiency in vs outside IACs (using talin as a marker) to quantify the spatial localization of integrin phosphorylation events. The authors should also investigate if integrin phosphorylation correlates with invadopodia formation. The localization of integrin phosphorylation with cell migration- leading vs lagging edge should also be commented on.

4. External regulation of integrin phosphorylation:

The authors intensively investigate the internal factors affecting integrin phosphorylation. However, cell external factors also regulate integrin phosphorylation. Since stiffness affects invadopodia formation (Chang J. et al, Biophys J., 2020), does integrin phosphorylation take place primarily on softer surfaces? It would be useful to show if integrin phosphorylation anti-correlates with substrate stiffness. Changing stiffness of collagen matrix in Fig 1b, for example, can help show if the change in invasion is due to invadopodia formation. The effect of other chemotactic ligands which regulate invadopodia formation on integrin phosphorylation can also be looked at.

5. Non consistent FRET results:

In 'no treatment' conditions, FRET efficiency of YYFF should be lower than WT as shown in Fig. S2c. However, in Fig 3e and Fig. S3d, FRET efficiency of WT vs YYFF (without VO4) looks similar. The authors should address this disparity.

6. Role of integrin beta3 phosphorylation in invadopodia formation:

Role of integrin beta3 in invadopodia formation has been shown before (Peláez, R. et al., PLoS One, 2017; Feng Z. et al, PNAS, 2021). Since integrin beta3 is also a substrate for Src kinase and PTP-PEST phosphatase, phosphorylation of integrin beta3 might also be involved in invadopodia formation. It is recommended that the authors investigate the role of integrin beta3 in invadopodia formation.

7. Mechanism of integrin beta1 phosphorylation dependent invadopodia formation:

It is advised that the authors describe the mechanism of integrin beta1 phosphorylation-dependent invadopodia formation through recruitment of invadopodia proteins in the discussion. This is perhaps that main novelty presented in the manuscript and is unfortunately not as well developed as it could be. In this regard, the authors are in a good position to explain the previously unknown link between a 'switch on signal' to invadopodia formation, so that adding this link would greatly improve the impact of the work presented in the manuscript.

Minor concerns:

1. The authors should clarify the mutations in their first mention (line 78-79).
2. In Fig 1d legend: The authors should clearly mention what the dotted black line is, what the * means, 'n' and the statistical analysis used.
3. In Figure S2d, the authors use GFP-Talin head (F0-F3) as a control. However, Talin head does not specifically bind integrins, and has a PIP2 binding function as well (Chinthalapudi K. et al, 2018, PNAS). This would affect the FRET efficiency. Using full length talin and quantifying the FRET at IACs might serve as a better control because of its specific localization at the IACs.
4. In Fig 2f, the authors should add statistics.
5. The authors should add densitometry measurements for ITGB1(Y783)/ITGB1 western blots in Fig. S4e,f and Fig. S5e.
6. The images in Fig 8g do not look representative for YYFF since in the graph, the average # pulmonary nodules are roughly equal in SHP099 vs YYFF, while the images show a stark difference.
7. In lines 459-462, "Together, we propose a mechanism where spatially and temporally controlled integrin phosphorylation..." Spatial control of integrin phosphorylation was not achieved by the authors. The authors should explain/modify this statement.
8. In general, since the figures use a variety of FRET donor-acceptor pairs, it becomes difficult to understand the FRET efficiency graphs without the donor-acceptor pairs mentioned in the graphs. Including this will help increase the readability of the graphs.

Methods should be written concisely, but should contain all elements necessary to allow interpretation and replication of the results. As a guideline, Methods sections typically do not exceed 3,000 words. The Methods should be divided into subsections listing reagents and techniques. When citing previous methods, accurate references should be provided and any alterations should be noted. Information must be provided about: antibody dilutions, company names, catalogue numbers and clone numbers for monoclonal antibodies;

sequences of RNAi and cDNA probes/primers or company names and catalogue numbers if reagents are commercial; cell line names, sources and information on cell line identity and authentication. Animal studies and experiments involving human subjects must be reported in detail, identifying the committees approving the protocols. For studies involving human subjects/samples, a statement must be included confirming that informed consent was obtained. Statistical analyses and information on the reproducibility of experimental results should be provided in a section titled "Statistics and Reproducibility".

All Nature Cell Biology manuscripts submitted on or after March 21 2016 must include a Data availability statement at the end of the Methods section. For Springer Nature policies on data availability see <http://www.nature.com/authors/policies/availability.html>; for more information on this particular policy see <http://www.nature.com/authors/policies/data/data-availability-statements-data-citations.pdf>. The Data availability statement should include:

- Accession codes for primary datasets (generated during the study under consideration and designated as "primary accessions") and secondary datasets (published datasets reanalysed during the study under consideration, designated as "referenced accessions"). For primary accessions data should be made public to coincide with publication of the manuscript. A list of data types for which submission to community-endorsed public repositories is mandated (including sequence, structure, microarray, deep sequencing data) can be found here <http://www.nature.com/authors/policies/availability.html#data>.
- Unique identifiers (accession codes, DOIs or other unique persistent identifier) and hyperlinks for datasets deposited in an approved repository, but for which data deposition is not mandated (see here for details <http://www.nature.com/sdata/data-policies/repositories>).
- At a minimum, please include a statement confirming that all relevant data are available from the authors, and/or are included with the manuscript (e.g. as source data or supplementary information), listing which data are included (e.g. by figure panels and data types) and mentioning any restrictions on availability.
- If a dataset has a Digital Object Identifier (DOI) as its unique identifier, we strongly encourage including this in the Reference list and citing the dataset in the Methods.

We recommend that you upload the step-by-step protocols used in this manuscript to the Protocol Exchange. More details can found at www.nature.com/protocolexchange/about.

All imaging data should be accompanied by scale bars, which should be defined in the legend. Cropped images of gels/blots are acceptable, but need to be accompanied by size markers, and to retain visible background signal within the linear range (i.e. should not be saturated). The boundaries of panels with low background have to be demarked with black lines. Splicing of panels should only be considered if unavoidable, and must be clearly marked on the figure, and noted in the legend with a statement on whether the samples were obtained and processed simultaneously. Quantitative comparisons between samples on different gels/blots are discouraged; if this is unavoidable, it should only be performed for samples derived from the same experiment with gels/blots were processed in parallel, which needs to be stated in the legend.

The total number of Supplementary Figures (not including the "unprocessed scans" Supplementary Figure) should not exceed the number of main display items (figures and/or tables (see our Guide to Authors and March 2012 editorial <http://www.nature.com/ncb/authors/submit/index.html#suppinfo>; <http://www.nature.com/ncb/journal/v14/n3/index.html#ed>). No restrictions apply to Supplementary Tables or Videos, but we advise authors to be selective in including supplemental data.

GUIDELINES FOR EXPERIMENTAL AND STATISTICAL REPORTING

REPORTING REQUIREMENTS – To improve the quality of methods and statistics reporting in our papers we have recently revised the reporting checklist we introduced in 2013. We are now asking all life sciences authors to complete two items: an Editorial Policy Checklist (found here <https://www.nature.com/authors/policies/Policy.pdf>) that verifies compliance with all required editorial policies and a reporting summary (found here <https://www.nature.com/authors/policies/ReportingSummary.pdf>) that collects information on experimental design and reagents. These documents are available to referees to aid the evaluation of the manuscript. Please note that these forms are dynamic 'smart pdfs' and must therefore be downloaded and completed in Adobe Reader. We will then flatten them for ease of use by the reviewers. If you would like to reference the guidance text as you complete the template, please access these flattened versions at <http://www.nature.com/authors/policies/availability.html>.

STATISTICS – Wherever statistics have been derived the legend needs to provide the n number (i.e. the sample size used to derive statistics) as a precise value (not a range), and define what this value represents. Error bars need to be defined in the legends (e.g. SD, SEM) together with a measure of centre (e.g. mean, median). Box plots need to be defined in terms of minima, maxima, centre, and percentiles. Ranges are more appropriate than standard errors for small data sets. Wherever statistical significance has been derived, precise p values need to be provided and the statistical test used needs to be stated in the legend. Statistics such as error bars must not be derived from n<3. For sample sizes of n<5 please plot the individual data points rather than providing bar graphs. Deriving statistics from technical replicate samples, rather than biological replicates is strongly discouraged. Wherever statistical significance has been derived, precise p values need to be provided and the statistical test stated in the legend.

Version 1:

Decision Letter:

*Please delete the link to your author homepage if you wish to forward this email to co-authors.

Dear Professor Ivaska,

I am sorry once again for the delay. As previously mentioned, although Reviewer #3 was unable to assess your revisions, we have instead asked Reviewer #1's comments on your responses to Reviewer #3's previous concerns.

Your manuscript, "Dynamic regulation of integrin β 1 phosphorylation supports invasive breast cancer progression", has now been seen by 2 of our original referees, who are experts in phosphatases in cell biology (referee 1); biosensors (referee 2). As you will see from their comments (attached below) they find this work of interest, but have raised some important points. Although we are also very interested in this study, we believe that their concerns should be addressed before we can consider publication in Nature Cell Biology.

Nature Cell Biology editors discuss the referee reports in detail within the editorial team, including the chief editor, to identify key referee points that should be addressed with priority, and requests that are overruled as being beyond the scope of the current study. To guide the scope of the revisions, I have listed these points below. We are committed to providing a fair and constructive peer-review process, so please feel free to contact me if you would like to discuss any of the referee comments further.

In particular, it would be essential to:

A) In particular, please assess potential Integrin Tyr-phosphorylation with Western blots as per Reviewer #1.

B) All referee concerns pertaining to strengthening existing data, providing controls, methodological details, clarifications and textual changes, should also be addressed.

C) Finally please pay close attention to our guidelines on statistical and methodological reporting (listed below) as failure to do so may delay the reconsideration of the revised manuscript. In particular please provide:

- a Supplementary Figure including unprocessed images of all gels/blots in the form of a multi-page pdf file. Please ensure that blots/gels are labeled and the sections presented in the figures are clearly indicated. Please be sure to name the filename as "Source data – uncropped blots" or something similar.

- a Supplementary Table including all numerical source data in Excel format, with data for different figures provided as different sheets within a single Excel file. The file should include source data giving rise to graphical representations and statistical descriptions in the paper and for all instances where the figures present representative experiments of multiple independent repeats, the source data of all repeats should be provided. Please be sure to name this file/files as Source Data.

We therefore invite you to take these points into account when revising the manuscript. In addition, when preparing the revision please:

- ensure that it conforms to our format instructions and publication policies (see below and www.nature.com/nature/authors/),
- please ensure that any and all tables are provided as XLSX files; currently some of your supplementary tables are in PDF format.
- provide a point-by-point rebuttal to the full referee reports verbatim, as provided at the end of this letter.

- provide the completed Editorial Policy Checklist (found here <https://www.nature.com/authors/policies/Policy.pdf>), and Reporting Summary (found here <https://www.nature.com/authors/policies/ReportingSummary.pdf>). This is essential for reconsideration of the manuscript and these documents will be available to editors and referees in the event of peer review. For more information see <http://www.nature.com/authors/policies/availability.html> or contact me.

Nature Cell Biology is committed to improving transparency in authorship. As part of our efforts in this direction, we are now requesting that all authors identified as 'corresponding author' on published papers create and link their Open Researcher and Contributor Identifier (ORCID) with their account on the Manuscript Tracking System (MTS), prior to acceptance. ORCID helps the scientific community achieve unambiguous attribution of all scholarly contributions. You can create and link your ORCID from the home page of the MTS by clicking on 'Modify my Springer Nature account'. For more information please visit <http://www.springernature.com/orcid>.

Link Redacted

We would like to receive the revision within four weeks. If submitted within this time period, reconsideration of the revised manuscript will not be affected by related studies published elsewhere, or accepted for publication in Nature Cell Biology in the meantime. We would be happy to consider a revision even after this timeframe, but in that case we will consider the published literature at the time of resubmission when assessing the file.

We hope that you will find our referees' comments, and editorial guidance helpful. Please do not hesitate to contact me if there is anything you would like to discuss.

Best wishes,

Daryl

Daryl Jason Verzosa David, PhD

Senior Editor, Nature Cell Biology
Advisory Editor, npj Biological Physics and Mechanics
Nature Portfolio

Heidelberger Platz 3, 14197 Berlin, Germany
Email: daryl.david@nature.com
ORCID: <https://orcid.org/0000-0002-9253-4805>

Reviewers' Comments:

Reviewer #1 (Remarks to the Author):

The authors have gone to great lengths to address the comments of all reviewers.

My only comment is that I am surprised that the Illusia reporter indicates increased phosphorylation on softer substrates (new data Figure 6a), which is at odds with other reports (e.g. Src activation with stiffer substrates : such as in review: Forcing a growth factor response – tissue-stiffness modulation of integrin signaling and crosstalk with growth factor receptors (DOI: 10.1242/jcs.242461) and stiffer substrates correlate with increased tyrosine phosphorylation e.g. (DOI: 10.1073/pnas.94.25.13661). Additionally, in our hands, we also see global tyrosine phosphorylation increase on stiff matrices vs soft. This does not mean ITGB1 follows that trend, and there could be technical differences. Therefore, could the authors use the same methodology as other papers - i.e. bulk Western blot, to confirm their result with the ITGB1 phospho ab? I just really want to make sure their reporter is reporting what is claimed. Could the authors comment on this? Could it also reflect the localisation of the FRET sensor?

Reviewer #1's comments on authors' responses to Reviewer #3's previous concerns:

The authors have done a very good job in addressing the concerns of reviewer 3, including toning down some claims, clarifying figures and legends, and carrying out new experiments. The concerns around mouse models are beyond the scope of the study; the metastasis model used is standard in the field, and use of immunocompetent mice would complicate matters further, and make an entirely new story.

Reviewer #2 (Remarks to the Author):

I am satisfied with the authors' responses to my questions and comments.

GUIDELINES FOR SUBMISSION OF NATURE CELL BIOLOGY ARTICLES

ARTICLE FORMAT

ABSTRACT – should not exceed 150 words and should be unreferenced. This paragraph is the most visible part of the paper and should briefly outline the background and rationale for the work, and accurately summarize the main results and conclusions. Key genes, proteins and organisms should be specified to ensure discoverability of the paper in online searches.

TEXT – the main text consists of the Introduction, Results, and Discussion sections and must not exceed 3500 words including the abstract. The Introduction should expand on the background relating to the work. The Results should be divided in subsections with subheadings, and should provide a concise and accurate description of the experimental findings. The Discussion should expand on the findings and their implications. All relevant primary literature should be cited, in particular when discussing the background and specific findings.

REFERENCES – are limited to a total of 70 in the main text and Methods combined. They must be numbered sequentially as they appear in the main text, tables and figure legends and Methods and must follow the precise style of Nature Cell Biology references. References only cited in the Methods should be numbered consecutively following the last reference cited in the main text. References only associated with Supplementary Information (e.g. in supplementary legends) do not count toward the total reference limit and do not need to be cited in numerical continuity with references in the main text. Only published papers can be cited, and each publication cited should be included in the numbered reference list, which should include the manuscript titles. Footnotes are not permitted.

Methods should be written concisely, but should contain all elements necessary to allow interpretation and replication of the results. As a guideline, Methods sections typically do not exceed 3,000 words. The Methods should be divided into subsections listing reagents and techniques. When citing previous methods, accurate references should be provided and any alterations should be noted. Information must be provided about: antibody dilutions, company names, catalogue numbers and clone numbers for monoclonal antibodies; sequences of RNAi and cDNA probes/primers or company names and catalogue numbers if reagents are commercial; cell line names, sources and information on cell line identity and authentication. Animal studies and experiments involving human subjects must be reported in detail, identifying the committees approving the protocols. For studies involving human subjects/samples, a statement must be included confirming that informed consent was obtained. Statistical analyses and information on the reproducibility of experimental results should be provided in a section titled "Statistics and Reproducibility".

All Nature Cell Biology manuscripts submitted on or after March 21 2016, must include a Data availability statement as a separate section after Methods but before references, under the heading "Data Availability". For Springer Nature policies on data availability see <http://www.nature.com/authors/policies/availability.html>; for more information on this particular policy see <http://www.nature.com/authors/policies/data/data-availability-statements-data-citations.pdf>. The Data availability statement should include:

- Accession codes for primary datasets (generated during the study under consideration and designated as "primary accessions") and secondary datasets (published datasets reanalysed during the study under consideration, designated as "referenced accessions"). For primary accessions data should be made public to coincide with publication of the manuscript. A list of data types for which submission to community-endorsed public repositories is mandated (including sequence, structure, microarray, deep sequencing data) can be found here <http://www.nature.com/authors/policies/availability.html#data>.
- Unique identifiers (accession codes, DOIs or other unique persistent identifier) and hyperlinks for datasets deposited in an approved repository, but for which data deposition is not mandated (see here for details <http://www.nature.com/sdata/data-policies/repositories>).
- At a minimum, please include a statement confirming that all relevant data are available from the authors, and/or are included with the manuscript (e.g. as source data or supplementary information), listing which data are included (e.g. by figure panels and data types) and mentioning any restrictions on availability.
- If a dataset has a Digital Object Identifier (DOI) as its unique identifier, we strongly encourage including this in the Reference list and citing the dataset in the Methods.

We recommend that you upload the step-by-step protocols used in this manuscript to [protocols.io](http://www.protocols.io). More details can be found at <https://www.protocols.io/help/publish-articles>.

DISPLAY ITEMS – main display items are limited to 6-8 main figures and/or main tables. For Supplementary Information see below.

FIGURES – Colour figure publication costs \$395 per colour figure. All panels of a multi-panel figure must be logically connected and arranged as they would appear in the final version. Unnecessary figures and figure panels should be avoided (e.g. data presented in small tables could be stated briefly in the text instead).

All imaging data should be accompanied by scale bars, which should be defined in the legend.

Cropped images of gels/blots are acceptable, but need to be accompanied by size markers, and to retain visible background signal within the linear range (i.e. should not be saturated). The boundaries of panels with low background have to be demarked with black lines. Splicing of panels should only be considered if unavoidable, and must be clearly marked on the figure, and noted in the legend with a statement on whether the samples were obtained and processed simultaneously. Quantitative comparisons between samples on different gels/blots are discouraged; if this is unavoidable, it has to be performed for samples derived from the same experiment with gels/blots were processed in parallel, which needs to be stated in the legend.

Regardless of format, all figures must be vector graphic compatible files, not supplied in a flattened raster/bitmap graphics format, but should be fully editable, allowing us to highlight/copy/paste all text and move individual parts of the figures (i.e. arrows, lines, x and y axes, graphs, tick marks, scale bars etc). The only parts of the figure that should be in pixel raster/bitmap format are photographic images or 3D rendered graphics/complex technical illustrations.

Unprocessed scans of all key data generated through electrophoretic separation techniques need to be presented in a supplementary figure that should be labeled and numbered as the final supplementary figure, and should be mentioned in every relevant figure legend. This figure does not count towards the total number of figures and is the only figure that can be displayed over multiple pages, but should be provided as a single file, in PDF or TIFF format. Data in this figure can be displayed in a relatively informal style, but size markers and the figures panels corresponding to the presented data must be indicated.

The total number of Supplementary Figures (not including the "unprocessed scans" Supplementary Figure) should not exceed the number of main display items (figures and/or tables (see our Guide to Authors and March 2012 editorial <http://www.nature.com/ncb/authors/submit/index.html#suppinfo>; <http://www.nature.com/ncb/journal/v14/n3/index.html#ed>). No restrictions apply to Supplementary Tables or Videos, but we advise authors to be selective in including supplemental data.

Each Supplementary Figure should be provided as a single page and as an individual file in one of our accepted figure formats and should be presented according to our figure guidelines (see above). Supplementary Tables should be provided as individual Excel files. Supplementary Videos should be provided as .avi or .mov files up to 50 MB in size. Supplementary Figures, Tables and Videos must be

accompanied by a separate Word document including titles and legends.

GUIDELINES FOR EXPERIMENTAL AND STATISTICAL REPORTING

REPORTING REQUIREMENTS – To improve the quality of methods and statistics reporting in our papers we have recently revised the reporting checklist we introduced in 2013. We are now asking all life sciences authors to complete two items: an Editorial Policy Checklist (found here https://www.nature.com/authors/policies/Policy.pdf) that verifies compliance with all required editorial policies and a Reporting Summary (found here https://www.nature.com/authors/policies/ReportingSummary.pdf) that collects information on experimental design and reagents. These documents are available to referees to aid the evaluation of the manuscript. Please note that these forms are dynamic 'smart pdfs' and must therefore be downloaded and completed in Adobe Reader. We will then flatten them for ease of use by the reviewers. If you would like to reference the guidance text as you complete the template, please access these flattened versions at http://www.nature.com/authors/policies/availability.html.

Version 2:

Decision Letter:

Our ref: NCB-A53616B

10th January 2025

Dear Dr. Ivaska,

Thank you for submitting your revised manuscript "Dynamic regulation of integrin $\beta 1$ phosphorylation supports invasive breast cancer progression" (NCB-A53616B). It has now been seen by the original referees and their comments are below. The reviewers find that the paper has improved in revision, and therefore we'll be happy in principle to publish it in Nature Cell Biology, pending minor revisions to satisfy the referees' final requests and to comply with our editorial and formatting guidelines.

Thank you again for your interest in Nature Cell Biology Please do not hesitate to contact me if you have any questions.

Sincerely,
Daryl

Daryl Jason Verzosa David, PhD

Senior Editor, Nature Cell Biology
Advisory Editor, npj Biological Physics and Mechanics
Nature Portfolio

Heidelberger Platz 3, 14197 Berlin, Germany
Email: daryl.david@nature.com
ORCID: <https://orcid.org/0000-0002-9253-4805>

Version 3:

Decision Letter:

Dear Dr Ivaska,

I am pleased to inform you that your manuscript, "Dynamic regulation of integrin β 1 phosphorylation supports invasive breast cancer progression", has now been accepted for publication in *Nature Cell Biology*.

Over the next few weeks, your paper will be copyedited to ensure that it conforms to *Nature Cell Biology* style. Once your paper is typeset, you will receive an email with a link to choose the appropriate publishing options for your paper and our Author Services team will be in touch regarding any additional information that may be required.

Publication is conditional on the manuscript not being published elsewhere and on there being no announcement of this work to any media outlet until the online publication date in *Nature Cell Biology*.

Please note that *Nature Cell Biology* is a Transformative Journal (TJ). Authors may publish their research with us through the traditional subscription access route or make their paper immediately open access through payment of an article-processing charge (APC). Authors will not be required to make a final decision about access to their article until it has been accepted. [Find out more about Transformative Journals](https://www.springernature.com/gp/open-research/transformative-journals)

If you have not already done so, we strongly recommend that you upload the step-by-step protocols used in this manuscript to protocols.io (<https://protocols.io>), an open online resource that allows researchers to share their detailed experimental know-how. All uploaded protocols are made freely available and are assigned DOIs for ease of citation. Protocols and Nature Portfolio journal papers in which they are used can be linked to one another, and this link is clearly and prominently visible in the online versions of both. Authors who performed the specific experiments can act as primary authors for the Protocol as they will be best placed to share the methodology details, but the Corresponding Author of the present research paper should be included as one of the authors. By uploading your Protocols onto protocols.io, you are enabling researchers to more readily reproduce or adapt the methodology you use, as well as increasing the visibility of your protocols and papers. You can also establish a dedicated workspace to collect your Lab Protocols. Further

information can be found at <https://www.protocols.io/help/publish-articles>.

Nature Cell Biology encourages authors presenting evidence for cell, biological, molecular, and genetic interactions to consider communicating these findings using Biofactoid (<https://biofactoid.org/>). This tool helps users share a searchable representation of interactions (e.g. binding, gene expression, post-translational modification) between genes, gene products, or chemicals. Information added to Biofactoid, with author attribution, is shared on social media and public databases, such as Pathway Commons, where it can be discovered and analyzed in the context of a large and growing corpus of knowledge.

With kind regards,

Daryl

Daryl Jason Verzosa David, PhD

Senior Editor, Nature Cell Biology
Advisory Editor, npj Biological Physics and Mechanics
Nature Portfolio

Heidelberger Platz 3, 14197 Berlin, Germany
Email: daryl.david@nature.com
ORCID: <https://orcid.org/0000-0002-9253-4805>

** Visit the Springer Nature Editorial and Publishing website at http://editorial-jobs.springernature.com?utm_source=ejp_NCB_email&utm_medium=ejp_NCB_email&utm_campaign=ejp_NCB for more information about our career opportunities. If you have any questions please click [here](mailto:editorial.publishing.jobs@springernature.com).

Response to Reviewers' Comments:

Reviewer #1:

Remarks to the Author:

Key results

It is known that two tyrosine phosphorylation sites in the integrin $\beta 1$ (ITGB1) tail are key for promoting migration and other cell behaviours through differential recruitment of key effectors. This study investigates the regulation of ITGB1 phosphorylation and its impact on invasion and metastasis. Using a novel FRET reporter based on Dok1 recruitment to the phosphorylated integrin tail, Arg (Abl2) and Src kinases are identified as regulators of phosphorylation of site Y783. Inhibition of these kinases phenocopies an integrin YYFF (non-phosphorylatable) mutant, which shows impaired invasion. They next investigate the counteracting phosphatase using an siRNA screen of the Dok1 recruitment FRET reporter. Of the phosphatases that decrease the FRET signal (i.e. promote phosphorylation and Dok1 binding), they focus on PTP-PEST (PTPN12) and SHP2 (PTPN11), since these are components of the integrin adhesome. The authors suggest that ITGB1 is a direct substrate of SHP2 (followed up on most due to availability of an inhibitor). Surprisingly, both SHP2 and Src inhibition lead to impaired cell invasion, which is initially counterintuitive. This leads the authors to propose that it is phosphorylation dynamics, and not just phosphorylation, that mediate invasion. There is then a characterisation of the Dok1 interactome, which identified cofilin. Cofilin and Dok1 are shown to bind to tks5 and cortactin via FRET reporters, to promote invasion. In a model of metastatic colonisation of the lung, SHP2 inhibition leads to a phenotype similar to the expression of ITGB1 YYFF.

Validity

Overall, the quality of the data is excellent and in large part the interpretation and conclusions are valid. There are a few things that limit the manuscript in terms of reaching definitive conclusions.

Au: We thank the reviewer for the very positive assessment of our work and the expert suggestions that further helped us strengthen the study.

First, the YYFF is a great tool for determining the impact of losing integrin phosphorylation. In the end, the conclusion of the manuscript is that impairing phosphorylation dynamics, in either direction, inhibits invasion. In an ideal world, the authors would use an equivalent dephosphorylation defective construct to illustrate this. Furthermore, there are times when a comparison with the YYFF mutant would be useful but is absent. SHP2 as a regulator of invasion is not novel (e.g. 10.2147/OTT.S138833, therefore the link to integrin dephosphorylation is key.

Au: While we found that the YYFF mutant is a highly effective model system to validate the importance of dynamic integrin phosphorylation for efficient cancer cell invasion, we agree that it would be ideal to also block dephosphorylation. Working under the assumption that glutamic acid is not a suitable analog of phosphorylation, we employed systems where integrin phosphorylation would be upregulated through PTP inhibition or through overexpression of constitutively-active kinases. This allowed us to demonstrate that pushing the balance of integrin phosphorylation in either direction has consistent effects in disrupting cancer cell invasion, metastatic dissemination and invadopodia formation. However, we have now interrogated the feasibility of using a YEE mutant as part of our additional validation of Illusia, as well as expanding our assessment of integrin phosphorylation during

cancer invasion using our inducible-overexpression cell lines with constitutively-active Src (data included below).

We thank the reviewer for recognizing the novelty of our work. However, we respectfully disagree that Shp2 inhibition having such a clear effect on invasion *in vivo* and in 3D models is not novel. The paper highlighted by the reviewer (PMID: 28814887) shows a clear effect for Shp2 overexpression on proliferation, as has been described extensively by others, but assesses migration in 2D. So, while Shp2 has been hypothesised as having a role in invasion in several studies (PMID: 26088100), a direct assessment of Shp2 inhibition on invasion using a 3D ECM platform had yet to be made. In addition, the effect of Shp2 inhibition in reducing metastatic dissemination in mouse models of breast cancer (PMID: 33033382) was thought to occur through immunomodulation and decreased growth *in vivo*. Thus, we believe that our described anti-invasive and anti-metastatic effects following Shp2 inhibition to indeed be novel, and that they further support the role of integrin phosphorylation in invasive cancer progression.

Second, the claim that integrin is a direct substrate for SHP2 and PTP PEST is not fully substantiated. Many PTPs have effects on kinases that could also explain the results. For example, SHP2 has been identified as a FAK phosphatase. There are also other pathways regulated by phosphatases that contribute to invasion.

Third, the most striking result is the similarity of phenotype of a phosphatase and a kinase. I'd like to see more insight into what the authors think could explain this or how it might work, since a lot of the manuscript focuses on Dok1 (i.e. integrin phosphorylation-dependent) mediated invasion. It is also worth noting that despite similar effects on integrin phosphorylation, PTP-PEST and SHP2 are reported to have opposing effects on invasion (e.g. <https://doi.org/10.1158/0008-5472.CAN-18-0085> and 10.2147/OTT.S138833). In the screening data (Supplementary Table S1), the 3 siRNAs targeting PTPN11 (SHP2) give 3 different results (based on robust z score), therefore the follow up is critical to the conclusion that it is dynamics and not phosphorylation driving invasion.

Au: We appreciate the reviewer pointing out these earlier studies, which report SHP2 overexpression inducing ovarian cancer cell proliferation and migration and PTP-PEST perturbation reducing cell viability, but inducing invasion in GBM. Indeed, the phenotypic outcome of activating or perturbing signalling pathways in cancer is very much context dependent and likely to vary between different cancer types.

The proposed mechanisms in these studies are not linked to integrin phosphorylation and we fully agree with the reviewer that for our study the link to integrin phosphorylation and the follow-up validation experiments are critical. With the help of the expert advice of the reviewer given below, we have been able to further strengthen our manuscript, providing new data that fully support the conclusion that phosphatase-regulated dynamic integrin phosphorylation is critical for cell invasion.

Fourth, using FRET, the authors identify an interaction between Dok1 and cortactin and Tks5, however, neither were identified in the interactome of Dok1 determined by BiCAP. This is not addressed in the text.

Au: We appreciate these criticisms and have now taken several approaches to address these points as described below:

Significance

The manuscript sets out to gain insights into phosphorylation dynamics using elegant imaging approaches. This is a tour de force in cellular interaction studies, and the live cell reporter for phosphorylation events in the cell is really exciting. Although the importance of dynamic regulation of phosphorylation in shaping signalling outcomes is not new (see Gelens and Saurin. Dev Cell 2018), the work highlights it in a new and highly dynamic, disease relevant system. Furthermore, the screen of phosphatases highlights the complexity of the upstream regulation of just a single phosphosite and yields many areas for future investigation. In addition, the mechanism by which Dok1 promotes invasion is further clarified by identifying its interactome when bound to integrin. The combination of approaches will be an invaluable addition to the tool kits and knowledge of the phosphorylation signalling and adhesion fields. Finally, the data are also highly significant for the cancer field; understanding invasion processes is key for understanding metastasis, and SHP2 inhibition is shown to prevent metastatic colonisation in the lung. Therefore, SHP2 inhibitors should be considered for prevention of metastasis (intervention is not investigated) as well as inhibition of MAPK signalling.

Au: We thank the reviewer for this highly supportive assessment of the significance of our study.

Suggested improvements to address concerns raised above

1. The ITGB1 YYFF mutant provides compelling evidence that phosphorylation at these sites is essential for invasion. However, an equivalent tool to demonstrate that dephosphorylation is equally essential is missing. I fully acknowledge that this may not be possible to address since glutamates (i.e. a YEE construct) do not always recapitulate tyrosine phosphorylation. However, the authors have a clean way to evaluate this using their Dok1-integrin FRET reporter (perhaps they already have?). The only other alternatives I can think of are use of a synthetic amino acid, which I think is well beyond the scope of the manuscript. Or, use of constitutively active Src as shown in Figure 2c (although this is somewhat confounded by other functions of Src).

Au: We thank the reviewer for this expert advice.

a. Is YEE a feasible way to “block” dephosphorylation?

Au: We agree with the reviewer that the use of glutamic acid mutation as a phospho-mimetic for tyrosine residues is not a generally accepted method, and hence we had not taken this approach initially. However, in response to a comment by reviewer #2, we cloned single mutations for the integrin $\beta 1$ (ITGB1)-equivalent membrane-proximal and -distal NPxY sites in Illusia (Y783F and Y795F), and a YEE construct to assess the ability of glutamic acid mutations to simulate tyrosine phosphorylation in this reporter (Extended Data Fig. 3e, also provided below for convenience). These data suggest that YEE is a feasible tool that locks the reporter in a “phosphorylated” state (low FRET) and “blocks” dephosphorylation. However, we have also performed a large number of experiments to validate that loss of PTP activity broadly (i.e. through broad-spectrum inhibition using sodium orthovanadate), or specifically through targeted disruption of Shp2 or PTP-PEST using siRNA silencing

or with SHP099 (small-molecule against Shp2), increases the phosphorylation state of ITGB1 and effectively provides a way to “block” dephosphorylation. Thus, we feel that we have sufficient evidence to demonstrate that blocking phosphorylation through kinase inhibition or YYFF mutation, or dephosphorylation through PTP inhibition, lock the integrin in a state that is detrimental to the dynamic requirements of cancer cell invasion.

Extended Data Figure 3.

Validation of the phosphorylation-dependent changes of Illusia and the active dephosphorylation of ITGB1 in cancer and normal cells.

e, Representative FLIM-FRET images (left) and apparent FRET efficiencies (right) from HEK293 cells transfected with different Illusia variants (WT, Y783F, Y795F, YYFF and YYEE; n=4 biological replicates; 20-58 cells/condition/replicate; significance assessed using one-way ANOVA with a Šidák correction for multiple comparisons; NS, not significant, ***p<0.001). Scale bars, 20 μm.

b. If not, or in addition, what is the consequence of expressing constitutively active Src for invasion?

Au: Thank you for this interesting suggestion. We tested the effect of expressing constitutively-active Src(E378G) in MDA-MB-231 (MM231) cells. Here we employed the three-dimensional (3D) fibroblast-contracted collagen matrix platform that has been used throughout the manuscript to assess cancer invasion into an *in vivo*-relevant extracellular matrix (Reviewer-only Fig. 1). Interestingly, expression of constitutively-active Src(E378G) in MM231s expressing wild-type ITGB1(WT) led to a dramatic decrease in invasion in this assay platform. This is consistent with our conclusion that phosphorylation dynamics are key and over-activating phosphorylation is detrimental to invasion. We also observed significant inhibition of the modest invasion capability of the ITGB1 YYFF cells, as well as significantly reduced proliferation regardless of the integrin status. It is fairly well established that constitutively-active (oncogenic) kinase signalling cascades result in a phenomenon called “oncogene-induced cell rounding” and impaired invasion (PMID: 24790222; PMID: 27336951; PMID: 17158954). Several mechanisms have been described linked to oncogenic rounding, including increased integrin endocytosis and phosphorylation of focal adhesion proteins such as paxillin (PMID: 9372922), thus the ability of constitutively-active Src(E378G) expression to block invasion may also involve pathways beyond ITGB1-tail phosphorylation.

Reviewer-only Fig. 1.

Constitutively-active Src(E378G) attenuates invasion of MM231 breast cancer cells into 3D fibroblast-contracted collagen matrices.

a, Representative images (left) of MM231 cells expressing either ITGB1(WT or YYFF) with doxycycline-inducible expression of constitutively-active Src(E378G) invading into 3D fibroblast-contracted collagen I and stained with pan-cytokeratin (PanCK) to specifically detect the cancer cells and exclude fibroblasts from the analysis. Scale bars, 100 μ m. Quantification of invasion beyond 100 μ m (right), normalised to the total number of cells/region is shown). **b**, Representative images (left) and quantification (right) from the invasion assay in **a**. Proliferating cells were stained with the marker Ki67 to quantify the proliferative index by normalising the number of Ki67-positive nuclei to the total number of cells/region (n=4 biological replicates; 8 regions/replicate/cell line; significance assessed using one-way ANOVA with a Šidák correction for multiple comparisons; NS, not significant, *p<0.05, ***p<0.001). Scale bars, 100 μ m.

c. Does YYFF cause cells to round up like SHP2 and Src inhibitors when imaged in 3D collagen? (like Supplementary video 2). This would further cement the commonality of the phenotypes. Since there is a good chance that Src and SHP2 have functions that impact invasion beyond their effects on ITGB1.

Au: We thank the reviewer for this valuable suggestion. We have now assessed the phenotype of MM231 cells with endogenous ITGB1 knock-down and re-expression of either wild-type ITGB1(WT) or phosphorylation-defective mutant ITGB1(YYFF) in 3D collagen (Fig. 1d, also provided below for convenience). As the reviewer suggests, the loss of ITGB1 phosphorylation results in a significantly rounded morphology in the 3D matrices compared to cells where integrin expression has been restored with phosphorylation-competent ITGB1(WT). Importantly, loss of ITGB1 also results in cell rounding. This cements the commonality of the phenotypes beyond the inhibition of tyrosine kinases or PTPs, which may have multiple targets within the same pathway.

Figure 1.

ITGB1 phosphorylation supports efficient cancer cell invasion.

d, Representative images (left) and quantification of cell shape (i.e. solidity; right) of MM231 cells (green – actin staining) embedded in 3D collagen matrices overnight (magenta; n=3 biological replicates; 15-50 cells/condition/replicate; significance assessed using a Kruskal-Wallis test with a Dunn’s correction for multiple comparisons; NS, not significant, ***p<0.001). Scale bars, 20 μm.

2. The phosphatase screen identified SHP2 and PTP-PEST as negative regulators of Integrin phosphorylation (as read out by Dok1 binding to phosphorylated integrin). The current data do not fully support that the two phosphatases directly dephosphorylate integrin.

a. Figure 5b: Could the FRET reporters reveal indirect phosphorylation-dependent interactions? (i.e. between SHP2 and ITGB1)

Au: FRET allows the detection of protein-protein interactions when those proteins are very close (i.e. <10 nm; PMID: 24739578). For this reason, it gives a more reliable measure of protein-protein interaction than colocalisation analysis, far smaller than super-resolution approaches have yet to achieve (>20 nm resolution limit; PMID: 32527967). Thus, we believe that the FRET reporters reveal direct interactions between the proteins assessed in this study.

b. Add PTP PEST FLIM-FRET image to Figure 5b.

Au: In line with the reviewer’s suggestion, we have added representative images for the PTP-PEST FLIM-FRET to Fig. 5d (formerly Fig. 5b).

c. The GFP trap pulldowns using wt and “substrate trapping” mutants of each phosphatase show greater binding of integrin to the mutant. However, it looks as though if this were quantified the binding might actually be equal since the wildtype phosphatases are significantly less well expressed. This appears to be a tag issue since untagged SHP2 is not sensitive to mutational status (Supp figure 5c). To increase the confidence in these assays, find a tag that shows similar expression levels. In addition, one would expect that phosphorylated integrin would bind to the trapping mutant and any bound to the wildtype would be dephosphorylated. (i.e. blot for pY783). This would rule out an association mediated by the SHP2 SH2 domains, which is another possible explanation for the data in Figure 5b. Similarly, a co-IP with WT and YYFF ITGB1 could address the phosphorylation dependence of the “trapping”.

d. Since vanadate is used to “block” the association of SHP2 and integrin by FRET (Figure 5B), this could also be used to compete ITGB1 off in the pull downs (Figure 5C).

Au (response to comments c and d): As the reviewer suggested, we performed additional co-immunoprecipitation experiments to assess the specificity of the binding of Shp2 and PTP-PEST to ITGB1 (Reviewer-only Fig. 2). These experiments demonstrated similar results to those previously presented in Fig. 5c of the original manuscript. However, the inability of VO_4^{3-} to displace the phosphatases from ITGB1 suggests that in a cell suspension, after lysis, ITGB1 is bound by Shp2 and/or PTP-PEST via an interaction that is not mediated by the phosphatase active site binding to the tyrosine-phosphorylated substrate. Furthermore, quantification of the ITGB1/PTP-PEST and ITGB1/Shp2 interactions between the wild-type (WT) and substrate-trapping phosphatase mutants (Mut) revealed that the pull-down was not significantly increased with the “substrate-trapping” mutants, indicating that the trapping was not successful for these phosphatases (Reviewer-only Fig. 2). Given the sensitivity of the intermolecular FRET approach to phosphatase inhibition (Fig. 5c & 5d, formerly Fig. 5b), and in light of the new data, we have now removed this co-immunoprecipitation data from the revised manuscript. We thank the reviewer for their support in guiding the revised manuscript towards greater accuracy.

Reviewer-only figure 2.

Shp2 and PTP-PEST co-immunoprecipitate ITGB1, regardless of their activation state

a - b, Representative immunoprecipitation blots (left) and densitometry (right) of Shp2 (a) and PTP-PEST (b) demonstrate binding to ITGB1, even in the presence of VO_4^{3-} (100 mM; n=3).

e. Use a direct dephosphorylation assay. Overexpression in cells can lead to indirect effects. This could be done using a similar approach taken to demonstrate Src phosphorylation (figure 2e) of ITGB1 tails. Or take an approach using alkylated lysates to carry out dephosphorylation with recombinant protein and western blot for the Y783 phosphosite, or enrich tyrosine phosphorylated proteins with phosphotyrosine Ips.

Au: We are grateful to the reviewer for this excellent suggestion, which enabled us to demonstrate that ITGB1 tails can be directly dephosphorylated by the phosphatases. In line with the Reviewer's comment, we have now directly assessed the ability of recombinant PTP-PEST and Shp2 to dephosphorylate ITGB1 tail peptides (Fig. 5a & 5b). This was achieved through the use of the malachite green reagent to detect free phosphate, released from the ITGB1 peptides in the presence of functionally-active PTPs.

Figure 5.
ITGB1 is a substrate for PTP-PEST and Shp2.

a & b, Malachite green assay for free phosphate release after incubation of phosphorylated/non-phosphorylated ITGB1 peptides with recombinant Shp2 (a; n=5, triplicate reactions) or PTP-PEST (b; n=4, triplicate reactions). Significance assessed using Kruskal-Wallis test with a Dunn's correction for multiple comparisons; *p<0.05, **p<0.01, ***p<0.001).

f. Does double knockdown of SHP2 and PTP-PEST show an additive effect on ITGB1 phosphorylation?

Au: This is an interesting point. We performed siRNA-mediated silencing of both Shp2/*PTPN11* and PTP-PEST/*PTPN12* and probed for changes in ITGB1 phosphorylation at the NPxY(Y783) site (Reviewer-only Fig. 3). However, we did not observe a greater change in phosphorylation state than we previously detected with silencing of these PTPs individually (Extended Data Fig. 4e & 4f), or through PTP inhibition with SHP099 (against Shp2; Extended Data Fig. 5i) or with the broad-spectrum PTP inhibitor VO₄³⁻ (Fig. 3b-d; Extended Data Fig. 3c).

Reviewer-only figure 3.

Double-silencing of both Shp2/*PTPN11* and PTP-PEST/*PTPN12* does not lead to an additive effect on ITGB1 phosphorylation

a - b, Representative western blot (a) and densitometry (b) of MM231 cells after dual silencing of the indicated PTPs (n=3 biological replicates; significance assessed using one-sample t test;

* $p < 0.05$). c, Quantitative real-time PCR of *PTPN11* and *PTPN12* after silencing in MM231 cells (n=3 biological replicates; significance assessed using one-sample t test; * $p < 0.05$, *** $p < 0.001$).

Minor comments related to dephosphorylation:

g. Supplementary Figures 5c and e both require quantification since the lanes where the active phosphatases are present seem to be underloaded which could explain the lower levels of ITGB1 phosphorylation.

Au: Densitometry for Extended Data Figures 5c and 5f (formerly Supplementary Fig. 5c & 5e) has now been included in the revised supplement. Indeed, we see that overexpression of the PTPs for a prolonged period has a significant effect on integrin phosphorylation, but is less striking than the effects we see from the short-term inhibition using small molecules.

h. What is the % input for fig. 5c? This is important since the SHP2 mutant enriches ITGB1 to higher levels than the input. Is this consistent with the stoichiometry of phosphorylated ITGB1 in these cells? I.e. if 50% of ITGB1 is pulled out – is it feasible that 50% of ITGB1 was phosphorylated in the cell lysates?

i. Check molecular weights in Figure 5c. SHP2 should be ~70kDa, so adding 20+ kDa via clover should increase its MW above the 75 kDa marker (I acknowledge MW ladders are not perfect – but it looks a bit odd). I could not see a method for the GFP trap experiment, please include.

Au (response to comments h and i): We thank the reviewer for their thorough assessment of our work. As described in our response to comments c and d above, we have now removed these immunoprecipitation experiments from the revised manuscript. However, the molecular weight ladders for the Reviewer-only Fig. 2 have been corrected.

We apologise for missing some of the details of the GFP-trap IPs in the Methods. We have now updated the method, “Immunoprecipitation (IP),” to include more detail on how these GFP-trap IPs were performed (lines 1039 to 1057). Notably, all immunoprecipitation experiments were performed using ~90% of the sample as input and keeping 10% for the loading control. These details are now also clearly described in the revised Online Methods section.

3. The dynamic phosphorylation of ITGB1 is used as a potential explanation for the similar phenotypes of inhibiting SHP2 (a phosphatase) and Src (a kinase).

a. Line 428. The title of this paragraph undermines the point made using the Shp2 inhibitor. Consider adding “dynamics” since one would expect ITGB1 to be phosphorylated upon SHP2 inhibition (according to the Illusia reporter data), yet this impairs invasion.

Au: We agree with the reviewer and have added “dynamics” to the Results paragraph title, “ITGB1 phosphorylation dynamics support invadopodia formation and metastatic colonisation.” We thank the reviewer for supporting the clarity of the manuscript.

b. I would like to read more about how the authors think this concept could work, the discussion is very brief. For example, it could be that Src and SHP2 elicit a “flickering” effect on ITGB1 phosphorylation, an effect at the single molecule scale. Alternatively, one could imagine that this occurs on a population scale, perhaps even setting up gradients of phosphorylation, where either Src or SHP2 predominates (but what acts upstream of them?). The video S2 appears to show front-rear differences in phosphorylation (via FRET reporter) on a whole cell scale. And then at the leading edge there are regional differences. Both SHP2 and Src inhibition lead to an effective loss of polarity (rounding up). Is this cause or consequence? On one hand SHP2 and Src could have other impacts on polarization and this is indirect. Alternatively, the inability to establish ITGB1 phosphorylation differentials may impair polarity. This could be addressed as suggested in point 1c using ITGB1 YYFF and/or YYEE in 3D culture.

Au: We are grateful for these excellent points and have now discussed the “flickering” at the single-molecule scale in the manuscript, “Integrins and their downstream signalling cascades regulate cell migration and adhesion on several scales. At the single-molecule level, their dynamic binding and unbinding to ECM ligands, and linkage to the acto-myosin cytoskeleton, define the cell-specific optimal conditions for cell migration³⁶” (lines 567 to 570). Similarly, we have also added text to outline the astute observation of front-rear polarisation, “On the scale of the whole cell, front-rear polarity establishment is regulated by integrins and their associated signalling^{37, 38}. At this scale, live-cell imaging of Illusia in invading cells appears to show front-rear differences in phosphorylation, with flickering and regional differences at the leading and trailing edges (Supplementary Video 2). This implies that integrin phosphorylation may be polarised in cells, which may contribute to cell migration that is likely co-ordinated by rapid phosphorylation cycles of individual integrin molecules. Future studies are needed to interrogate how this is governed across scales, and the fine-tuning of this process by tyrosine kinase/phosphatase networks” (lines 570 to 578). Furthermore, we performed 3D culture experiments to compare the 3D spreading of MDA-MB-231 cells with ITGB1(WT or YYFF), in response to comment 1c above (Fig. 1d in the revised manuscript), and found a clear morphological defect in the cells lacking ITGB1, or with rescue using the phosphorylation-defective YYFF mutant isoform.

c. Does Src inhibition have the same effect as SHP099 and ITGB1 YYFF on tumour colonization? (Figure 8e) Why was it not included?

Au: Several studies have already demonstrated reduced MM231 cell colonization of the skeletal muscle, lung and brain after Src inhibition with either dasatinib or saracatenib (PMID: 32170411; PMID: 23913825); an approach that is also effective in reducing metastasis to the lung by hepatocellular carcinoma (PMID: 27460949) and murine sarcoma (PMID: 21115886) cells. This established effect of Src inhibition on reducing metastatic colonisation in breast and other cancers led to our decision to reduce the animal numbers for our own experiment, in line with the 3Rs principle. Instead, we focused on the previously unknown effect of integrin phosphorylation on lung colonisation, as well as the less well-established role of Shp2 inhibition in this process. However, we have now included discussion of this point, “Indeed, Src inhibition has already been demonstrated as an effective anti-metastatic therapy in MM231 cells^{50, 51}, and recent work has found similar

effectiveness for SHP099 in reducing lung colonisation by mouse 4T1 breast cancer cells⁵² (lines 613 to 616).

Minor points:

1. Define ITGB1 in introduction (mostly integrin β 1 is used, but occasionally ITGB1 is used. E.g. line 80 vs line 88).

Au: We apologize for the lack of clarity. We have now stated that integrin β 1 is “referred to as ITGB1 from hereon” in the revised manuscript text (line 76) and harmonized the revised text to use ITGB1 after this instance.

2. Include quantification of Western blots e.g. Supp Figures 2i, 3e and f, 4e and f, 5c and e

Au: In line with the reviewer’s request, Extended Data Fig. 2i has now been quantified and the densitometry is included in the new panel Extended Data Fig. 2j. Similarly, for Extended Data Fig. 5c & 5f (formerly Supplementary Fig. 5c & 5e), the densitometry is now included to the right of the representative western blots. Related to Extended Data Fig. 4e & 4f, the densitometry for PTP-PEST and Shp2 was previously included in Fig. 4. However, as reviewer #3 also raised this point, we have now updated the figure legends for Fig. 4d and 4e, and Extended Data Fig. 4e & 4f to reference the location of the quantification and representative western blots between the two figures (lines 304-305 in the revised manuscript and lines 137 to 141 in the revised Extended Data). We have also added densitometry for pITGB1(Y783) to show the significant upregulation after silencing of Shp2/*PTPN11* or PTP-PEST/*PTPN12* individually (Extended Data Fig. 4e & 4f). Lastly, we feel that Extended Data Fig. 3f & 3g (formerly Supplementary Fig. 3e & 3f) serve to demonstrate the lack of cleavage of Illusia upon treatment, or at baseline, and as such, quantification of a band that is not present is neither reliable, nor will it support the clear result that is presented. Thus, we have opted not to include quantification of these panels, instead allowing the reader to assess the data in the form of representative western blots.

3. Supp figure 5 b – no image for mScarlet control (control images are otherwise present)

Au: We have now provided a representative image for the mScarlet control in the revised Extended Data Fig. 5b (revised panel given below for convenience).

Extended Data Figure 5.

Inhibition or overexpression of Shp2 or PTP-PEST modulates ITGB1 phosphorylation.

b, Representative FLIM-FRET images (left) and quantification of apparent FRET efficiency (right) in MM231 cells with stable Illusia expression and constitutive overexpression of PTPN11 (Shp2, WT) or a phosphatase-dead mutant (Shp2, Mut; PTPN11(D425A, C459S)) (n=4 biological replicates; 24-25 cells/condition/replicate; significance assessed

using a one-way ANOVA with a Tukey correction for multiple comparisons; NS, not significant, *** $p < 0.001$). Scale bars, 20 μm .

4. Supp Figure 7c. The study implicates SHP2 in invadopodia formation. Could it be incorporated into this diagram?

Au: The invadopodium scheme in Extended Data Fig. 7b was included to help introduce invadopodia to a non-expert reader, and as such, included components that were well-established to localize to invadopodia. However, we also included Src in the previous version, as we and others have demonstrated that Src activity is intricately linked to invadopodia formation. And yet, we did not show that Src, Shp2, Dok1, PTP-PEST or Arg localise to invadopodia. So, while there is a clear role for these proteins in invadopodia dynamics, their precise localization with relation to invadopodium remains unexplored. In line with this, we have updated the scheme to include only components known to localize to invadopodia, in their rough locations (Extended Data Fig. 7b, reproduced below for convenience).

Extended Data Figure 7.
Cofilin, VPS35 and annexinA6 are recruited to the phosphorylated Dok1/ITGB1 complex.
b, Schematic of an invadopodium degrading the ECM.

5. Supplementary video 2. Switch order of Sara and SHP099 in description to correspond to order in video. Could also include time frames in video legend for ease of interpretation.

Au: Information for the time frames has been added to Supplementary video 2, as well as switching the order of Sara and SHP099 in the video description.

6. Typos in Figure 8f YYYYF and Supp video 2: Illusia

Au: The typos have been corrected in the revised Fig. 8f and Supplementary video 2.

7. Figure 7e. Loss of Dok1 has a greater impact than ITGB1 YYYYF. Are there other integrins that bind to Dok1 that could explain this? (i.e. Dok1 is downstream and therefore a core component for invasion mediated by multiple integrins)

Au: The reviewer makes an important point. We have now expanded the Results section to include details related to the phosphorylation-specific binding of Dok1 to several integrin β isoforms, "Notably, silencing of Dok1 resulted in a more pronounced phenotype than non-phosphorylatable

ITGB1, suggesting that the role of ITGB1 can be compensated by other adhesion receptors, but that the scaffolding functions of Dok1 are less dispensable for invadopodia formation. This is further supported by the established phosphorylation-specific binding of Dok1 to other integrin β isoforms^{11, 35} (lines 450 to 455). We have also expanded the discussion to include a brief description of the possible wider role of Dok1 and its likely binding to multiple integrins during cancer invasion, “Notably, there are no obvious cell migration defects reported for mice carrying the YYFF mutant ITGB1^{5, 9}. However, in organisms with fewer integrin isoforms, YYFF mutation of the *ITGB1* orthologs leads to defects in distal tip cell migration and ovulation (*C. elegans*), while in *D. melanogaster* it results in disruption of the normal functions of myotendinous junctions^{44, 45}. It is important to note that the two conserved cytoplasmic NxxY motifs are present in multiple integrin β -subunits and Dok1 can interact with them⁴⁶. The significant decrease in invadopodia observed after Dok1 knock-down (Fig. 7e) suggests that Dok1 could be a core component for invasion mediated by multiple integrins and implies a wider impact of our findings” (lines 598 to 606).

8. Include an acknowledgement in the text that cortactin and Tks5 were not identified in the BiCAP experiment. This could well reflect the different approaches used.

Au: This is important to acknowledge and was something that surprised us when we performed our FRET experiments. Thus, we have now added an additional statement to reflect this difference in approaches providing different hits, “However, very few invadopodia components were identified in the BiCAP, and so we took a more targeted approach using intermolecular FRET to assess the interaction of the phosphorylated ITGB1 complex with these scaffold proteins” (see revised manuscript, lines 485 to 488).

Clarity

The text is very clear and accessible to a broad audience. There are some areas where previous knowledge, especially on phosphatases, is not provided. The functions of both Src and SHP2 in focal adhesions have been described and this should be acknowledged as it clearly has a bearing on the conclusions drawn here. The flow is logical, but I have the feeling that this study initially set out to characterise the phosphorylated integrin-Dok1 complex and then sought to understand its regulation. The initially counterintuitive role of the phosphatases requires further elaboration, since it is contradictory to the pro-invasion role of the phosphorylated complex. The authors are attempting to address a challenging question with innovative and clearly described tools. I have added a few references below that could add to the discussion.

References

The references are appropriate. The authors might consider adding references related to SHP2 in focal adhesions and polarity (e.g.10.26508/lsa.202201557 and 10.1158/1541-7786.MCR-12-0578). Interestingly, it was recently shown that ITGB1 and FRS2 (mentioned in discussion) knockdown sensitise cells to Shp2 inhibition (<https://doi.org/10.1158/0008-5472.CAN-23-1127>). As mentioned above, dynamic phosphorylation regulation has been discussed previously and might help shape the discussion on the topic: <https://doi.org/10.1016/j.devcel.2018.03.002>

Au: We thank the reviewer for suggesting the above citations, which we have now included at relevant points in the revised manuscript. Notably, we expand on the possibility of Frs2 as a phosphorylation-

sensitive adaptor mediating downstream integrin functions, “Interestingly, loss of Frs2 has recently been shown to sensitise cells to Shp2 inhibition, suggesting a further link between cancer progression and additional phosphorylation-sensitive integrin complexes⁴³” (lines 593 to 595; additional reference 43 (PMID: 37934115)). We also add a new paragraph in the discussion to outline the role of Shp2 in migration, as well as outlining the role of phosphorylation dynamics in our model and its place in the literature, “One insight comes from a study looking at the role of Shp2 in regulating focal adhesion kinase activity at the cell periphery, which was shown to encourage nascent adhesion formation consistent with lamellipodia spreading³⁹. Similarly, the ability of Shp2 to promote IAC maturation through ROCK2 activation indicates that phosphorylation dynamics are important to modulate the activation state of different IAC components to ensure that dynamic processes are not stalled by over/underactivation⁴⁰ (further discussed in ⁴¹)” (lines 578 to 584; Additional references 39 (PMID: 23512980), 40 (PMID: 36096674) and 41 (PMID: 29587141)).

Reviewer #2:

Remarks to the Author:

Summary:

Authors report the regulatory mechanisms governing integrin phosphorylation and their efforts to understand the role of integrin phosphorylation in the progression of cancer. They started with cell line experiments that using $\beta 1$ integrins with non-phosphorylatable residues at the cytoplasmic domain (YYFF) and showed reduction in invasion through the collagen matrix. This observation suggests the importance of $\beta 1$ integrin phosphorylation at the NPxY motifs. Then they developed a FRET-based biosensor, termed Illusia, that utilizes fluorescence proteins, a portion of $\beta 1$ cytoplasmic tail (aa772-798), and Dok1 protein that preferentially binds to a phosphorylated tyrosine, and uses it as a proxy of the phosphorylation status of the integrin $\beta 1$ tail. Authors showed that the integrin $\beta 1$ phosphorylation is related to cancer cell motility and identified tyrosine phosphatases and regulators for $\beta 1$ phosphorylation. The manuscript is informative and identifying direct involvement of other components in the integrin/Dok1 complex using FRET sensors is exciting. A suitably revised version of the manuscript will be a good candidate for Nature Cell Biology. Please consider the following questions and comments.

Au: We thank the reviewer for their positive comments and supportive feedback.

Major comments:

1. Authors reported the downregulated invasion index of the YYFF mutant using 3D collagen matrix invasion assay in fig.1. Their claim, I believe, that this invasion phenotype difference with the mutations is responsible for cancer cell invasion. However, this may be a general motility difference that is not specific to cancer cells. So they need to test if the YYFF effect is cancer cell specific, and if normal cells do the same, perhaps the invasion index is just the cell motility index in the 3D collagen matrix?

Au: We agree with the reviewer that the reduced invasion that we see in the MDA-MB-231 (MM231) cells with non-phosphorylatable integrin \$\beta 1\$ (ITGB1(YYFF)), compared to the wild-type (ITGB1(WT)), could also be related to reduced cell motility. To test this possibility, we performed a random migration assay on cell-derived matrices (CDMs; PMID: 29048422; PMID: 37436978) to make the same comparison as in Fig. 1c. Interestingly, we observe a significantly higher migration speed in the ITGB1(YYFF)-expressing MM231s compared to the ITGB1(WT)-expressing MM231s on the CDMs. Given the impaired invasion of the YYFF cells, these data suggest that the differences in invasion are not owing to differences in cell migration (Reviewer-only Fig. 4).

Reviewer-only figure 4.

Random migration of MM231 cells on CDMs shows minimal differences between those with WT or YYFF ITGB1.

a, Representative migration tracks from a single random migration experiment, where MM231 cells with ITGB1(WT) or ITGB1(YYFF) were seeded overnight on CDMs and the SiR-DNA-stained nuclei imaged every 5 minutes. $n = 1156$ (WT) or 1290 (YYFF) tracks. **b**, Migration speed for the random migration of MM231 cells with ITGB1(WT) or ITGB1(YYFF). $n=3$ biological replicates (750 – 1826 tracks/condition/replicate). P-value from a Mann-Whitney test (** $p < 0.001$).

Related to the cancer-specificity, we have previously attempted to generate MCF10A (normal epithelial cells) without ITGB1 expression, which would allow us to overexpress ITGB1(WT) and ITGB1(YYFF) to then assess the functional effects in normal cells. However, the cells that survived the puromycin selection did not show any loss of ITGB1 (Reviewer-only Fig. 5). Further to these attempts, generation of TIFs with ITGB1 depletion also results in complete cell death after puromycin selection, suggesting that any cells that are positive for viral transduction are no longer viable. Together, these challenges have limited our ability to assess the role of ITGB1 phosphorylation in normal cells. However, we have validated the bulk of our findings in this manuscript in normal cells with endogenous ITGB1, either MCF10A, TIFs or HEK293 cells. Work by others has demonstrated that while mutation of both NPxY sites of ITGB1 to alanines is embryonic lethal (PMID: 16618804) and demonstrates a severe skin phenotype if restricted to the keratinocyte lineage (PMID: 23702582; PMID: 16954348), mutation to phenylalanines has a milder phenotype and minimal effect on normal tissue development (PMID: 16954348). This suggests that normal invasion processes occurring during mouse organogenesis, and as part of the oestrous cycle in the mammary gland, are not largely affected by loss of ITGB1 phosphorylation dynamics. However, when the skin of mice with ITGB1(YYFF) mutation were exposed to a potent carcinogen, the loss of ITGB1 phosphorylation dynamics delayed tumour onset (PMID: 21876123). Of note, in organisms with a reduced number of integrin β isoforms, similar mutations have been shown to have more pronounced effects. For example, Y/F mutation of the NPxY site in the *C. elegans* ITGB1 equivalent gene *β pat-3* leads to defects in distal tip cell migration and ovulation, while not adversely affecting viability (PMID: 20063417). Similarly, YY/FF mutations in the NPxY sites of the ITGB1 equivalent gene *β ps* in *D. melanogaster* showed defects in their myotendinous junctions (MTJs) due to a low turnover of *β ps* (PMID: 21089076); a concept now included in the discussion (lines 598 to 606). Thus, it is likely that in a normal tissue setting, mutation of the NPxY sites of ITGB1 can be compensated for by other integrin β isoforms, resulting in a mild phenotype in the mouse models previously assessed.

Reviewer-only figure 5.

MCF10A cells were resistant to shRNA-mediated depletion of ITGB1.

a, Representative western blot with MCF10A cells either before treatment with the ITGB1 shRNA (i.e. WT) or after treatment with shβ1 lentivirus and selection with puromycin. **b**, Densitometry assessing relative ITGB1 levels in WT or shβ1 MCF10A cells (n=4 biological replicates; significance assessed with a one-sample t test; NS – not significant).

2. To my understanding, Illusia reflects the phosphorylation activities that can be targeted to the tail motif (i.e. kinase/phosphatase activity for the freely diffusing β1 tail-like substrate), rather than directly reporting integrin β1 phosphorylation. The sensor may not properly reflect the status of ‘active’ integrins connected to cytoskeletal structure via adaptor proteins. Please make this clear and discuss the limitations and validity of the Illusia.

Au: We appreciate this criticism and agree that this is an important distinction that needs to be explained. We have now clearly stated the indirect readout that Illusia provides in the Results section when the FRET reporter is first introduced, “Illusia supports live assessment of the kinase/phosphatase balance in the cell. However, it is important to note that Illusia reflects the phosphorylation status of the integrin tail motif (as a freely diffusing, membrane-tethered β1 tail-like substrate), rather than directly reporting ITGB1 phosphorylation” (line 144 to 147).

3. It is unclear if Illusia requires phosphorylation of the both NPxY motifs within the sensor β1 tail region for FRET change. What could be the FRET change or lifetime change when only one of them is phosphorylated? Another question is, can Illusia also bind to a phosphorylated integrin in cell membrane.

Au: The reviewer raises an interesting question. The previous work using integrin peptides and recombinant Dok1 found an increase in binding to phosphorylated β3(Y747), β3(Y759), β1A(Y783) and β7(Y778), when compared to the non-phosphorylated peptides (PMID: 19843520). However, β1A(Y795) was not assessed in this study, and yet several other studies have found that mutation of the individual NPxY sites has less of an effect than the double mutation, suggesting some compensation or synergy in function between the sites (PMID: 23702582; PMID: 21876123; PMID: 33119040). To interrogate this important point in detail, we have assessed the sensitivity of Illusia to mutation of the equivalent NPxY sites to ITGB1(Y783) and ITGB1(Y795), finding that mutation of the membrane-proximal ITGB1(Y783) site in Illusia to phenylalanine removes the sensitivity to PTP inhibition (Extended Data Fig. 3e). This suggests that the Dok1 PTB domain (included in Illusia) primarily binds to the ITGB1(Y783) site upon phosphorylation.

Extended Data Figure 3.

Validation of the phosphorylation-dependent changes of Illusia and the active dephosphorylation of ITGB1 in cancer and normal cells.

e, Representative FLIM-FRET images (left) and apparent FRET efficiencies (right) from HEK293 cells transfected with different Illusia variants (WT, Y783F, Y795F, YYFF and YYEE; n=4 biological replicates; 20-58 cells/condition/replicate; significance assessed using one-way ANOVA with a Šidák correction for multiple comparisons; NS, not significant, ***p<0.001). Scale bars, 20 μm.

Related to the second point, we performed a GFP-trap immunoprecipitation in MDA-MB-231 (MM231) cells stably overexpressing Illusia or mTurquoise2 alone (FRET donor fluorescent protein; Extended Data Fig. 3h, also included below for convenience; GFP-trap binds to mTurquoise2, which is a GFP variant). We also included sodium orthovanadate (VO₄³⁻) treatment to maximize the level of phosphorylated ITGB1 in cells prior to lysis. However, we were unable to co-immunoprecipitate ITGB1 with either Illusia or mTurquoise2 (mT2), suggesting that the Illusia does not bind phosphorylated integrin in the cell membrane, or that any binding of the Dok1 phosphotyrosine-binding domain of Illusia to ITGB1 is weak/transient and not strong enough to interfere with the normal function of phosphorylated ITGB1.

Extended Data Figure 3.

Validation of the phosphorylation-dependent changes of Illusia and the active dephosphorylation of ITGB1 in cancer and normal cells.

h, Representative GFP-trap IP of mT2 or Illusia from MM231 cells stably expressing the reported constructs (n=3 biological replicates).

4. Illusia uses the phosphotyrosine binding domain of Dok1. How does overexpressed Illusia, localized on the membrane, affect the Dok1 binding proteins (Cofilin, CTTN, and TKS5 in fig.8)?

Au: Indeed, we had not previously assessed the effect of Illusia overexpression on components of the Dok1-ITGB1 complex. We have now assessed the levels of Dok1, Cofilin, ITGB1, TKS5 and Cortactin in MM231 cells stably-overexpressing either Illusia or mTurquoise2 alone (FRET donor fluorescent protein; Reviewer-only Fig. 6). In these cells, we observe no significant differences in the expression levels of those proteins assessed. We are grateful for the reviewer for helping us to clarify this.

Reviewer-only Figure 6.

Expression of Illusia and mT2 does not lead to a significant change in the expression level of the invadopodia components assessed in this study.

a-b, Representative western blots (**a**) and densitometry (**b**) assessing the expression of Dok1, ITGB1, Cofilin, TKS5 and Cortactin in MM231 cells stably expressing mT2 or Illusia, compared to parental control cells (n=4 biological replicates; significance assessed with a one-sample t test; NS – not significant).

5. The structure of unphosphorylated Illusia is not clear (in main text and figure). Why does it have the well-folded structure with high FRET? What is the expected distance between donor and acceptor in the unphosphorylated and phosphorylated states?

Au: The structure of intramolecular FRET reporters is commonly estimated based on experimental observations. For example, the FRET reporters for Src and focal adhesion kinase (FAK) show a similar change in FRET to Illusia, whereby the SH2 domain in the Src and FAK reporters binds to a kinase-specific substrate domain that leads to a conformational change when phosphorylated that decreases the FRET (PMID: 15846350; PMID: 21792185; PMID: 22900044). Conversely, reporters of Ser/Thr phosphorylation typically use the FHA1 domain of yeast Rad53 (a.a. 241–382), pairing this with a kinase-specific substrate that when phosphorylated results in an increase in the FRET (PMID: 21976697). Given that both the Src/FAK and Illusia FRET reporters utilise protein domains that bind to phosphorylated tyrosine (i.e. SH2 in the Src/FAK FRET reporters and the Dok1 phosphotyrosine-binding domain (PTB) in Illusia), we designed Illusia under the assumption that a similar FRET change would occur in parallel with changes in the phosphorylation state of the ITGB1 cytoplasmic domain

(i.e. at the NPxY sites). Observing a higher FRET in the less phosphorylated state, and a lower FRET in the phosphorylated state, we created our schematics in a similar way to those of the previously published schematics for the Src and FAK FRET reporters. To avoid misleading the readership, we have now added the additional clarification in the Fig. 2a legend to say, “Schematic representation of the possible conformations for the ITGB1 intramolecular FRET biosensor (Illusia), where the mTurquoise2-YPet FRET pair is separated by the phosphotyrosine-binding domain (PTB) from Dok1, a linker and the cytoplasmic domain (aa772-798) from ITGB1 (including the two NPxY motifs).”

6. It would be nice if ‘FRET efficiency (%)’ is better defined in the main text and figure. In FRET efficiency box plots, the y-axis may be ‘apparent FRET efficiency’ defined in method ($E = (1 - \tau DA / \tau D) \times 100$). If so, please make that explicit. Is a single data point in the box plots the apparent FRET efficiency from each pixel (the average of molecules in a diffraction limit) or the average value of a single cell?

Au: We thank the reviewer for clarifying our terminology to “apparent FRET efficiency”. We have now corrected the plots and legends to say, “apparent FRET efficiency.” We also now state in the methods that the presented FLIM-FRET data uses an average value from a single cell, and not from each pixel in diffraction-limited space.

7. Authors used the siRNA library to identify specific PTPs responsible for $\beta 1$ dephosphorylation and found PTPN11 and PTPN12. Also, 57 of them showed an unexpected increase of FRET in Illusia. It would be beneficial if authors check any compensatory effects of other genes (upregulation or downregulation). Although it might be out of the scope for the current work to elucidate the indirect role of the phosphatase in upregulating integrin $\beta 1$ phosphorylation, authors can provide expression level of a few phosphatases using western blot to make sure their expression level is similar to WT cells.

Au: It was also a surprise to us to find that PTP hits from the siRNA screen showed an unexpected increase in FRET, consistent with an indirect role in increasing ITGB1 phosphorylation within the cell. The focus of the current manuscript was to identify direct PTPs with a role in regulating ITGB1 functions through dephosphorylation. However, we recognize that gene knock-down (KD) often leads to compensatory upregulation of other genes in order to sustain cellular functions. Thus, to ascertain if there is a compensatory upregulation of the known direct PTPs, Shp2/*PTPN11* or PTP-PEST/*PTPN12*, we analysed their expression in samples that showed an increase in FRET after siRNA-mediated KD. This assessment was performed using the same RNA samples as were assessed for siRNA-specific KD in Extended Data Fig. 4c, to give the clearest comparison to the siRNA FRET screen (Reviewer-only Fig. 7a). Of the six PTPs that were silenced, none showed a clear change in either Shp2/*PTPN11* or PTP-PEST/*PTPN12*. Furthermore, we assessed the compensatory changes that might occur as a result of silencing the known direct PTPs, Shp2/*PTPN11* or PTP-PEST/*PTPN12*, again observing no compensation (Reviewer-only Fig. 7b). Future work would involve bulk RNA sequencing to obtain a broader picture into the changes occurring that result in this increased FRET after siRNA-specific KD, but we do not feel that this fits within the scope of the current manuscript.

Reviewer-only Figure 7.

No significant compensation in PTP expression is observed upon Shp2/*PTPN11* or PTP-PEST/*PTPN12* silencing.

a, Quantitative real-time PCR of *PTPN11* and *PTPN12* after silencing *SSH2*, *PTPN11*, *CDC25B*, *PTPN12*, *CDC25A* and *TPE* with three siRNAs/target (A, B and C) in MM231 cells (n=1 biological replicate). **b**, Quantitative real-time PCR of *PTPN11* and *PTPN12* after silencing *PTPN11* and *PTPN12* in MM231 cells (n=4 biological replicates; significance assessed using one-sample t test; *p<0.05, ***p<0.001).

8. Co-IP results in Fig. 5C are not clear on how GFP Trap was done and unclear why there are stronger bands for mutants and why only Shp2 CoIP showed anti-GFP band. Why was this not done in MM231 cells.

Au: We apologise for the lack of clarity in the methodology applied. We have now updated the method, “Immunoprecipitation (IP),” to include more details on how the pY and GFP-Trap IPs were performed (lines 1039 to 1057).

Related to the use of the HEK293 cells for the IPs, overexpression of the PTPs was challenging in the MM231 cells. This wasn’t an issue for the FRET experiments, as we could select positively transfected cells for the microscopy, but when we attempted IPs there was too low protein yield from the MM231s. So, we chose to use FRET to validate the interaction in MM231 cells and performed the molecular biology with HEK293 cells, as these are easy to transfect and overexpressed sufficient protein levels for this assay. However, based on the recommendations of reviewer 1, we performed additional experiments using VO_4^{3-} to block the phosphatase activity of the PTPs. These experiments showed that the interaction between the PTPs and ITGB1 in the HEK293 lysates was not specific to the phosphatase-substrate pY interaction mediated by the phosphatase catalytic site, suggesting that in cell lysates the interaction was occurring in some other domain of Shp2 and PTP-PEST. For that reason, we have removed this data from Fig. 5, instead including new data that demonstrates the ability of recombinant Shp2 and PTP-PEST to dephosphorylate ITGB1 peptides (Fig. 5a & 5b). We thank the reviewer for their support in guiding the revised manuscript towards greater accuracy.

9. I am not sure if 'dynamic regulation' in the title clearly describes the ambivalent effect of phosphorylation and dephosphorylation shown in this manuscript. What's a little confusing about the data is that it is not the absolute level of phosphorylation but the ability to change phosphorylation state that is important, and perhaps that is what they mean by dynamic regulation?

Au: Indeed, the use of "dynamic regulation" in the title is indicative of the manuscript's conclusion, that is referring to the ability of invading cancer cells to dynamically switch the phosphorylation state of ITGB1, as the reviewer has correctly concluded.

Minor comments:

1. It is up to the author to decide, but we recommend using 'fluorescence resonance energy transfer' instead of 'Förster resonance energy transfer' honoring Theodor Förster who was a longtime member of the Nazi party and chaired a department in occupied Poland. Even though he contributed to the understanding of FRET, we can name the phenomena without honoring someone associated with inhumane atrocities.

Au: We agree that the phenomenon of resonance energy transfer is a fundamental physical process, in which Theodor Förster contributed to our understanding, but the use of fluorescence resonance energy transfer (FRET) is increasingly common. Thus, we have updated the text to use the terminology suggested, "fluorescence resonance energy transfer" (line 135).

2. Fig 1d quantification shows the significance symbol (*), but it is not described. According to the main text the p-value would be higher than >0.05. Please specify p-value and let readers interpret the data and its statistical significance. Using the symbol without the description will mislead readers. In Fig. 5C caption, PTPN11 and PTPN12 would need to change to Shp2 and PTP-PEGT, respectively for consistency.

Au: We have now updated the Fig. 1e legend (formerly Fig. 1d) in the revised manuscript text to indicate that the (*) represents a p value less than 0.05 (see revised Fig. 1e legend). Furthermore, in line with the Springer Nature publishing guidelines, all the statistical source data and exact p values are now provided as a separate supplementary file to ensure that readers are not misled by the findings of the manuscript (see new Supplementary Table 6), as well as including the following at the end of each figure legend, "Source data and exact p-values are provided in Supplementary Table 6." We appreciate the point that the different nomenclature for the genes (*PTPN11* and *PTPN12*), and the proteins (Shp2 and PTP-PEST), can be confusing and have now update the figure legends accordingly. For example, current Figure 4c "siRNAs (A, B or C) against either *PTPN11* (b; Shp2) or *PTPN12* (c; PTP-PEST)."

3. I cannot find the main text describing the data of Fig. 3d.

Au: We thank the reviewer for their careful review and apologize for the mistake. The reference to this data was indeed missing from the text and has since been added in the revised manuscript (lines 229 to 230).

4. Author may use one of VO₃⁻ or Vanadate (Fig. 3e) for consistency of figures.

Au: In line with the reviewer's comment, we have now updated all of the figures to include "VO₄³⁻" after the initial introduction of "sodium orthovanadate (VO₄³⁻)" on lines 227 to 228.

5. 'NTC' should be defined in the figure legend. Please check all acronyms.

Au: We thank the reviewer for noting this error. We have added the expanded form of "siNTC - non-targeting control siRNA" to the legend for Figure 4 and confirm that all other acronyms are introduced at their first use.

6. Both 'FRET' and 'Phosphorylation' are used for the guide of lifetime color code randomly. Please be consistent.

Au: We understand the reviewer's concern related to the labelling of the lifetime colour code. However, the use of "FRET" and "Phosphorylation" were included for intermolecular interactions between different proteins or phosphorylation respectively and so were not used randomly, but with the purpose of clarifying for the reader the data that is being presented. To improve the clarity of the FRET annotations, we have now modified the labels for the lifetime colour code to include either "interaction" or "phosphorylation" to demarcate the different applications of the FRET systems used throughout, while clearly stating that this is a FRET readout by FLIM in the figure legends. Furthermore, in line with reviewer #3, minor point 8, that the different FRET pairs used could be more clearly indicated, we have now included schemes to clarify the use of intermolecular FRET using GFP-/RFP-variants or Illusia (i.e. intramolecular FRET between mTurquoise2 and YPet).

Reviewer #3:

Remarks to the Author:

Overview: The manuscript 'Dynamic regulation of integrin β 1 phosphorylation supports invasive breast cancer progression' focuses on the mechanism and implications of integrin phosphorylation at the NPxY (783/795) sites using Förster resonance energy transfer and proteomics approaches. It is well recognized that integrins play an important role in various extracellular matrix (ECM) dependent cellular functions such as adhesion formation, migration, invasion, rigidity sensing, and mechanotransduction. However, the control and modulation of these various functions by integrins is not fully understood. In this manuscript, the authors investigate the phosphorylation of integrin beta1 at Y783 and Y795 as a functional requirement for invadopodia formation, and its potential role in cancer cell invasion and metastasis. The delicate balance of this phosphorylation state mediated by kinases and phosphatases explored in this manuscript helps to clarify one of the intrinsic mechanisms by which a cell can modulate invadopodia formation and invasion to regulate cancer metastasis through phosphorylation of key integrins that control tumor cell adhesion activity. The role of proteins such as integrin beta1, Dok1 and Cortactin in invadopodia formation has been extensively studied and previously reported. Nevertheless, how these proteins are recruited to the invadopodia complex remains an open question. The authors present comprehensive cell biological studies to show that phosphorylated integrin beta1 can serve as a base for the recruitment of proteins involved in invadopodia formation. Mechanistically the authors determined that the regulation of integrin beta 1 phosphorylation is mediated through a balanced combination of the activity of Src and Arg kinases, and PTP-PEST and Shp2 phosphatases. Details were presented demonstrating how inhibiting either these phosphatases or kinases regulates integrin phosphorylation, integrin-Dok1 interactions, cancer cell invasion, and invadopodia formation. Quantification and biochemical evidence for these adhesion interactions was presented using a novel FRET sensor for integrin phosphorylation- Illusia that was coupled with FLIM imaging, which was inspired by previous kinase FRET sensors. The authors also used a variety of additional FRET pairs, pulldown assays and western blots to verify key interactions in this phosphorylated integrin-Dok1 interactome. These cell biology studies overall were well executed and comprehensive although some critical details and controls are recommended, as outlined below.

However, despite the compelling in vitro studies and intriguing evidence for a kinase/phosphatase regulated phosphorylation adhesion complex regulating tumor cell invadopodia, evidence in support of the impact of this interactome on tumor cell phenotype and specifically metastasis in vivo is rudimentary. To begin with unfortunately the authors failed to conduct critical analysis of key metrics that would argue for any impact on tumor cell dissemination in vivo, and the in vivo models used for their experiments are less than ideal. It remains unclear which step during metastasis they maintain is critically modulated by integrin phosphorylation. Metastasis is a multi step process that depends upon efficient extravasation into the circulation, intravasation into the metastatic site and most importantly survival and expansion at the metastatic site to form a viable metastatic lesion. To be clear, tumor cell invasion into the parenchyma does not equate with metastasis and merely defines what is considered to be a malignant lesion (i.e. all malignant lesions by definition invade into the associated stroma but not all malignant lesions in fact generate metastatic tumors). It remains unclear what if anything informative the subcutaneous tumor xenograft studies presented in the first part of this manuscript reveal that wasn't already shown by the data presented in the in vitro collagen studies which clearly demonstrate an impact on tumor cell invasion. At the very least the authors should have quantified circulating tumor cells. Such a metric would have made a good case if there were

quantifiable differences for an impact on potential tumor metastasis. In addition, the rationale for the tail vein studies remains ill-defined as this manipulation essentially monitors for metastatic outgrowth in the lungs and to a smaller extent to intravasation into the metastatic tissue, which are different criteria to that expounded upon in the earlier tumor *in vivo* studies and argued by the invadopodia studies. It is also somewhat disappointing that the authors confined the *in vivo* metastasis studies only to subcutaneous xenografts and immunocompromised mice, especially given the large body of impressive data that has accumulated regarding the role of the immune system in cancer metastasis and specifically to dissemination and metastatic outgrowth. Finally, and importantly, while the cell biology work is well executed, and the authors should be commended, the conceptual advance remains modest so that while the results are certainly interesting, the novelty is not evident.

Au: We thank the reviewer for their positive assessment of our cell biology studies, and the expert suggestions regarding the *in vivo* work that further helped to strengthen the study by more clearly explaining the rationale. We fully agree with the reviewer that metastasis is a complicated multi-step process with each individual step representing a significant hurdle to cancer dissemination. Concordant with the cell biological and mechanistic focus of our work, we opted to employ *in vivo* assays that in our view support the assessment of these steps individually, rather than the full metastatic cascade. In doing this, we are aware that there are limitations when seeking to evaluate the full process of metastasis.

To explain our reasoning, for the first step of invasion into the tumour-proximal stroma, we chose the local invasion assay presented in Figure 1e. We felt this would have added value over the *in vitro* three-dimensional (3D) fibroblast-contracted collagen invasion assays, as the *in vivo* ECM is more complex, and has a greater range of stromal cell types to more accurately assess the process of invasion *in vivo*. As such, we included in the text a mention that this assay is "...to assess the role of integrin β 1 phosphorylation on local invasion *in vivo*, ..." (lines 97 to 98).

Another key step mentioned by the reviewer is the invasion from the circulation to the metastatic site. To assess this, we chose to employ the widely used tail-vein injection model to evaluate lung metastatic foci. We fully agree that this assay also monitors growth in the secondary organ, as well as survival of the cancer cells in circulation, and have indicated this in the text "To investigate the role of integrin phosphorylation dynamics during later stages of the metastatic cascade, we utilized an animal model of extravasation and metastatic colonisation of the lung" (line 501 to 503). In response to these valid criticisms, we have now also included careful evaluation of cancer cell survival in circulation-like conditions in a microfluidic assay (see below).

Finally, we fully acknowledge the important role of the immune system in cancer dissemination. However, since all the cellular assays are done with human cells, and the *in vivo* assays were set-up to validate these *in vitro* findings, we were limited to the use of immunocompromised mice, to avoid the immunogenic reaction of the mouse immune system to the human cancer cells. To acknowledge this limitation, we have included the following sentence into the discussion, "(these data) also open the door for future investigations into the possible role of integrin phosphorylation in modulating cancer cell crosstalk with immune cells in the tumour microenvironment, as well as in the circulation, with syngeneic cancer models and immunocompetent mice" (lines 626-630).

Regarding novelty, it is important to emphasize that we report a previously unknown, functionally relevant role for dynamic control of integrin phosphorylation and dephosphorylation in cell invasion. To the best of our knowledge, this is the first demonstration that Src directly phosphorylates ITGB1, and that Shp2 and PTP-PEST directly dephosphorylate ITGB1. Furthermore, Illusia is the first probe that enables visualization of integrin-tail phosphorylation in live invading cells. Finally, our data showing that Shp2 inhibitors have additional effects in reducing invasion and metastatic colonisation may become relevant considerations in future clinical trial designs for Shp2 inhibitors currently being assessed in mutant-KRas-driven cancers.

A number of clarifications and questions need to be addressed by the authors.

Major concerns:

1. Reconsideration of murine models to assess metastasis:

The authors conducted subcutaneous injection studies into immune-compromised mice and quantified invasion into the surrounding parenchyma as an indication of impact on tumor cell invasion in vivo. Subcutaneous injection does not reconstitute the stromal microenvironment of tumors (unless studying skin cancer) with much fidelity so it is unclear how this manipulation would reflect impact on tumor invasion in the mammary gland. It is recommended that the authors consider conducting fat pad injection studies which would better assess impact on tumor cell invasion into the breast tissue stroma. Moreover, as the data are currently presented it remains unclear what new information is obtained by merely measuring invasion in these tissues that was not already demonstrated using the collagen matrices in vitro? A malignant tumor is an invasive tumor! There is no correlation between extent of invasion of a primary tumor and metastatic potential. Tumor size yes! But tumor invasion that describes whether it is a benign tumor or a malignant tumor. A better metric to measure impact on metastatic potential would be measuring circulating tumor cell load. This is easily done and at the very least should be done. Furthermore, while not essential, clearly the use of immune competent mouse models is a much better system to use to assess impact on metastasis rather than these human lines into immune compromised mice.

Regarding the tail vein injection in Fig. 8e, the rationale for performing this experiment is confusing. There is no convincing indication as to whether the integrin YYFF mutation actually impact tumor cell metastasis and if this is mediated via its effects on invadopodia formation. The tail vein injection studies measure the ability of the tumor cells to survive and grow in the lungs of the mice and to a lesser extent some level of intravasation. The impact on invadopodia could impact intravasation - however these assays over load the system with a bolus of cells and essentially swamp the lungs with cells. Even nonmalignant cells will get into the lungs with this type of manipulation. The difference is that nonmalignant cells won't grow to form viable metastatic lesions. What would be very constructive is if the authors were to quantify micro and macro metastasis as well as metastatic area. This analysis would reveal whether or not the cells are competent to gain access (intravasate) into the lungs but fail to grow out (micro mets) or are able to gain access and grow out (macro mets). Metastatic area would also be reflected in this analysis. In addition, conducting these studies in an immunocompetent mouse would provide much needed evidence in support of the impact of this phosphorylated

interactome on metastatic potential of tumor cells. It is very possible that the impact of integrin phosphorylation extends well beyond classic invadopodia formation!

Au: We agree with the reviewer that the use of immunocompetent mouse models provides a more physiologically relevant assay for assessment of metastatic dissemination. However, generating syngeneic lines to use with immunocompetent mouse strains is beyond the scope of this study, particularly due to the risk that generating mutations in ITGB1 restores immunogenicity (PMID: 34861158). Thus, to perform experiments in immunocompetent models would require the generation of genetically engineered mouse models with these NPxY(Y783F) and NPxY(Y795F) mutations in ITGB1 to then be crossed to a breast cancer model, such as the MMTV-PyMT system. While this is an exciting option for future work, we believe that the cell biology work that was commended by the reviewer is of greater relevance to the journal and should remain the focus of the study.

The goal of the animal experiments conducted in this study was to provide orthogonal validation of the 3D assays in different systems. As the reviewer noted, the fibroblast-contracted collagen matrices provide a powerful platform for assessing invasion in a 3D setting. However, the assay includes only two cell types, fibroblasts and cancer cells, and growth factors are supplied from culture media, as with any *in vitro* system. Thus, we chose to validate our findings in a more relevant model by injecting cells as xenografts into the flank of nude mice. This allowed us to assess the invasive capacity of the breast cancer cells in a more representative ECM and cellular environment to that of the native disease state. In this way, we were able to reproduce our findings and support the *in vitro* results. While we agree that orthotopic injection might provide a more relevant initial environment for assessment of breast cancer invasion, when compared to the subcutaneous xenografts performed, the modification of the tumour microenvironment by the cancer cells at either site would quickly normalize the system. Thus, we do not feel that using further animals to reproduce this finding in the mammary fat pad is ethically responsible or beneficial to the findings of the manuscript.

The reviewer mentions that assessing circulating tumour cell (CTC) load would be a better metric. However, detection of CTCs from MDA-MB-231 (MM231s) xenografts requires orthotopic injection, as they are not detectable from subcutaneous xenografts (PMID: 35216615). However, we do agree that assessing the metastatic potential of these cells with ITGB1 (WT or YYFF), or with disruption of ITGB1 phosphorylation dynamics through Shp2 inhibition, provides an important link between reduced invasive ability and changes in the ability of cells to perform the next steps in the metastatic cascade. For this reason, we performed the tail vein experiments (Fig. 8e-i) to investigate the role of ITGB1 phosphorylation in this phase of the metastatic cascade. Prior to performing this assay, we discussed the appropriate number of cells to inject to avoid “swamping” the lungs. Indeed, for another project the lungs had been fixed and imaged after tail vein injection of different cancer cells lines, and using 500,000 cells, it was clear that only very few made it to the lungs after injection (Reviewer-only Fig. 8). For this reason, we injected 750,000 cells/mouse to ensure that sufficient cells reached the lungs to then allow us to track the colonisation (Extended Data Fig. 8e) over 24 hours, as well as to quantify metastatic growth through assessment of the number of pulmonary nodules (Fig. 8h) and cell number using qRT-PCR (Fig. 8i).

Reviewer-only Figure 8. Representative images of mouse lungs from 7–8-week-old female athymic nude mice (*Foxn1^{nu}*, Envigo, UK) 15–20 minutes after lateral tail vein injection of 500,000 AsPC-1, MIA PaCa-2 or Panc 10.05 PDAC cells stably expressing EGFP/Luciferase. Scale bar: 500 μ m, Insets: 100 μ m.

Further to the use of the tail vein assay to assess metastatic colonization of the lungs, we set up an *in vitro* microfluidic assay to mimic the non-adherent, flow-induced stress condition of vessels (PMID: 29634935). These data indicated that the MM231 cells with or without flow observed no significant changes in anoikis with modulation of integrin phosphorylation (Extended Data Fig. 8j, provided below for convenience).

Extended Data Figure 8j.

Expression of invadopodia components is co-regulated at the protein level.

j, Flow cytometric analysis of annexinV-stained MM231s with ITGB1(WT or YYFF) and treated with either DMSO or SHP099 (SHP; 100 nM) while grown overnight in suspension on ultra-low attachment plates and either subjected to flow and/or treated with SHP099 (100 nM) or DMSO control (negative (-ve): unstained negative control samples; positive (+ve): MM231s treated with a cell-death-inducing cocktail of doxorubicin (10 μ g/ml), VO43- (50 μ M), gemcitabine (10 μ M); n = 3-4 biological replicates/cell line/condition; Statistics from a one-way ANOVA with a Tukey correction for multiple comparisons; NS, not significant).

2. It is recommended that the author improve the characterization of the shRNA mediated integrin beta1 knockdown:

The authors show an ~90% reduction in integrin beta1 expression with the shB1. However, no significant reduction in proliferation was seen, which is different from previous studies (Hou et al, Scientific Reports, 2016; Grzesiak et al, Cancer Therapy, 2011). The authors are advised to check proliferation and adhesion formation on different beta1 specific substrates such as Type 1 collagen, Fibronectin and Laminin to better understand the effect of beta1 knockdown and mutant on basic cell functions such as proliferation, adhesion formation, and 2D cell migration.

Au: It is true that in some studies integrin $\beta 1$ (ITGB1) knock-down has led to slight reductions in proliferation *in vitro* within very defined culture conditions. In the studies cited above, the pancreatic cancer cells showed some ligand-specific proliferative defects, but on tissue-culture plastic no effect was observed (PMID: 21491421), which is consistent with our own findings (Extended Data Fig. 1b,c). Similarly, when breast cancer cells were used (PMID: 26728650), the cells grew slightly faster after integrin $\beta 1$ knock-down on fibronectin (FN) and laminin (Lam) coated substrates. Of note, the pancreatic cancer cells were grown in serum-free media to reduce the effect of ECM proteins contained in the serum (PMID: 21491421), and the breast cancer cells were initially grown in serum-free, but after 24 hours serum was added to the media, ultimately changing the ligands present in the tissue culture wells (PMID: 26728650). The challenge of maintaining a single-ECM ligand over a long period of time, where serum that contains ECM proteins is desired to maintain the proliferative capacity of the cells, and the cells themselves secrete their own ECM proteins, presents a significant obstacle to accurate experimental design. To attempt to overcome this challenge, we used FN-depleted serum to reduce the effect of FN when assessing proliferation on single ECM ligands. Importantly, this FN-depleted serum was shown to support the growth of MDA-MB-231 (MM231) cells as well as full serum, and the cells grew poorly in serum-starved conditions (Reviewer-only Fig. 9).

Reviewer-only Figure 9. Quantification of relative proliferation of MM231 breast cancer cells seeded in FN-coated or uncoated 96-well wells and imaged for 4 days with an IncuCyte S3 Live-Cell Analysis system using a 10x objective (n=1 biological replicate, triplicate 24-well wells). Starting from 10,000 cells/well.

Using this FN-depleted serum, we seeded MM231s for 16 h on collagen I (Coll), FN or Lam, and observed an increased proliferative capacity for ITGB1-depleted MM231s on FN, which was not observed on Coll or Lam (Reviewer-only Fig. 10). While $\alpha_5\beta_1$ has been shown to be important for the initial binding to the RGD-motifs present in FN during nascent adhesion formation, there are several other RGD-binding integrin heterodimers that form larger and more stable integrin-adhesion complexes (PMID: 23708002), and this may account for the greater proliferative capacity in this condition.

Reviewer-only Figure 10.

Knock-down of ITGB1 in MM231 cells has no significant effect on proliferation on Coll or Lam, but increases proliferation on FN.

Representative images (left) and quantification (right) of MM231 parental or shβ1 cells seeded for 16 h on collagen I (Coll), fibronectin (FN) or laminin (Lam) in complete media containing FN-depleted serum, prior to fixation and staining for the proliferation marker Ki67 (green) and actin (magenta; n=3 biological replicates; 10 regions/condition/replicate; significance assessed by a Kruskal-Wallis test with a Dunn’s correction for multiple comparisons; ***p<0.001, NS, not significant). Scale bars, 20 μm.

As an additional characterization prompted by a comment from reviewer 1, we also assessed the phenotype of MM231 cells embedded in 3D collagen gels (Fig. 1d, reproduced below for convenience). In this experiment, we observed a clear increase in the roundness of the shβ1 cells, which were less able to spread out in the 3D collagen gel, when compared to the parental MM231s. Similarly, the MM231s expressing the phosphorylation-defective mutant ITGB1(YYFF) were found to have a significant increase in their roundness in 3D gels.

Figure 1.

ITGB1 phosphorylation supports efficient cancer cell invasion.

d, Representative images (left) and quantification of cell shape (i.e. solidity; right) of MM231 cells (green – actin staining) embedded in 3D collagen matrices overnight (magenta; n=3 biological replicates; 15-50 cells/condition/replicate; significance assessed using a Kruskal-Wallis test with a Dunn’s correction for multiple comparisons; NS, not significant, ***p<0.001). Scale bars, 20 μm.

3. Spatial localization of the integrin beta1 YYFF:

Activated integrins have been shown to co-localize with talin, while the non-phosphorylatable mutant YYFF has a lower affinity towards talin. However, in Fig, S1f,g the authors show a high colocalization between integrin YYFF and activated integrin beta1. They also show a higher staining of YYFF at the adhesions (marked by paxillin). It is strongly recommended that the authors address this disparity. I

Au: The reviewer is correct in that engaged integrins colocalise with talin, as talin is recruited to integrins as part of their activation (PMID: 30002479). However, talin has higher affinity towards the non-phosphorylated integrin, such that Src phosphorylation disrupts talin recruitment to integrins as the talin head PTB-domain cannot interact with tyrosine-phosphorylated integrin tail (PMID: 2468126; PMID: 12535520). Thus, the fact that we see a small increase in colocalisation between the extended open ITGB1 marker, 12G10, and ITGB1(YYFF), compared to ITGB1(WT) is concordant with these published data (Extended Data Fig. 1g, quantified in 1h; formerly Supplementary Fig. 1f & 1g). Moreover, non-phosphorylated ITGB1 fragments have been reported to exhibit increased interaction with recombinant talin head domain, compared to phosphorylated ITGB1 fragments (updated schematic in Fig. 3a; and published work, e.g., PMID: 19843520). This is consistent with our intermolecular FRET results, where we see a significant increase in interaction between the talin head domain fragment and the ITGB1(YYFF) construct, compared to the ITGB1(WT) control (Extended Data Fig. 2d). Further to this previously established interaction, Dok1 is shown to interact more strongly with phosphorylated ITGB1 (updated schematic in Fig. 3a), either using recombinant proteins (PMID: 19843520), or with our own FRET approach (Extended Data Fig. 2c). Given that this switch in binding from talin (unphosphorylated integrin) to Dok1 (phosphorylated integrin) was not clear, we have now added more details into the figure legends, included small FRET diagrams into figure panels, and an updated scheme in Fig. 3a to increase the clarity of the data presented. In the manuscript text, we also indicate the following: “Notably, the reverse was observed for a talin-1 head domain fragment (F₀-F₃ from mouse *Tln1*), which showed an increased interaction with the non-phosphorylatable YYFF receptor (Extended Data Fig. 2d), and is in line with talin-1 having a reduced affinity for phosphorylated integrins $\beta 1$ and $\beta 3^{11}$ ” (lines 156 to 160). This is also in response to the reviewer’s comment 5, and minor comment 8 below.

The authors should also quantify the Illusia FRET efficiency in vs outside IACs (using talin as a marker) to quantify the spatial localization of integrin phosphorylation events.

Au: We agree with the reviewer that assessment of the spatial localisation of integrin phosphorylation would be an important area for future research. When we were initially developing Illusia, we tested the functionality of the ITGB1-targeting peptide, which would allow us to track changes in ITGB1 in response to various perturbations, as well as to see where and when the phosphorylation occurs. However, the targeting peptide retained Illusia in the Golgi, where ITGB1 is recruited by alpha subunits prior to reaching the cell membrane. Given this limitation, we instead applied an acylation substrate sequence derived from Lyn kinase to recruit Illusia to cholesterol-rich membrane regions where fluorescent reporters for focal adhesion kinase (FAK) have shown the greatest sensitivity (PMID: 21792185). This forced recruitment allowed the development of Illusia towards the siRNA FRET screen presented in the manuscript, but limits our ability to interpret the spatial localisation of integrin phosphorylation in IACs. So, while Illusia has allowed us to make a significant step forward in our understanding of the regulation of integrin phosphorylation, work is still required to understand the precise spatial location of ITGB1 when phosphorylated.

The authors should also investigate if integrin phosphorylation correlates with invadopodia formation.

Au: We agree with the reviewer, and in the previous manuscript submission we included data that looked at Src-induced invadopodia formation, which was dramatically reduced in MM231 cells that had non-phosphorylatable ITGB1(YYFF), when compared to the ITGB1(WT)-expressing cells (Extended Data Fig. 7c). These data demonstrate a correlation between integrin phosphorylation and invadopodia formation. We also found that several invadopodia components, notably TKS5, Cofilin and CTTN, showed a significant correlation between their expression and that of Dok1, which is a key adaptor protein for mediation of ITGB1 functions when phosphorylated (Extended Data Fig. 8b). We regret that we have not been able to directly explore integrin phosphorylation and invadopodia formation using imaging. Unfortunately, the ITGB1-phosphorylation-specific antibody is not suitable for immunofluorescence. Furthermore, in line with the Reviewer's comment 7 below, we have also expanded the discussion to include a section on the importance of ITGB1 phosphorylation in the recruitment of invadopodia proteins to the invadopodium, "It is important to note that the two conserved cytoplasmic NxxY motifs are present in multiple integrin β -subunits and Dok1 can interact with them⁴⁶. The significant decrease in invadopodia observed after Dok1 knock-down (Fig. 7e) suggests that Dok1 could be a core component for invasion mediated by multiple integrins and implies a wider impact of our findings" (lines 602 to 606).

The localization of integrin phosphorylation with cell migration- leading vs lagging edge should also be commented on.

Au: We agree with the reviewer, and in response to both this comment, and that of reviewer #1, we have now expanded the discussion to include a commentary on the role of integrin phosphorylation in the polarization of integrin functions during cell migration, "Integrins and their downstream signalling cascades regulate cell migration and adhesion on several scales. At the single-molecule level, their dynamic binding and unbinding to ECM ligands, and linkage to the acto-myosin cytoskeleton, define the cell-specific optimal conditions for cell migration³⁶. On the scale of the whole cell, front-rear polarity establishment is regulated by integrins and their associated signalling^{37, 38}. At this scale, live-cell imaging of Illusia in invading cells appears to show front-rear differences in phosphorylation, with flickering and regional differences at the leading and trailing edges (Supplementary Video 2). This implies that integrin phosphorylation may be polarised in cells, which may contribute to cell migration that is likely co-ordinated by rapid phosphorylation cycles of individual integrin molecules. Future studies are needed to interrogate how this is governed across scales, and the fine-tuning of this process by tyrosine kinase/phosphatase networks" (lines 567 to 578).

4. External regulation of integrin phosphorylation:

The authors intensively investigate the internal factors affecting integrin phosphorylation. However, cell external factors also regulate integrin phosphorylation. Since stiffness affects invadopodia formation (Chang J. et al, Biophys J., 2020), does integrin phosphorylation take place primarily on softer surfaces? It would be useful to show if integrin phosphorylation anti-correlates with substrate stiffness. Changing stiffness of collagen matrix in Fig 1b, for example, can help show if the change in invasion is due to invadopodia formation. The effect of other chemotactic ligands which regulate invadopodia formation on integrin phosphorylation can also be looked at.

Au: The reviewer makes an interesting point. We have primarily assessed changes in integrin phosphorylation on stiff glass substrates in the manuscript, while also observing reduced cancer invasion using 3D fibroblast-contracted collagen matrices (Fig. 1b,c), which have a stiffness close to that of the invasive front of invasive ductal carcinoma (~5 kPa; PMID: 29203823; PMID: 32503141). Similarly, in the mixed stiffness environment of our xenograft and lung metastasis mouse models, we again see a reduction in the ability of cells to invade locally (Fig. 1e) or form metastasis at secondary sites (Fig. 8e-i) respectively. Together, these data suggest that integrin phosphorylation is important in a range of stiffness contexts. In line with the reviewer's suggestion, we have now also assessed whether the total level of ITGB1 phosphorylation increases on softer substrates by seeding MM231 cells with Illusia on hydrogels of defined stiffness (Fig. 6a & 6b, also included below for convenience). This suggests that cells on softer substrates increase their ITGB1 phosphorylation and may explain the requirement for dynamic regulation of integrin tail phosphorylation in cells invading within the softer external environments present in the 3D fibroblast-contracted collagen matrices and mouse lungs. This is a new result that will open doors for future investigation by many in the field of mechanobiology and adhesion. We are extremely grateful to the reviewer for this exciting suggestion that helps strengthen our study further.

Figure 6.

Loss of ITGB1 phosphorylation dynamics results in equivalent phenotypes for Src and Shp2 inhibition.

a-b, Representative FLIM images (a, left) and apparent FRET efficiencies (a, right) and cell area (b) of MM231 cells stably expressing Illusia and seeded on either glass, or hydrogels with different stiffness (60 kPa, 2 kPa or 0.5 kPa; n=4 biological replicates; 25-31 (a) 23-48 (b) cells/condition/replicate; significance assessed using a one-way ANOVA with a Tukey correction for multiple comparisons; NS, not significant; *p<0.05, ***p<0.001). Scale bars, 10 μm.

5. Non consistent FRET results:

In 'no treatment' conditions, FRET efficiency of YYFF should be lower than WT as shown in Fig. S2c. However, in Fig 3e and Fig. S3d, FRET efficiency of WT vs YYFF (without VO4) looks similar. The authors should address this disparity.

Au: We apologise for the lack of clarity in the figure presentation. The data presented in Fig. S2c uses wild-type (WT) and non-phosphorylatable (YYFF) ITGB1, a.k.a. ITGB1(WT) or ITGB1(YYFF) respectively. These constructs were expressed in MM231 breast cancer cells that have had their endogenous ITGB1 knocked-down using an shRNA. Thus, the cells now have either ITGB1(WT) or ITGB1(YYFF) only, and these constructs have mRuby2 tags, which are red fluorescent proteins that can function as effective FRET acceptors for green fluorescent protein variants, such as Clover. In Fig. S2c, we overexpressed a

Dok1 construct with a Clover tag and observed greater FRET efficiency between the ITGB1(WT)-mRuby2 and Dok1-Clover, than between the ITGB1(YYFF)-mRuby2 and Dok1-Clover.

As for Fig. 3e and Fig. S3d, these FRET experiments were performed using our newly developed Illusia FRET reporter, which reads out the phosphorylation state of ITGB1 through a phosphorylation-dependent conformational change that pulls the cyan fluorescent protein variant, mTurquoise2, and yellow fluorescent protein variant, YPet, further apart upon phosphorylation of a cytoplasmic fragment of ITGB1 that is included in the reporter. In order to demonstrate the phosphorylation-dependence of this change, we created a mutant Illusia construct where the NPxY sites in the cytoplasmic domain fragment (aa772-798) were mutated to create Illusia(YYFF). We understand that this nomenclature is confusing, as the ITGB1 mutant is also referred to as ITGB1(YYFF), but we feel that it is also important to keep the clear link between this part of the ITGB1 tail containing both of these fragments. Furthermore, as in your minor comment 8 below, we have now added more details into the figure legends, and small FRET diagrams into figures, to increase the clarity of the FRET pairs used.

6. Role of integrin beta3 phosphorylation in invadopodia formation:

Role of integrin beta3 in invadopodia formation has been shown before (Peláez, R. et al., PLoS One, 2017; Feng Z. et al, PNAS, 2021). Since integrin beta3 is also a substrate for Src kinase and PTP-PEST phosphatase, phosphorylation of integrin beta3 might also be involved in invadopodia formation. It is recommended that the authors investigate the role of integrin beta3 in invadopodia formation.

Au: We agree with the reviewer that there are many other proteins involved in invadopodia formation and several integrin receptors have also been linked to these structures. Indeed, it has previously been suggested that integrin $\beta 3$ is upregulated to compensate for the loss of ITGB1 in some cell line models (PMID: 11598197; PMID: 21491421). However, when we assessed the surface levels of integrin $\beta 3$ after shRNA-mediated ITGB1 knock-down, we observed no significant changes in integrin $\beta 3$ surface levels (Extended Data Fig. 1e, quantified in 1f), suggesting that integrin $\beta 3$ is not compensating for ITGB1 loss in our cells. Furthermore, in cells where we rescue ITGB1 levels using ITGB1(WT or YYFF) constructs, we find that induction of invadopodia formation through overexpression of a constitutively-active Src construct is significantly less effective in cells with the ITGB1(YYFF) construct (Extended Data Fig. 7c). This suggests that while ITGB1 phosphorylation is not essential for the formation of invadopodia, other components are able to support their formation when this is not possible. In line with this observation, we have added further clarification to the Results section to address this point, “Notably, silencing of Dok1 resulted in a more pronounced phenotype than non-phosphorylatable ITGB1, suggesting that the role of ITGB1 can be compensated by other adhesion receptors, but that the scaffolding functions of Dok1 are less dispensable for invadopodia formation. This is further supported by the established phosphorylation-specific binding of Dok1 to other integrin β isoforms^{11, 35}” (lines 450 to 455). We thank the reviewer for supporting us to improve the presentation of our findings.

7. Mechanism of integrin beta1 phosphorylation dependent invadopodia formation:

It is advised that the authors describe the mechanism of integrin beta1 phosphorylation-dependent invadopodia formation through recruitment of invadopodia proteins in the discussion. This is perhaps

that main novelty presented in the manuscript and is unfortunately not as well developed as it could be. In this regard, the authors are in a good position to explain the previously unknown link between a 'switch on signal' to invadopodia formation, so that adding this link would greatly improve the impact of the work presented in the manuscript.

Au: We thank the reviewer for their suggestion and have now expanded the discussion, "Here, we describe integrin phosphorylation as an apparent 'switch on signal' to invadopodia formation. Integrin phosphorylation recruits Dok1 to the cytoplasmic tail of the receptor and Dok1 seemingly facilitates complex formation with invadopodia components CTTN, Cofilin and TKS5" (lines 621 to 624), to highlight the importance of ITGB1 phosphorylation in the recruitment of invadopodia proteins to invadopodia.

Minor concerns:

1. The authors should clarify the mutations in their first mention (line 78-79).

Au: The exact mutations (Y783F and Y795F) have been added into the revised manuscript text (see line 81).

2. In Fig 1d legend: The authors should clearly mention what the dotted black line is, what the * means, 'n' and the statistical analysis used.

Au: The legend has now been corrected to include the relevant details (see revised Fig. 1e, formerly Fig. 1d, legend). Furthermore, in line with the Springer Nature publishing guidelines, all of the statistical source data and exact p values are now provided as a separate supplementary file (see new Supplementary Table 6).

3. In Figure S2d, the authors use GFP-Talin head (F0-F3) as a control. However, Talin head does not specifically bind integrins, and has a PIP2 binding function as well (Chinthalapudi K. et al, 2018, PNAS). This would affect the FRET efficiency. Using full length talin and quantifying the FRET at IACs might serve as a better control because of its specific localization at the IACs.

Au: We thank the reviewer for this point. There seems to be some conflicting data on talin head binding to integrins and PIP2, possibly owing to different experimental set ups and distinct read-outs. We have shown, using reconstituted recombinant proteins on liposomes, that talin head binds PIP2 and ITGB1-tail and when both are present on the membrane, they function in a synergistic manner to elevate talin-head liposome interaction 10-fold (PMID: 30072441). Here, we demonstrate that an F0-F3 domain construct of the talin head domain interacts significantly more with the phospho-deficient ITGB1(Y7FF) mutant, when compared to MM231 cells expressing the wild-type (WT) ITGB1(WT) construct (Extended Data Fig. 2d). This supports the previous finding that recombinant talin fragments bind more strongly to non-phosphorylated ITGB1 cytoplasmic domain peptides (PMID: 19843520). It also allowed us to demonstrate that the ITGB1(Y7FF) construct is capable of acting as an acceptor for intermolecular FRET, which has reduced interaction (lower apparent FRET efficiency) with a fluorescently-tagged Dok1 construct (Extended Data Fig. 2c). This was again in line with work using a recombinant Dok1 phosphotyrosine-binding domain (PTB) and peptides from the ITGB1 cytoplasmic

domain (PMID: 19843520), which saw reduced binding when integrin was not phosphorylated, the opposite to talin. While we fully agree with the reviewer that full-length talin possesses many biological functions beyond the talin head-ITGB1-tail interaction, we feel that the talin F0-F3 head domain construct is nevertheless a valid control and comparison for the Dok1-ITGB1 intermolecular-FRET assay in this study.

4. In Fig 2f, the authors should add statistics.

Au: In line with the reviewer's suggestion, we have now added an additional replicate of the pY ELISA to allow complete statistical analysis to be performed (See revised Fig. 2f; Data included below for convenience). This additional data further validates the direct phosphorylation of ITGB1 by Src kinase and we thank the reviewer for the opportunity to cement our claim.

Figure 2. e, Scheme of ELISA for phospho-tyrosine (pY). **f**, ELISA for changes in ITGB1 phosphorylation using recombinant ITGB1 peptide and Src kinase in the absence or presence of ATP and the Src inhibitor saracatenib (Sara; 1 μ M; n=3 biological replicates; triplicate wells/replicates; significance assessed using a one-way ANOVA with a Šidák correction for multiple comparisons; ***p<0.001). Unphosphorylated ITGB1 and phosphorylated ITGB1 p(Y783) peptides were included as negative and positive controls, respectively. Data are mean \pm SEM.

5. The authors should add densitometry measurements for ITGB1(Y783)/ITGB1 western blots in Fig. S4e,f and Fig. S5e.

Au: We have now added densitometry for the western blot data presented in Extended Data Fig. 5f (formerly Supplementary Fig. 5e), in response to the above comment, as well as for Extended Data Fig. 5c & 2i, in response to a similar request for densitometry from reviewer #1.

As to the densitometry for the western blots in Extended Data Fig. 4e and 4f, this was included in the previous submission in Fig. 4d and 4e and referenced in the text as such on line 286. Given the lack of clarity in the previous manuscript related to these western blots, we have now updated the figure legends for Fig. 4d and 4e, and Extended Data Fig. 4e and 4f, to reference the location of the quantification and representative western blots between the two figures (lines 304 to 305, in the revised manuscript, and lines 137 to 140 in the revised Extended Data). We have also added densitometry for pITGB1(Y783) to show the significant upregulation after silencing of Shp2/*PTPN11* or PTP-PEST/*PTPN12* individually (Extended Data Fig. 4e and 4f).

6. The images in Fig 8g do not look representative for YYFF since in the graph, the average # pulmonary nodules are roughly equal in SHP099 vs YYFF, while the images show a stark difference.

Au: We understand the reviewer's concern and have now updated the representative image for the colonization of the lungs by the MM231 ITGB1(YYFF) cells (Fig. 8g, included below for convenience).

Fig. 8.
The Dok1/ITGB1 complex recruits cofilin and other invadopodia components to adhesion sites, to mediate efficient cancer dissemination.

e, Representative images of mice with MM231 ITGB1(WT or YYFF) cells stably-expressing the luciferase/EGFP construct. Oral gavage of Vehicle or SHP099 (100 mg/kg) proceeded for 5 days from the day of injection. **f**, Box and whiskers plot highlighting the endpoint metastatic burden as an average (avg) radiance value from the luciferase signal of the MM231 cells in **e** (n=9 mice tracked/group). **g**, Representative lung sections stained for EGFP-positive MM231 cells. Scale bars, 2 mm; insets: 200 μ m. **h**, Quantification of pulmonary nodule number (i.e. clusters of >10 cells) in lungs from EGFP-positive MM231 cells (n=10 mice/group). **i**, qRT-PCR of RNA samples collected from the MM231 ITGB1(WT or YYFF) cells stably-expressing the luciferase/EGFP construct. Mice were designated as either “Metastatic” or “Low signal” after setting a threshold for “Metastatic” as having an expression fold change >1 compared to the mean of the WT/Vehicle control with human GAPDH normalised to mouse/human GAPDH (n=10 mice/group).

7. In lines 459-462, “Together, we propose a mechanism where spatially and temporally controlled integrin phosphorylation...” Spatial control of integrin phosphorylation was not achieved by the authors. The authors should explain/modify this statement.

Au: We understand the reviewer's concern that we did not spatially control the changes in integrin phosphorylation in the study, but we believe that there is spatiotemporal control of integrin phosphorylation that supports the optimal function of integrin adhesion receptors. Regardless, to tone down the statement and keep within the findings of the study, we have replaced “spatially and temporally” with “finely” (line 522 of the revised manuscript text), to reflect the balance between states that we identified as important for efficient breast cancer invasive progression.

8. In general, since the figures use a variety of FRET donor-acceptor pairs, it becomes difficult to understand the FRET efficiency graphs without the donor-acceptor pairs mentioned in the graphs. Including this will help increase the readability of the graphs.

Au: We appreciate this point and thank the reviewer for helping us improve the clarity of our data representation further. We understand that the different FRET systems applied might be confusing for the reader and have now added additional annotations to the figures to indicate the donor-acceptor pairs used, as suggested. These schemes now clearly indicate for each panel the use of intermolecular FRET using GFP-/RFP-variants to assess protein-protein interactions, or when Illusia (i.e. intramolecular FRET between mTurquoise2 and YPet) is applied to assess changes in integrin phosphorylation. Similarly, in line with reviewer #2's comment related to the labelling of the lifetime colour code, we have now modified the labels for these colour codes to always indicate either "interaction" or "phosphorylation" to demarcate the different applications of the FRET systems used throughout.

Response to Reviewers' Comments:

Reviewers' Comments:

Reviewer #1 (Remarks to the Author):

The authors have gone to great lengths to address the comments of all reviewers.

Au: We are encouraged that both reviewer #1 and reviewer #2 feel that we have satisfactorily addressed their previous concerns.

My only comment is that I am surprised that the Illusia reporter indicates increased phosphorylation on softer substrates (new data Figure 6a), which is at odds with other reports (e.g. Src activation with stiffer substrates : such as in review: Forcing a growth factor response – tissue-stiffness modulation of integrin signaling and crosstalk with growth factor receptors (DOI: 10.1242/jcs.242461) and stiffer substrates correlate with increased tyrosine phosphorylation e.g. (DOI: 10.1073/pnas.94.25.13661). Additionally, in our hands, we also see global tyrosine phosphorylation increase on stiff matrices vs soft. This does not mean ITGB1 follows that trend, and there could be technical differences. Therefore, could the authors use the same methodology as other papers - i.e. bulk Western blot, to confirm their result with the ITGB1 phospho ab? I just really want to make sure their reporter is reporting what is claimed. Could the authors comment on this? Could it also reflect the localisation of the FRET sensor?

Reviewer #1's comments on authors' responses to Reviewer #3's previous concerns:

The authors have done a very good job in addressing the concerns of reviewer 3, including toning down some claims, clarifying figures and legends, and carrying out new experiments. The concerns around mouse models are beyond the scope of the study; the metastasis model used is standard in the field, and use of immunocompetent mice would complicate matters further, and make an entirely new story.

Au: We are heartened that the reviewer thinks that we have done a good job addressing the concerns of reviewer #3. Related to the increased integrin \$\beta 1\$ (ITGB1) phosphorylation result, we were initially also surprised by this result; indicated by a decrease in FRET from Illusia, relative to cells on stiffer substrates (Fig. 6a,b). The review cited by the reviewer (DOI: 10.1242/jcs.242461) suggests that integrin phosphorylation may be increased on soft substrates in response to the lower number of mature adhesions, and our data is in line with that hypothesis. We find that the cells on the soft hydrogels are less spread with few mature adhesions. The other paper they cite (DOI: 10.1073/pnas.94.25.13661) is in our view a bit problematic, as it relies on a pY antibody, which would not allow detection of phosphorylated ITGB1, as the bound Dok1 would block the antibody binding site, and this is the reason that we haven't used immunofluorescence with either non-specific pY or slightly-more-specific ITGB1(Y783) antibodies. This limitation was one of the main reasons we set about designing and validating Illusia. As such, and in response to the previous comments by both reviewer #1 and reviewer #2, we provided additional controls for the effectiveness of Illusia in reporting on ITGB1 phosphorylation state (Extended data Fig. 3e,h, along with additional densitometry for all western blots), and these were found to be satisfactory by both reviewers. This is in addition to

the controls used throughout Figure 2, Figure 3, and the Extended Data Figures 2 and 3, where we carefully compare Illusia reporter readouts with bulk Western blot and observe the same effects. Thus, we do not feel like a bulk western blot is necessary to again validate the fidelity of Illusia to assess ITGB1 phosphorylation state between the soft and stiff conditions.

However, we do agree that the interesting response on soft is worthy of further commentary, as suggested by the reviewer. The main conclusion of our study is that integrin phosphorylation isn't just Src activation, but other kinases, such as Arg, and the balance that they share with the protein tyrosine phosphates; the levels of which have not been assessed between soft and stiff hydrogels. In line with this, we have extended the discussion in results section (lines 358 to 363, also included below for convenience with new sections highlighted in yellow) to expand upon the significance of the finding, and its relationship to other published results.

"This prompted us to investigate whether stiffness influences integrin phosphorylation. We seeded Illusia-expressing cells on hydrogels of different stiffness, closer to that found in mammalian tissues and 3D cultures^{26,27}. By doing so, we observed increased integrin phosphorylation at lower stiffnesses (Fig. 6a), occurring in parallel with a reduced cell area (Fig. 6b) and together suggesting that adhesion in softer environments, such as 3D and mouse models, may have a greater reliance on integrin phosphorylation than typical 2D cultures on glass. This is in line with previous work demonstrating a continued reliance on Src and FAK for survival of cancer cells in conditions with reduced adhesion^{28,29}. Indeed, it has been previously suggested that integrin phosphorylation may be increased on soft substrates in response to the lower number of mature adhesions, and our data is in line with that hypothesis³⁰."

Similarly, we have added further commentary into the discussion (lines 611 to 619, also included below for convenience), along with the additional references suggested by the reviewer (new reference 30 and 50) and further references to support the expanded commentary (references 28, 29, 51 and 52).

"Furthermore, the finding that ITGB1 phosphorylation is increased on softer substrates (Fig. 6a,b), was initially surprising given previous findings that global tyrosine phosphorylation is reduced in softer environments⁵⁰. And yet, primary human fibroblasts have been shown to increase their invadopodia formation on softer hydrogels⁵¹, and Src activation state is maintained in both soft and stiff conditions in both cancer and fibroblast cell lines^{51,52}. Taken together, these data affirm the significance of ITGB1 phosphorylation dynamics for cell invasiveness and highlight the need for continued investigation into the role of adhesion signalling in different stiffness environments."

We thank the reviewer for their initial comments, but also for their expanded role in assessing our response to the concerns of reviewer #3.

Reviewer #2 (Remarks to the Author):

I am satisfied with the authors' responses to my questions and comments.

Au: We thank the reviewer for the contributions to the betterment of our manuscript.